# Deciphering the regulatory genome of *Escherichia coli*, one hundred promoters at a time

William T Ireland[1], Suzannah M Beeler[2], Emanuel Flores-Bautista[2], Nicholas S McCarty[2], Tom Röschinger[3], Nathan M Belliveau[2†], Michael J Sweredoski[4], Annie Moradian[4], Justin B Kinney[5], Rob Phillips[1,2]*

[1]Department of Physics, California Institute of Technology, Pasadena, United States; [2]Division of Biology and Biological Engineering, California Institute of Technology, Pasadena, United States; [3]Division of Chemistry and Chemical Engineering, California Institute of Technology, Pasadena, United States; [4]Proteome Exploration Laboratory, Division of Biology and Biological Engineering, Beckman Institute, California Institute of Technology, Pasadena, United States; [5]Simons Center for Quantitative Biology, Cold Spring Harbor Laboratory, Cold Spring Harbor, United States

*For correspondence:
phillips@pboc.caltech.edu

Present address: †Howard Hughes Medical Institute and Department of Biology, University of Washington, Seattle, United States

Competing interests: The authors declare that no competing interests exist.

**Abstract** Advances in DNA sequencing have revolutionized our ability to read genomes. However, even in the most well-studied of organisms, the bacterium *Escherichia coli*, for $\approx 65\%$ of promoters we remain ignorant of their regulation. Until we crack this regulatory Rosetta Stone, efforts to read and write genomes will remain haphazard. We introduce a new method, Reg-Seq, that links massively parallel reporter assays with mass spectrometry to produce a base pair resolution dissection of more than a *E. coli* promoters in 12 growth conditions. We demonstrate that the method recapitulates known regulatory information. Then, we examine regulatory architectures for more than 80 promoters which previously had no known regulatory information. In many cases, we also identify which transcription factors mediate their regulation. This method clears a path for highly multiplexed investigations of the regulatory genome of model organisms, with the potential of moving to an array of microbes of ecological and medical relevance.

## Introduction

DNA sequencing is as important to biology as the telescope is to astronomy. We are now living in the age of genomics, where DNA sequencing has become cheap and routine. However, despite these incredible advances, how all of this genomic information is regulated and deployed remains largely enigmatic. Organisms must respond to their environments through the regulation of genes. Genomic methods often provide a 'parts list' but leave us uncertain about how those parts are used creatively and constructively in space and time. Yet, we know that promoters apply all-important dynamic logical operations that control when and where genetic information is accessed. In this paper, we demonstrate how we can infer the logical and regulatory interactions that control bacterial decision making by tapping into the power of DNA sequencing as a biophysical tool. The method introduced here provides a framework for solving the problem of deciphering the regulatory genome by connecting perturbation and response, mapping information flow from individual nucleotides in a promoter sequence to downstream gene expression, determining how much information each promoter base pair carries about the level of gene expression.

The advent of RNA-Seq (*Lister et al., 2008*; *Nagalakshmi et al., 2008*; *Mortazavi et al., 2008*) launched a new era in which sequencing could be used as an experimental read-out of the

biophysically interesting counts of mRNA, rather than simply as a tool for collecting ever more complete organismal genomes. The slew of 'X'-Seq technologies that are available continues to expand at a dizzying pace, each serving their own creative and insightful role: RNA-Seq, ChIP-Seq, Tn-Seq, SELEX, 5C, etc (*Stuart and Satija, 2019*). In contrast to whole genome screening sequencing approaches, such as Tn-Seq (*Goodall et al., 2018*) and ChIP-Seq (*Gao et al., 2018*), which give a coarse-grained view of gene essentiality and regulation respectively, another class of experiments known as massively parallel reporter assays (MPRA) have been used to study gene expression in a variety of contexts (*Patwardhan et al., 2009*; *Kinney et al., 2010*; *Sharon et al., 2012*; *Patwardhan et al., 2012*; *Melnikov et al., 2012*; *Kwasnieski et al., 2012*; *Fulco et al., 2019*; *Kinney and McCandlish, 2019*). One elegant study relevant to the bacterial case of interest here by *Kosuri et al., 2013* screened more than $10^4$ combinations of promoter and ribosome-binding sites (RBS) to assess their impact on gene expression levels. Even more recently, the same research group has utilized MPRAs in sophisticated ways to search for regulated genes across the genome (*Urtecho et al., 2019*; *Urtecho et al., 2020*), in a way we see as being complementary to our own. While their approach yields a coarse-grained view of where regulation may be occurring, our approach yields a base-pair-by-base-pair view of how exactly that regulation is being enacted.

One of the most exciting X-Seq tools based on MPRAs with broad biophysical reach is the Sort-Seq approach developed by *Kinney et al., 2010*. Sort-Seq uses fluorescence activated cell sorting (FACS) based on changes in the fluorescence due to mutated promoters combined with sequencing to identify the specific locations of transcription factor binding in the genome. Importantly, it also provides a readout of how promoter sequences control the level of gene expression with single base-pair resolution. The results of such a massively parallel reporter assay make it possible to build a biophysical model of gene regulation to uncover how previously uncharacterized promoters are regulated. In particular, high-resolution studies like those described here yield quantitative predictions about promoter organization and protein-DNA interactions (*Kinney et al., 2010*). This allows us to employ the tools of statistical physics to describe the input-output properties of each of these promoters which can be explored much further with in-depth experimental dissection like those done by *Razo-Mejia et al., 2018* and *Chure et al., 2019* and summarized in *Phillips et al., 2019*. In this sense, the Sort-Seq approach can provide a quantitative framework to not only discover and quantitatively dissect regulatory interactions at the promoter level, but also provides an interpretable scheme to design genetic circuits with a desired expression output (*Barnes et al., 2019*).

Earlier work from *Belliveau et al., 2018* illustrated how Sort-Seq, used in conjunction with mass spectrometry, can be used to identify which transcription factors bind to a given binding site, thus enabling the mechanistic dissection of promoters which previously had no regulatory annotation. However, a crucial drawback of the approach of *Belliveau et al., 2018* is that while it is high-throughput at the level of a single gene and the number of promoter variants it accesses, it was unable to readily tackle multiple genes at once. Even in one of biology's best understood organisms, the bacterium *Escherichia coli*, for more than 65% of its genes, we remain completely ignorant of how those genes are regulated (*Belliveau et al., 2018*; *Santos-Zavaleta et al., 2019*). If we hope to some day have a complete base pair resolution mapping of how genetic sequences relate to biological function, we must first be able to do so for the promoters of this 'simple' organism.

What has been missing in uncovering the regulatory genome in organisms of all kinds is a large-scale method for inferring genomic logic and regulation. Here, we replace the low-throughput, fluorescence-based Sort-Seq approach with a scalable, RNA-Seq based approach that makes it possible to attack many promoters at once. Accordingly, we refer to the entirety of our approach (MPRA, information footprints and energy matrices, and transcription factor identification) as Reg-Seq, which we employ here on over one hundred promoters. The concept of MPRA methods is to perturb promoter regions by mutating their sequences, and then to use next-generation sequencing (NGS) methods to read out how those mutations impact the expression level of each promoter (*Patwardhan et al., 2009*; *Kinney et al., 2010*; *Sharon et al., 2012*; *Patwardhan et al., 2012*; *Melnikov et al., 2012*; *Kwasnieski et al., 2012*; *Fulco et al., 2019*; *Kinney and McCandlish, 2019*). We generate a broad diversity of promoter sequences for each promoter of interest and use mutual information as a metric to measure the information flow from that distribution of sequences to gene expression. Thus, Reg-Seq is able to collect causal information about candidate regulatory sequences that is then complemented by techniques such as mass spectrometry, which allows us to find which transcription factors mediate the action of those newly discovered candidate regulatory

sequences. Hence, Reg-Seq solves the causal problem of linking DNA sequence to regulatory logic and information flow.

To demonstrate our ability to perform Reg-Seq at scale, we report here our results for 113 *E. coli* genes, whose regulatory architectures (i.e. gene-by-gene distributions of transcription-factor-binding sites and identities of the transcription factors that bind those sites) were determined in parallel for multiple different growth conditions. Although we make substantial progress in mapping the regulatory information for a swath of *E. coli* genes in this study (the 'regulome'), the field still remains limited in its understanding of which specific growth conditions, small molecules and metabolites (the allosterome) are responsible for altering the milieu of transcription factor activities (*Lindsley and Rutter, 2006*; *Piazza et al., 2018*; *Huang et al., 2018*). We hope to address this shortcoming in future studies by appealing to recent work on solving the 'allosterome problem' (*Piazza et al., 2018*). By taking the Sort-Seq approach from a gene-by-gene method to a larger scale, more multiplexed approach, we can begin to piece together not just how individual promoters are regulated, but also the nature of gene-gene interactions by revealing how certain transcription factors serve to regulate multiple genes at once. This approach has the benefits of a high-throughput assay without sacrificing any of the resolution afforded by the previous gene-by-gene approach, allowing us to uncover the gene regulation of over 100 operons, with base-pair resolution, in one set of experiments.

The organization of the remainder of the paper is as follows. In the Results section, we benchmark Reg-Seq against our own earlier Sort-Seq experiments to show that the use of RNA-Seq as a readout of the expression of mutated promoters is equally reliable as the fluorescence-based approach. Additionally, we provide a global view of the discoveries that were made in our exploration of more than 100 promoters in *E. coli* using Reg-Seq. These results are described in summary form in the paper itself, with a full online version of the results (www.rpgroup.caltech.edu/RegSeq/interactive) showing how different growth conditions elicit different regulatory responses. This section also follows the overarching view of our results by examining several biological stories that emerge from our data and serve as case studies in what has been revealed in our efforts to uncover the regulatory genome. The Discussion section summarizes the method and the current round of discoveries it has afforded with an eye to future applications to further elucidate the *E. coli* genome and open up the quantitative dissection of other non-model organisms. Lastly, in the Materials and methods section and Appendices, we describe our methodology and the false positive and false negative rates of the method.

## Results

### Selection of genes and methodology

As shown in *Figure 1*, we have explored more than 100 genes from across the *E. coli* genome. Our choices were based on a number of factors (see Appendix 1 Section 'Choosing target genes' for more details); namely, we wanted a subset of genes that served as a 'gold standard' for which the hard work of generations of molecular biologists have yielded deep insights into their regulation. Our set of gold standard genes is *lacZYA*, *znuCB*, *znuA*, *ompR*, *araC*, *marR*, *relBE*, *dgoR*, *dicC*, *ftsK*, *xylA*, *xylF*, *rspA*, *dicA*, and *araAB*. By using Reg-Seq on these genes, we were able to demonstrate that this method recovers not only what was already known about binding sites of transcription factors for well-characterized promoters (*Appendix 2—figures 2* and *3*), but also whether there are any important differences between the results of the methods presented here and the previous generation of experiments based on fluorescence and cell-sorting as a readout of gene expression (*Kinney et al., 2010*; *Belliveau et al., 2018*). These promoters of known regulatory architecture are complemented by an array of previously uncharacterized genes that we selected in part using data from a recent proteomic study, in which mass spectrometry was used to measure the copy number of different proteins in 22 distinct growth conditions (*Schmidt et al., 2016*). We selected genes that exhibited a wide variation in their copy number over the different growth conditions considered, reasoning that differential expression across growth conditions implies that those genes are under regulatory control.

As noted in the introduction, the original formulation of Reg-Seq, termed Sort-Seq, was based on the use of fluorescence activated cell sorting, one gene at a time, as a way to uncover putative

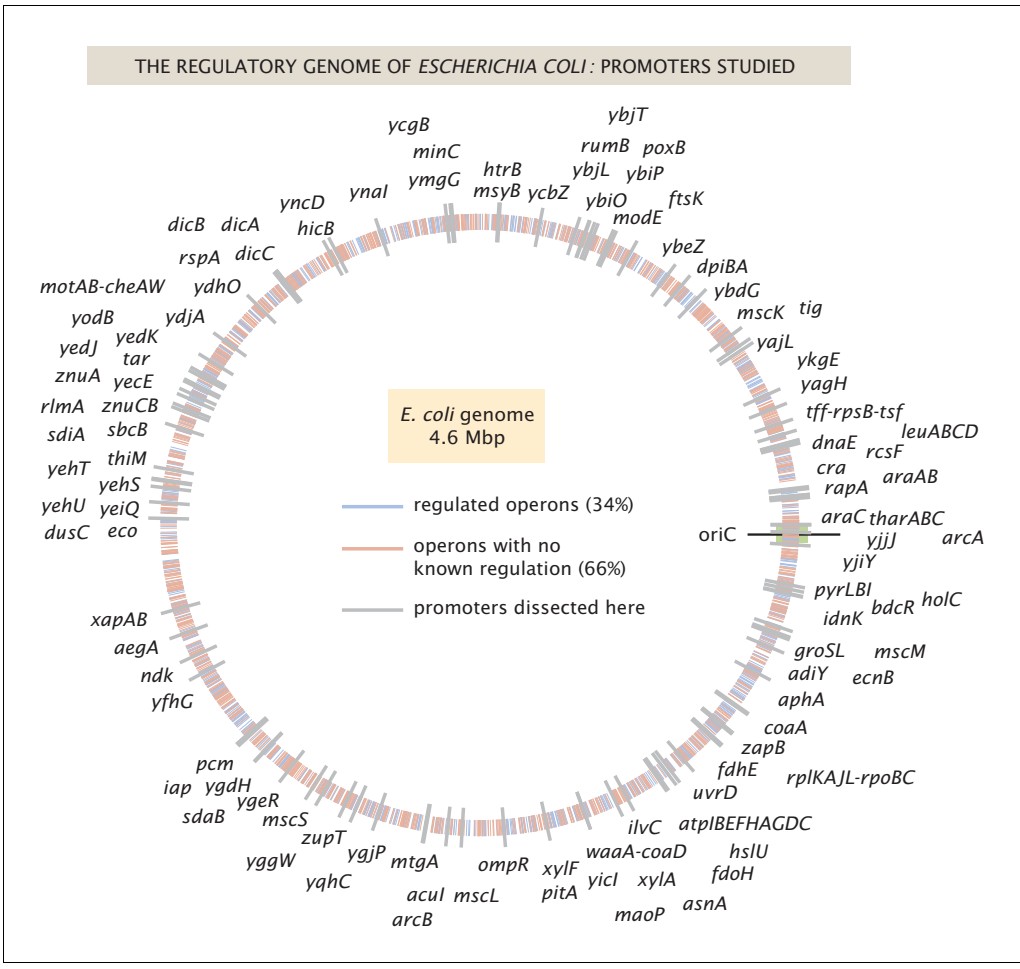

**Figure 1.** The *E. coli* regulatory genome. Illustration of the current ignorance with respect to how genes are regulated in *E. coli*. Genes with previously annotated regulation (as reported on RegulonDB [*Gama-Castro et al., 2016*]) are denoted with blue ticks and genes with no previously annotated regulation denoted with red ticks. The 113 genes explored in this study are labeled in gray, and their precise genomic locations can be found in *Figure 1—source data 1*.

The online version of this article includes the following source data for figure 1:

**Source data 1.** Locations of TSS for all promoters in *Figure 1*.

binding sites for previously uncharacterized promoters (*Belliveau et al., 2018*). As a result, as shown in *Figure 2*, we have formulated a second generation version that permits a high-throughput interrogation of the genome. A comparison between the Sort-Seq and Reg-Seq approaches on the same set of genes is shown in *Figure 3*. In the Reg-Seq approach, for each promoter interrogated, we generate a library of mutated variants and design each variant to express an mRNA with a unique sequence barcode. By counting the frequency of each expressed barcode using RNA-Seq, we can assess the differential expression from our promoter of interest based on the base-pair by base-pair sequence of its promoter. Using the mutual information between mRNA counts and sequences, we develop an information footprint that reveals the importance of different bases in the promoter region to the overall level of expression. We locate potential transcription-factor-binding regions by looking for clusters of base pairs that have a significant effect on gene expression. Further details on how potential binding sites are identified are found in the Methods Section 'Automated putative binding site algorithm' and 'Manual selection of binding sites', while determination of the false positive and false negative rates of the method can be found in Appendix 2 Section 'False positive and false negative rates'. Blue regions of the histogram shown in the information footprints of *Figure 2*

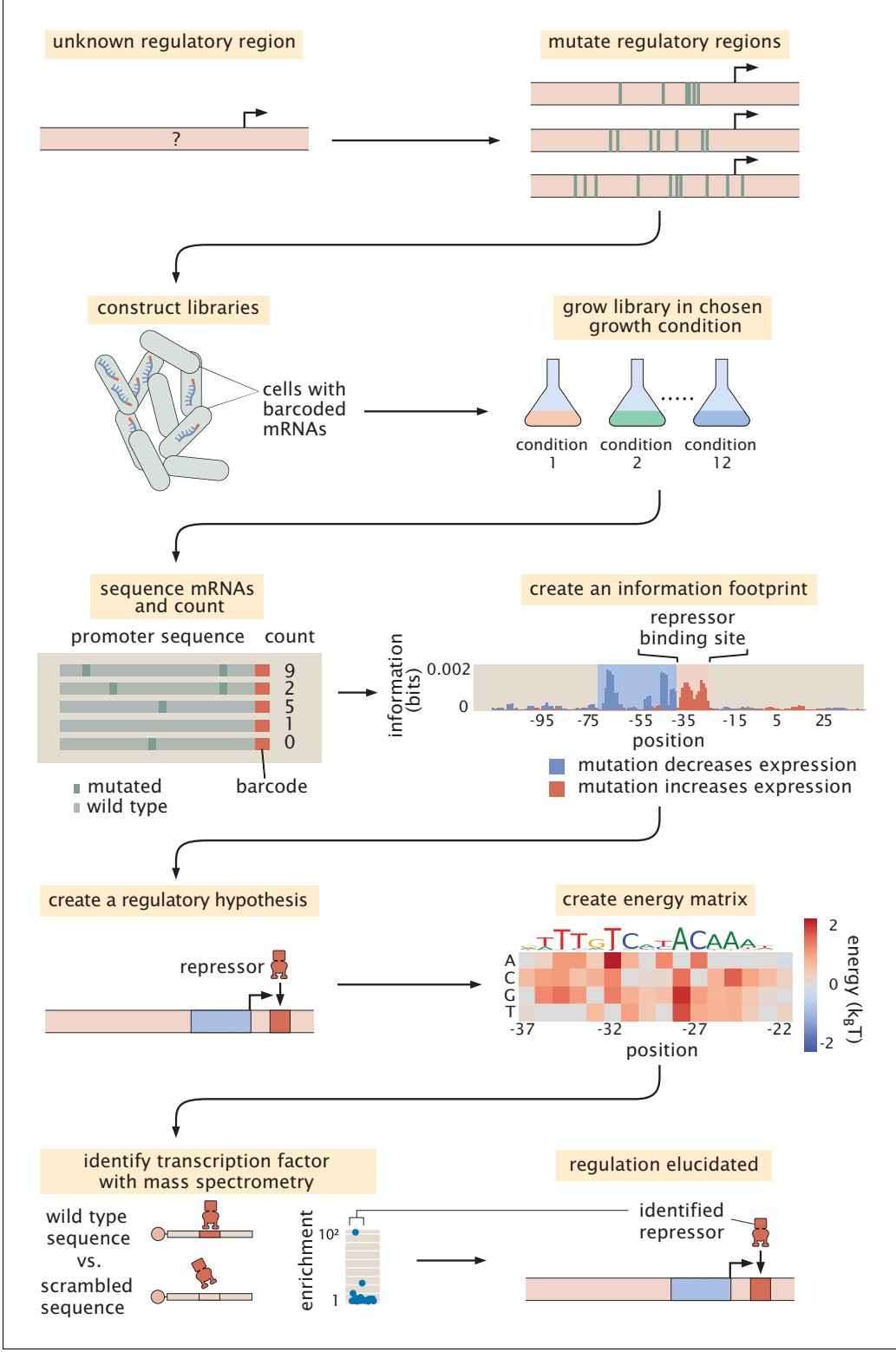

**Figure 2.** Schematic of the Reg-Seq procedure as used to recover a repressor-binding site. The process is as follows: After constructing a promoter library driving expression of a randomized barcode (an average of five barcodes for each promoter), RNA-Seq is conducted to determine the frequency of these mRNA barcodes across different growth conditions (list included in Appendix 1 Section 'Growth conditions'). By computing the mutual information between DNA sequence and mRNA barcode counts for each base pair in the promoter region, an 'information footprint' is constructed that yields a regulatory hypothesis for the putative binding sites (with the

*Figure 2 continued on next page*

*Figure 2 continued*

RNAP-binding region highlighted in blue and the repressor-binding site highlighted in red). Energy matrices, which describe the effect that any given mutation has on DNA-binding energy, as well as sequence logos, are inferred for the putative transcription-factor-binding sites. Next, we identify which transcription factor preferentially binds to the putative binding site via DNA-affinity chromatography followed by mass spectrometry. This procedure culminates in a coarse-grained, cartoon-level view of our regulatory hypothesis for how a given promoter is regulated.

The online version of this article includes the following source data for figure 2:

**Source data 1.** Information footprint data displayed in *Figure 2*.

correspond to hypothesized activating sequences and red regions of the histogram correspond to hypothesized repressing sequences.

With the information footprint in hand, we can then determine energy matrices and sequence logos (described in the next section). Given putative binding sites, we use synthesized oligonucleotides that serve as fishing hooks to isolate the transcription factors that bind to those putative binding sites using DNA-affinity chromatography and mass spectrometry (*Mittler et al., 2009*). Given all of this information, we can then formulate a schematized view of the newly discovered regulatory architecture of the previously uncharacterized promoter. For the case schematized in *Figure 2*, the experimental pipeline yields a complete picture of a simple repression architecture (i.e. a gene regulated by a single binding site for a repressor).

## Visual tools for data presentation

Throughout our investigation of the more than 100 genes explored in this study, we repeatedly relied on several key approaches to help make sense of the immense amount of data generated in these experiments. As these different approaches to viewing the results will appear repeatedly throughout the paper, here we familiarize the reader with five graphical representations referred to respectively as information footprints, energy matrices, sequence logos, mass spectrometry enrichment plots and regulatory cartoons, which taken together provide a quantitative description of previously uncharacterized promoters.

### Information footprints

From our mutagenized libraries of promoter regions, we can build up a base-pair by base-pair graphical understanding of how the promoter sequence relates to level of gene expression in the form of the information footprint shown in *Figure 2*. In this plot, the bar above each base pair position represents how large of an effect mutations at this location have on the level of gene expression. Specifically, the quantity plotted is the mutual information $I_b$ at base pair $b$ between mutation of a base pair at that position and the level of expression. In mathematical terms, the mutual information measures how much the joint probability $p(m, \mu)$ differs from the product of the probabilities $p_{mut}(m)p_{expr}(\mu)$ which would be produced if mutation and gene expression level were independent. Formally, the mutual information between having a mutation at position $b$ and level of expression is given by

$$I_b = \sum_{m=0}^{1} \sum_{\mu=0}^{1} p(m, \mu) \log_2\left(\frac{p(m, \mu)}{p_{mut}(m)p_{expr}(\mu)}\right). \tag{1}$$

Note that both $m$ and $\mu$ are binary variables that characterize the mutational state of the base of interest and the level of expression, respectively. Specifically, $m$ can take the values

$$m = \begin{cases} 0, & \text{if } b \text{ is a mutated base} \\ 1, & \text{if } b \text{ is a wild-type base.} \end{cases} \tag{2}$$

and $\mu$ can take on values

$$\mu = \begin{cases} 0, & \text{for sequencing reads from the DNA library} \\ 1, & \text{for sequencing reads originating from mRNA,} \end{cases} \tag{3}$$

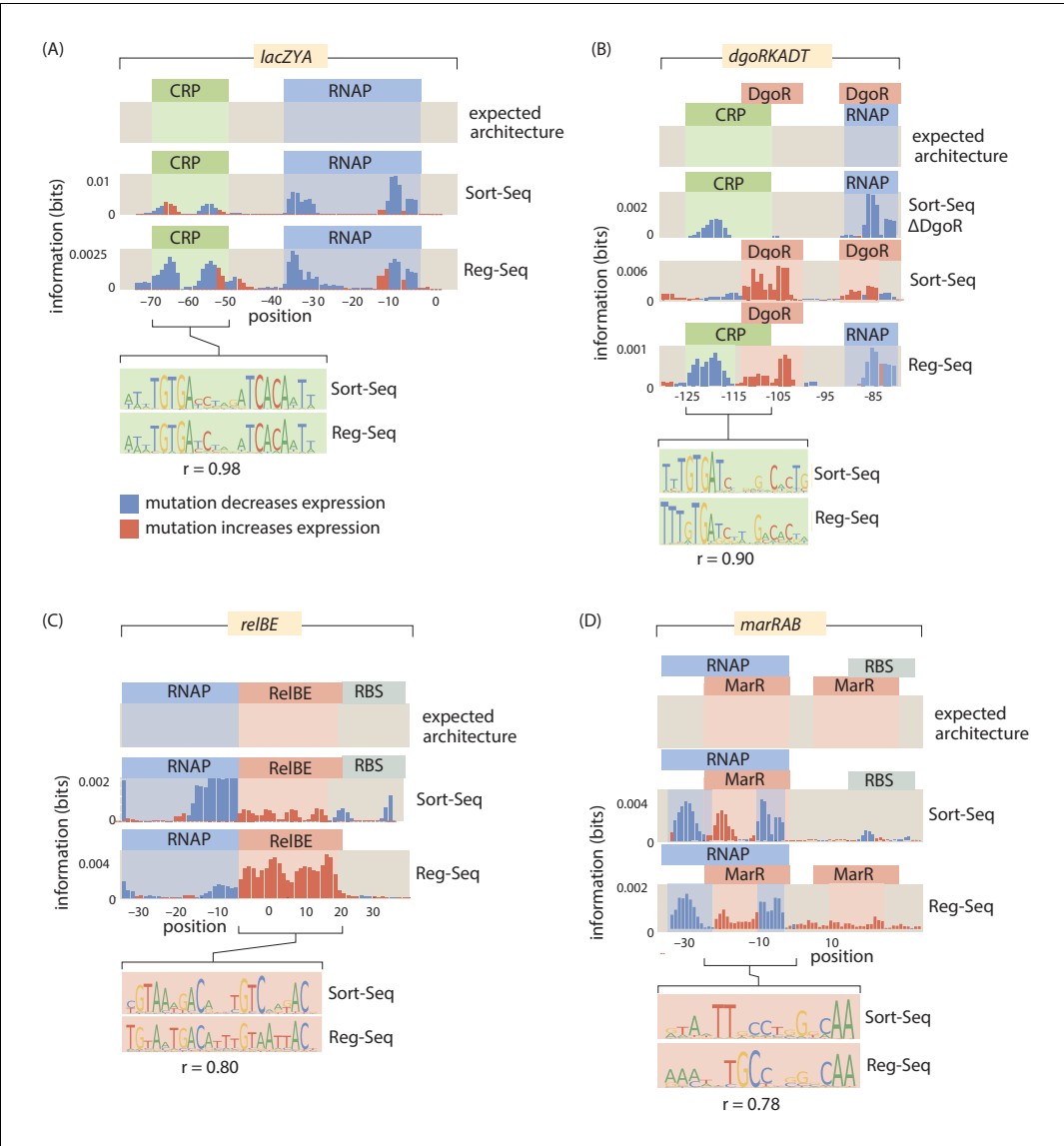

**Figure 3.** A summary of four direct comparisons of measurements from Sort-Seq and Reg-Seq. We show the identified regulatory regions as well as quantitative comparisons between inferred position weight matrices. (**A**) CRP binds upstream of RNAP in the *lacZYA* promoter. Despite the different measurement techniques for the two inferred position weight matrices, the CRP-binding sites have a Pearson correlation coefficient of $r = 0.98$. (**B**) The *dgoRKADT* promoter is activated by CRP in the presence of galactonate and is repressed by DgoR. For Sort-Seq and Reg-Seq, type II activator-binding sites can be identified based on the signals in the information footprint in the area indicated in green. Additionally, the quantitative agreement between the CRP position weight matrices are strong, with $r = 0.9$. (**C**) The *relBE* promoter is repressed by RelBE as can be identified algorithmically in both Sort-Seq and Reg-Seq. The inferred logos for the two measurement methods have $r = 0.8$. (**D**) The *marRAB* promoter is repressed by MarR. The inferred energy matrices (data not shown) and sequence logos shown have $r = 0.78$. The right most MarR site overlaps with a ribosome-binding site. The overlap has a stronger obscuring effect on the sequence specificity of the Sort-Seq measurement, which measures protein levels directly, than it does on the output of the Reg-Seq measurement. Numeric values for the displayed data can be found in *Figure 3—source data 1*.

The online version of this article includes the following source data for figure 3:

**Source data 1.** Data for information footprints and PWMs in *Figure 3*.

where both $m$ and $\mu$ are index variables that tell us whether the base has been mutated and if so, how likely that the read at that position will correspond to an mRNA, reflecting gene expression or a

promoter, reflecting a member of the library. The higher the ratio of mRNA to DNA reads at a given base position, the higher the expression. $p_{mut}(m)$ in *Equation 1* refers to the probability that a given sequencing read will be from a mutated base. $p_{expr}(\mu)$ is a numeric value that gives the ratio of the number of DNA or mRNA sequencing counts to the total number of sequencing counts for each barcode.

Furthermore, we color the bars based on whether mutations at this location lowered gene expression on average (in blue, indicating an activating role) or increased gene expression (in red, indicating a repressing role). In this experiment, we targeted the regulatory regions based on a guess of where a transcription start site (TSS) will be, based on experimentally confirmed sites contained in RegulonDB (*Santos-Zavaleta et al., 2019*), a 5' RACE experiment (*Mendoza-Vargas et al., 2009*), or by targeting small intergenic regions so as to capture all likely regulatory regions. Further details on TSS selection can be found in the Materials and methods Section 'Library design and construction'. After completing the Reg-Seq experiment, we note that many of the presumed TSS sites are not in the locations assumed, the promoters have multiple active RNA polymerase (RNAP) sites and TSS, or the primary TSS shifts with growth condition. To simplify the data presentation, the '0' base pair in all information footprints is set to the originally assumed base pair for the primary TSS, rather than one of the TSS that was found in the experiment.

## Energy matrices

Focusing on an individual putative transcription-factor-binding site as revealed in the information footprint, we are interested in a more fine-grained, quantitative understanding of how the underlying protein-DNA interaction is determined. An energy matrix displays this information using a heat map format, where each column is a position in the putative binding site and each row displays the effect on binding that results from mutating to that given nucleotide (given as a change in the DNA-transcription factor interaction energy upon mutation) (*Berg and von Hippel, 1987*; *Stormo and Fields, 1998*; *Kinney et al., 2010*). These energy matrices are scaled such that the wild type sequence is colored in white, mutations that improve binding are shown in blue, and mutations that weaken binding are shown in red. These energy matrices encode a full quantitative picture for how we expect sequence to relate to binding for a given transcription factor, such that we can provide a prediction for the binding energy of every possible binding site sequence as

$$\text{binding energy} = \sum_{i=1}^{N} \varepsilon_i, \tag{4}$$

where the energy matrix is predicated on an assumption of a linear binding model in which each base within the binding site region contributes a specific value ($\varepsilon_i$ for the $i^{th}$ base in the sequence) to the total binding energy. Energy matrices are either given in A.U. (arbitrary units) or, for several cases where the gene has a simple repression or activation architecture with a single RNA polymerase (RNAP) site, are assigned $k_B T$ energy units following the procedure in *Kinney et al., 2010* and validated on repression by *lac* repressor in *Barnes et al., 2019*. The details of how and when absolute units are determined can be found in Appendix 3 Section 'Inference of scaling factors for energy matrices'.

## Sequence logos

From an energy matrix, we can also represent a preferred transcription-factor-binding site with the use of the letters corresponding to the four possible nucleotides, as is often done with position weight matrices (*Schneider and Stephens, 1990*). In these sequence logos, the size of the letters corresponds to how strong the preference is for that given nucleotide at that given position, which can be directly computed from the energy matrix. This method of visualizing the information contained within the energy matrix is more easily digested and allows for quick comparison among various binding sites.

## Mass spectrometry enrichment plots

As the final piece of our experimental pipeline, we wish to determine the identity of the transcription factor we suspect is binding to our putative binding site that is represented in the energy matrix and

sequence logo. While the details of the DNA-affinity chromatography and mass spectrometry can be found in the Materials and methods, the results of these experiments are displayed in enrichment plots such as is shown in the bottom panel of *Figure 2*. In these plots, the relative abundance of each protein bound to our site of interest is quantified relative to a scrambled control sequence. The putative transcription factor is the one we find to be highly enriched compared to all other DNA-binding proteins.

## Regulatory cartoons

The ultimate result of all these detailed base-pair-by-base-pair resolution experiments yields a cartoon model of how we think the given promoter is being regulated. A complete set of cartoons for all the architectures considered in our study is presented later in *Figure 4*. While the cartoon serves as a convenient, visual way to summarize our results, it is important to remember that these cartoons are a shorthand representation of all the data in the four quantitative measures described above and are, further, backed by quantitative predictions of how we expect the system to behave when tested experimentally. Throughout this paper, we use consistent iconography to illustrate the regulatory architecture of promoters with activators and their binding sites in green, repressors in red, and RNAP in blue.

## Newly discovered *E. coli* regulatory architectures

### Elucidating individual promoters

With the tools outlined above, we are positioned to explore individual promoters, specifically those belonging to the part of the *E. coli* genome for which the function of the genes is unknown. Previously christened as the 'y-ome', *Ghatak et al., 2019* surprisingly found that roughly 35% of the genes in *E. coli* lack experimental evidence of function. The situation is likely worse for other organisms. For many of the genes in the y-ome, we remain similarly ignorant of how those genes are regulated. *Figures 4* and *5* provide several examples of genes which until now had unknown regulation. As shown in *Figure 5*, our study has found the first examples that we are aware of in the entire *E. coli* genome of a binding site for YciT. These examples are intended to show the outcome of the methods developed here and to serve as an invitation to browse the online resource (https://www.rpgroup.caltech.edu/RegSeq/interactive) where our full dataset is presented.

The ability to find binding sites for both widely acting regulators and transcription factors which may have only a few sites in the whole genome allows us to get an in-depth and quantitative view of any given promoter. As indicated in *Figure 5(A) and (B)*, we were able to perform the relevant search and capture for the transcription factors that bind our putative binding sites. In both of these cases, we now hypothesize that these newly discovered binding site-transcription factor pairs exert their control through repression. The ability to extract the quantitative features of regulatory control through energy matrices means that we can take a nearly unstudied gene such as *ykgE*, which is regulated by an understudied transcription factor YieP, and quickly get to the point at which we can do quantitative modeling in the style that we and many others have performed on the *lac* operon (*Vilar and Leibler, 2003*; *Vilar et al., 2003*; *Bintu et al., 2005*; *Kinney et al., 2010*; *Garcia and Phillips, 2011*; *Vilar and Saiz, 2013*; *Barnes et al., 2019*; *Phillips et al., 2019*).

### A panoply of promoter results

*Figure 6* (and *Tables 1* and *2*) provides a summary of the discoveries made in the work done here using our next-generation Reg-Seq approach. The outcome of our study is a set of hypothesized regulatory architectures as characterized by a suite of binding sites for RNAP, repressors, and activators, as well as the extremely potent binding energy matrices. We do not assume, a priori, that a particular collection of such binding sites is AND, OR, or any other logic (*Galstyan et al., 2019*). *Figure 6(A)* provides a shorthand notation that conveniently characterizes the different kinds of regulatory architectures found in bacteria. In this ($n_a$, $n_r$) notation, $n_a$ and $n_r$ correspond to the number of recovered activator- and repressor-binding sites, respectively. In previous work (*Rydenfelt et al., 2014*), we have explored the entirety of what is known about the regulatory genome of *E. coli*, revealing that the most common motif is the (0, 0) constitutive architecture, although we hypothesized that this is not a statement about the facts of the *E. coli* genome, but rather a reflection of our collective regulatory ignorance in the sense that we suspect that with further investigation, many of

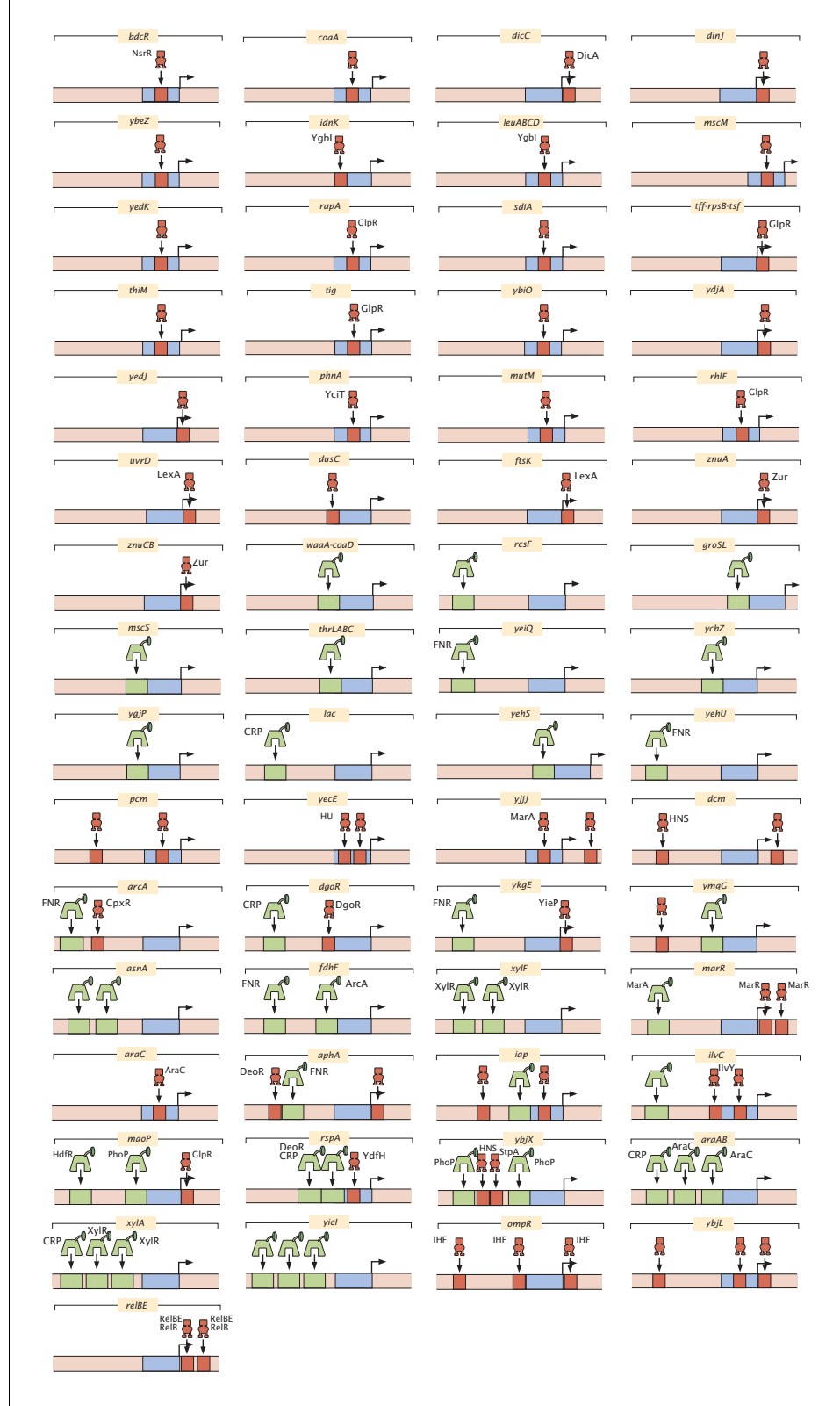

**Figure 4.** All regulatory architectures uncovered in this study. For each regulated promoter, activators and their binding sites are labeled in green, repressors and their binding sires are labeled in red, and RNAP-binding sites are labeled in blue. All cartoons are displayed with the transcription direction to the right. Only one RNAP site is depicted per promoter. The transcription-factor-binding sites displayed have either been identified by the method

*Figure 4 continued on next page*

*Figure 4 continued*
described in the Section 'Automated putative binding site algorithm' or have additional evidence for their presence as described in *Table 2*. Binding sites found for these promoters in the EcoCyc or RegulonDB databases are only depicted in these cartoons if the sites are within the 160 bp mutagenized region studied, and are detected by Reg-Seq.

these apparent constitutive architectures will be found to be regulated under the right environmental conditions. The two most common regulatory architectures that emerged from our previous database survey are the (0, 1) and the (1, 0) architectures, the simple repression motif and the simple activation motif, respectively. It is interesting to consider that the (0, 1) architecture is in fact the repressor-operon model originally introduced in the early 1960s by *Jacob and Monod, 1961* as the concept of gene regulation emerged. Now we see retrospectively the far reaching importance of that architecture across the regulatory genome.

For the 113 genes we considered, *Figure 6(B)* summarizes the number of simple repression (0, 1) architectures discovered, the number of simple activation (1, 0) architectures discovered and so on. A comparison of the frequency of the different architectures found in our study to the frequencies of all the known architectures in the RegulonDB database is provided in *Appendix 4—figure 2*. *Tables 1* and *2* provide a more detailed view of our results. As seen in *Table 1*, of the 113 genes we considered, 34 of them revealed no signature of any transcription-factor-binding sites and they are labeled as (0, 0). The simple repression architecture (0, 1) was found 26 times, the simple activation architecture (1, 0) was found 11 times, and more complex architectures featuring multiple binding sites (e.g. (1, 1), (0, 2), (2, 0), etc.) were revealed as well. Further, for 18 of the genes that we label 'inactive', Reg-Seq did not reveal a potential RNAP-binding site. The lack of observable RNAP site could be because the proper growth condition to get high levels of expression was not used, or because the mutation window chosen for the gene does not capture a highly transcribing TSS.

The tables also include our set of 15 'gold standard' genes for which previous work has resulted in a knowledge (sometimes only partial) of their regulatory architectures. We find that our method recovers the regulatory elements of these gold standard cases fully in 11 out of 15 cases, and the majority of regulatory elements in two of the remaining cases. Overall, the performance of Reg-Seq in these gold-standard cases (for more details see *Appendix 2—figures 2* and *3*) builds confidence in the approach. Further, the failure modes inform us of the blind spots of Reg-Seq. For example, we find it challenging to observe weaker binding sites when multiple strong binding sites are also present such as in the *marRAB* operon. The *araC* case study shows that Reg-Seq does not perform well when many repressor sites regulate the promoter. Additionally the method will fail when there is no active TSS in the mutation window, as occurred in the case of *dicA*. Further details on the comparison to gold standard genes can be found in Appendix 2 Section 'False positive and false negative rates'.

We observe that the most common motif to emerge from our work (with the exception of constitutive expression) is the simple repression motif. Another relevant regulatory statistic is shown in *Figure 6(C)* where we see the distribution of binding site positions. Our own experience in the use of different quantitative modeling approaches to transcriptional regulation reveal that, for now, we remain largely ignorant of how to account for transcription-factor-binding site positions, and datasets like the one presented here will begin to provide data that can help us uncover how this parameter dictates gene expression. Indeed, with binding site positions and energy matrices in hand, we can systematically move these binding sites and explore the implications for the level of gene expression, providing a systematic tool to understand the role of binding-site position.

## Uncovering the action of global regulators

One of the revealing case studies that demonstrates the broad reach of our approach for discovering regulatory architectures is offered by the insights we have gained into two widely acting regulators, GlpR (*Figure 7*; *Schweizer et al., 1985*) and FNR (*Figure 8*; *Körner et al., 2003*; *Kargeti and Venkatesh, 2017*). In both cases, we have expanded the array of promoters that they are now known to regulate. Further, these two case studies illustrate that even for widely acting transcription factors, there is a large gap in regulatory knowledge and the approach advanced here has the power to discover new regulatory motifs. The newly discovered binding sites in *Figure 7(A)*, with additional

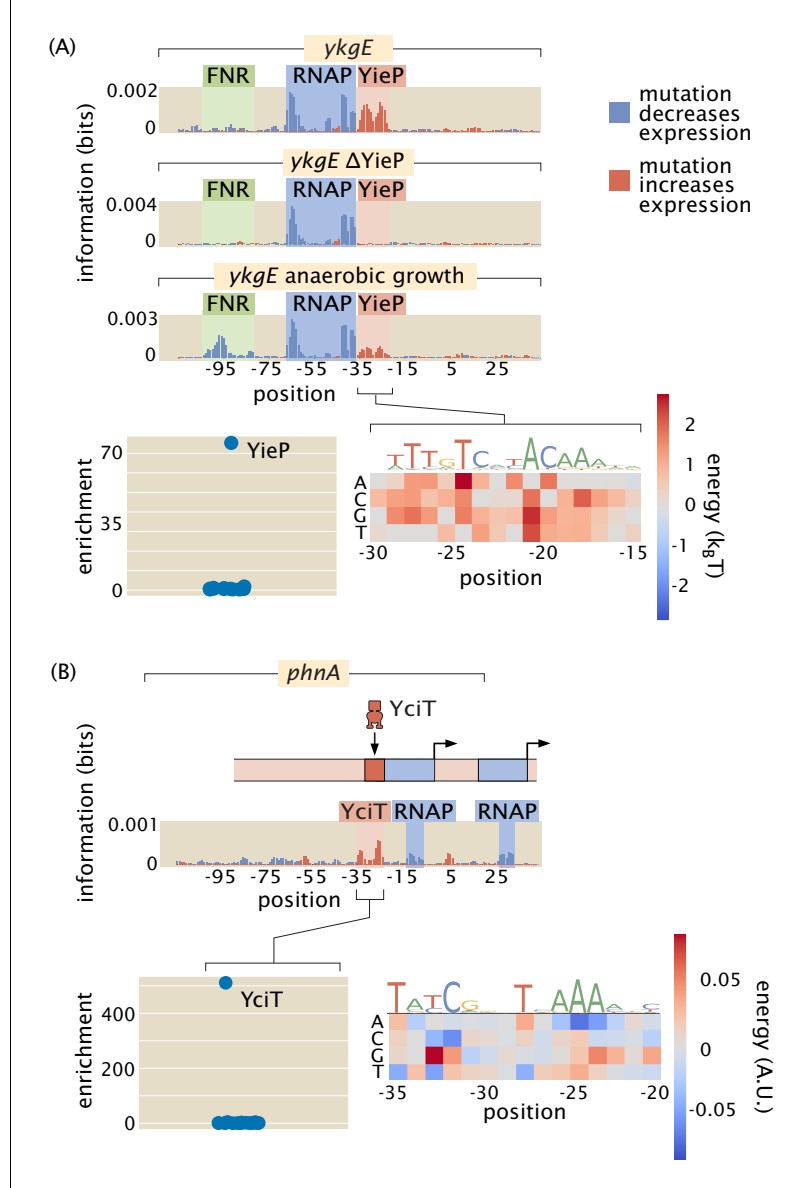

**Figure 5.** Examples of the insight gained by Reg-Seq in the context of promoters with no previously known regulatory information. Activator-binding regions are highlighted in green, repressor binding regions in red, and RNAP binding regions in blue. (**A**) From the information footprint of the *ykgE* promoter under different growth conditions, we can identify a repressor-binding site downstream of the RNAP-binding site. From the enrichment of proteins bound to the DNA sequence of the putative repressor as compared to a control sequence, we can identify YieP as the transcription factor bound to this site as it has a much higher enrichment ratio than any other protein. Lastly, the binding energy matrix for the repressor site along with corresponding sequence logo shows that the wild-type sequence is the strongest possible binder and it displays an imperfect inverted repeat symmetry. (**B**) Illustration of a comparable dissection for the *phnA* promoter. Numeric values for the displayed data can be found in *Figure 5—source data 1*.

The online version of this article includes the following source data for figure 5:

**Source data 1.** Data for information footprints, energy matrices, PWMs, and mass spectrometry in *Figure 5*.

evidence for GlpR binding in *Figure 7(B) and (C)*, more than double the number of operons known to be regulated by GlpR as reported in RegulonDB (*Santos-Zavaleta et al., 2019*). We found five newly regulated operons in our data set, even though we were not specifically targeting GlpR regulation. Although the number of example promoters across the genome that we considered is too

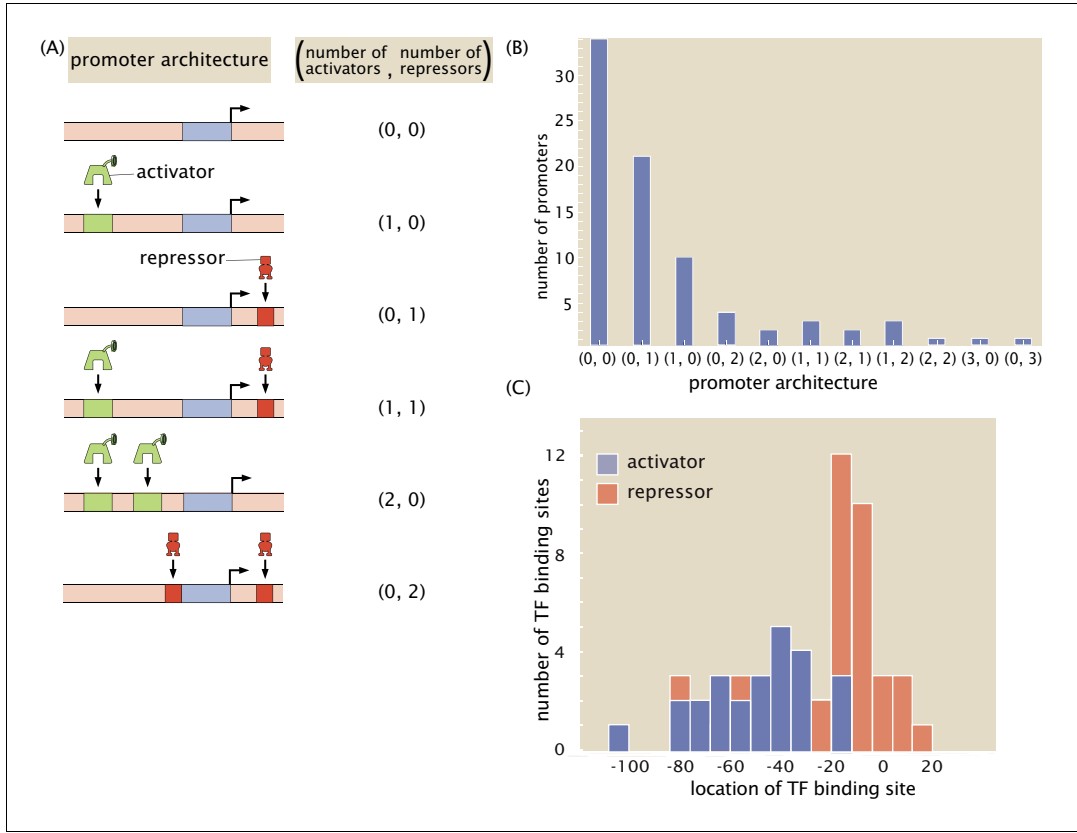

**Figure 6.** A summary of regulatory architectures discovered in this study. (A) The cartoons display a representative example of each type of architecture, along with the corresponding shorthand notation. (B) Counts of the different regulatory architectures discovered in this study. We exclude the 'gold-standard' promoters (listed in *Appendix 2—table 1*) unless new transcription factors are also discovered in the promoter. If, for example, one repressor was newly discovered and two activators were previously known, then the architecture is still counted as a (2,1) architecture. (C) Distribution of positions of binding sites discovered in this study for activators and repressors. Only newly discovered binding sites are included in this figure. The position of the transcription-factor-binding sites are calculated relative to the estimated TSS location, which is based on the location of the associated RNAP site. Numeric values for the binding locations can be found in *Figure 6—source data 1*. The online version of this article includes the following source data for figure 6:

**Source data 1.** Data for binding site locations in *Figure 6*.

small to make good estimates, finding five regulated operons out of approximately 100 examined operons supports the claim that GlpR widely regulates and many more of its sites would be found in a full search of the genome. The regulatory roles revealed in *Figure 7(A)* also reinforce the evidence that GlpR is a repressor.

For the GlpR-regulated operons newly discovered here, we found that this repressor binds strongly in the presence of glucose while all other growth conditions result in greatly diminished, but not entirely abolished, binding (*Figure 7(A)*). As there is no previously known direct molecular interaction between GlpR and glucose and the repression is reduced but not eliminated, the derepression in the absence of glucose is likely an indirect effect. As a potential mechanism of the indirect effect, *gpsA* is known to be activated by CRP (*Seoh and Tai, 1999*), and GpsA is involved in the synthesis of glycerol-3-phosphate (G3P), a known binding partner of GlpR which disables its repressive activity (*Larson et al., 1987*). Thus, in the presence of glucose, GpsA and consequently G3P will be found at low concentrations, ultimately allowing GlpR to fulfill its role as a repressor.

Prior to this study, there were four operons known to be regulated by GlpR, each with between 4 and 8 GlpR-binding sites (*Larson et al., 1992*; *Zhao et al., 1994*; *Yang and Larson, 1996*; *Ye and Larson, 1988*; *Weissenborn et al., 1992*), where the absence of glucose and the partial induction of GlpR was not enough to prompt a notable change in gene expression (*Lin, 1976*). These previously

**Table 1.** All promoters examined in this study, categorized according to type of regulatory architecture.
Those promoters which have no recognizable RNAP site are labeled as inactive rather than constitutively expressed (0, 0).

| Architecture | Total number of promoters | Number of promoters with at least one newly discovered binding site |
|---|---|---|
| All Architectures | 113 | 48 |
| (0,0) | 34 | 0 |
| (0,1) | 26 | 21 |
| (1,0) | 11 | 10 |
| (1,1) | 4 | 3 |
| (0,2) | 4 | 4 |
| (2,0) | 3 | 2 |
| (1,2) | 4 | 3 |
| (2,1) | 2 | 2 |
| (2,2) | 1 | 1 |
| (3,0) | 3 | 1 |
| (0,3) | 2 | 1 |
| (0,4) | 1 | 0 |
| inactive | 18 | 0 |

explored operons seemingly are regulated as part of an AND gate. *glpTQ*, *glpRABC*, *glpD*, and *glpFKX* have high gene expression when grown in growth media that does *not* contain glucose but does contain contain G3P (or glycerol, which leads to high concentrations of G3P). All other combinations of growth media, such as M9 glucose with G3P, or growth in LB without G3P, lead to low gene expression (*Lin, 1976*). In contrast, we have discovered operons whose regulation appears to be mediated by a single GlpR site per operon. With only a single site, GlpR functions as an indirect glucose sensor, as only the absence of glucose is needed to relieve repression by GlpR.

The second widely acting regulator our study revealed, FNR, has 151 binding sites already reported in RegulonDB and is well studied compared to most transcription factors (*Santos-Zavaleta et al., 2019*). However, the newly discovered FNR sites displayed in *Figure 8(A)*, with sequence logos of the respective sites displayed in *Figure 8(B)*, demonstrate that even for well-understood transcription factors there is much still to be uncovered. Our information footprints are in agreement with previous studies suggesting that FNR acts as an activator. In the presence of $O_2$, dimeric FNR is converted to a monomeric form and its ability to bind DNA is greatly reduced (*Myers et al., 2013*). Only in low oxygen conditions did we observe a binding signature from FNR, and we show a representative example of the information footprint from one of 11 aerobic growth conditions in *Figure 8(A)*.

We observe quantitatively how FNR affects the expression of *fdhE* both directly through transcription factor binding (*Figure 9B and C*) and indirectly through increased expression of ArcA (*Figure 9A, B, C and D*). Also, fully understanding even a single operon often requires investigating several regulatory regions as we have in the case of *fdoGHI-fdhE* by investigating the main promoter for the operon as well as the promoter upstream of *fdhE*. 36% of all multi-gene operons have at least one TSS which transcribes only a subset of the genes in the operon (*Conway et al., 2014*). Regulation within an operon is even more poorly studied than regulation in general. The main promoter for *fdoGHI-fdhE* has a repressor-binding site, which demonstrates that there is regulatory control of the entire operon. However, we also see in *Figure 9(B)* that there is control at the promoter level, as *fdhE* is regulated by both ArcA and FNR and will therefore be upregulated in anaerobic conditions (*Compan and Touati, 1994*). The main TSS transcribes all four genes in the operon, while the secondary site shown in *Figure 9(B)* only transcribes *fdhE*, and therefore anaerobic conditions

**Table 2.** All genes investigated in this study categorized according to their regulatory architecture, given as (number of activators, number of repressors).

The regulatory architectures as listed reflect only the binding sites that would be able to be recovered within our 160 bp constructs, but include both newly discovered and previously known binding sites. In those cases where binding sites that appear in RegulonDB or Ecocyc are omitted from this tally, the Section 'Explanation of included binding sites' in Appendix 4 has the reasoning, for each relevant gene, why the binding sites are not shown. The table also lists the number of newly discovered binding sites, previously known binding sites, and number of identified transcription factors. The evidence used for the transcription factor identification is given in the final column. 'Bioinformatic' evidence implies that discovered position weight matrices were compared to known transcription factor position weight matrices. The literature sites column contains only those sites that are both expected to be and are, in actuality, observed in the Reg-Seq data.

| Architecture | Promoter | Newly discovered binding sites | Literature binding sites | Identified binding sites | Evidence |
|---|---|---|---|---|---|
| (0, 0) | acuI | 0 | 0 | 0 | |
| | aegA | 0 | 0 | 0 | |
| | arcB | 0 | 0 | 0 | |
| | cra | 0 | 0 | 0 | |
| | dnaE | 0 | 0 | 0 | |
| | ecnB | 0 | 0 | 0 | |
| | fdoH | 0 | 0 | 0 | |
| | holC | 0 | 0 | 0 | |
| | hslU | 0 | 0 | 0 | |
| | htrB | 0 | 0 | 0 | |
| | minC | 0 | 0 | 0 | |
| | modE | 0 | 0 | 0 | |
| | ycgB | 0 | 0 | 0 | |
| | mscL | 0 | 0 | 0 | |
| | pitA | 0 | 0 | 0 | |
| | poxB | 0 | 0 | 0 | |
| | rlmA | 0 | 0 | 0 | |
| | rumB | 0 | 0 | 0 | |
| | sbcB | 0 | 0 | 0 | |
| | sdaB | 0 | 0 | 0 | |
| | tar | 0 | 0 | 0 | |
| | ybdG | 0 | 0 | 0 | |
| | ybiP | 0 | 0 | 0 | |
| | ybjT | 0 | 0 | 0 | |
| | yehT | 0 | 0 | 0 | |
| | yfhG | 0 | 0 | 0 | |
| | ygdH | 0 | 0 | 0 | |
| | ygeR | 0 | 0 | 0 | |
| | yggW | 0 | 0 | 0 | |
| | ynaI | 0 | 0 | 0 | |
| | yqhC | 0 | 0 | 0 | |
| | zapB | 0 | 0 | 0 | |
| | zupT | 0 | 0 | 0 | |
| | amiC | 0 | 0 | 0 | |
| (0, 1) | araC | 0 | 1 | 0 | |
| | bdcR | 1 | 0 | 1 | Known binding location (NsrR) (*Partridge et al., 2009*) |

*Table 2 continued on next page*

*Table 2 continued*

| Architecture | Promoter | Newly discovered binding sites | Literature binding sites | Identified binding sites | Evidence |
|---|---|---|---|---|---|
| | coaA | 1 | 0 | 0 | |
| | dicC | 0 | 1 | 0 | |
| | dinJ | 1 | 0 | 0 | |
| | ybeZ | 1 | 0 | 0 | |
| | idnK | 1 | 0 | 1 | Mass- Spectrometry (YgbI) |
| | leuABCD | 1 | 0 | 1 | Mass- Spectrometry (YgbI) |
| | mscM | 1 | 0 | 0 | |
| | yedK | 1 | 0 | 1 | Mass- Spectrometry (TreR) |
| | rapA | 1 | 0 | 1 | Growth condition Knockout (GlpR), Bioinformatic (GlpR) |
| | sdiA | 1 | 0 | 0 | |
| | tff-rpsB-tsf | 1 | 0 | 1 | Growth condition Knockout (GlpR), Bioinformatic (GlpR), Knockout (GlpR) |
| | thiM | 1 | 0 | 0 | |
| | tig | 1 | 0 | 1 | Growth condition Knockout (GlpR), Bioinformatic (GlpR), Knockout (GlpR) |
| | ybiO | 1 | 0 | 0 | |
| | ydjA | 1 | 0 | 0 | |
| | yedJ | 1 | 0 | 0 | |
| | phnA | 1 | 0 | 1 | Mass- Spectrometry (YciT) |
| | mutM | 1 | 0 | 0 | |
| | rhlE | 1 | 0 | 1 | Growth condition Knockout (GlpR), Bioinformatic (GlpR), Mass-Spectrometry (GlpR) |
| | uvrD | 1 | 0 | 1 | Bioinformatic (LexA) |
| | dusC | 1 | 0 | 0 | |
| | ftsK | 0 | 1 | 0 | |
| | znuA | 0 | 1 | 0 | |
| | znuCB | 0 | 1 | 0 | |
| (1, 0) | waaA-coaD | 1 | 0 | 0 | |
| | rcsF | 1 | 0 | 0 | |
| | groSL | 1 | 0 | 0 | |
| | mscS | 1 | 0 | 0 | |
| | thrLABC | 1 | 0 | 0 | |
| | yeiQ | 1 | 0 | 1 | Growth condition Knockout (FNR), Bioinformatic (FNR) |
| | ycbZ | 1 | 0 | 0 | |
| | ygjP | 1 | 0 | 0 | |
| | lac | 0 | 1 | 0 | Bioinformatic (CRP) |
| | yehS | 1 | 0 | 0 | |
| | yehU | 1 | 0 | 1 | Growth condition Knockout (FNR), Bioinformatic (FNR) |
| (0, 2) | pcm | 2 | 0 | 0 | |
| | yecE | 2 | 0 | 1 | Mass- Spectrometry (HU) |
| | yjjJ | 2 | 0 | 1 | Growth condition Knockout (MarA), Bioinformatic (MarA) |
| | dcm | 2 | 0 | 1 | Mass- Spectrometry (HNS) |
| (1, 1) | arcA | 2 | 0 | 2 | Growth condition Knockout (FNR), Bioinformatic (FNR), Mass-Spectrometry (FNR, CpxR) |

*Table 2 continued on next page*

*Table 2 continued*

| Architecture | Promoter | Newly discovered binding sites | Literature binding sites | Identified binding sites | Evidence |
|---|---|---|---|---|---|
| | *dgoR* | 0 | 2 | 0 | Bioinformatic (CRP) Bioinformatic (DgoR) |
| | *ykgE* | 2 | 0 | 2 | Growth condition Knockout (FNR), Bioinformatic (FNR), Mass-Spectrometry(YieP) Knockout (YieP) |
| | *ymgG* | 2 | 0 | 0 | |
| (2, 0) | *asnA* | 2 | 0 | 0 | |
| | *fdhE* | 2 | 0 | 2 | Growth condition Knockout (FNR, ArcA), Bioinformatic (FNR, ArcA), Knockout (ArcA) |
| | *xylF* | 0 | 2 | 0 | |
| (1, 2) | *marR* | 0 | 3 | 0 | Mass- Spectrometry (MarR) |
| | *aphA* | 3 | 0 | 2 | Growth condition Knockout (FNR), Bioinformatic (FNR), Mass-Spectrometry (DeoR) |
| | *iap* | 3 | 0 | 0 | |
| | *ilvC* | 3 | 0 | 1 | Mass- Spectrometry (IlvY) (**Rhee et al., 1998**) |
| (2, 1) | *maoP* | 3 | 0 | 3 | Growth condition Knockout (GlpR), Bioinformatic (GlpR), Knockout (PhoP, HdfR, GlpR) |
| | *rspA* | 1 | 2 | 1 | Mass- Spectrometry (DeoR) |
| (2, 2) | *ybjX* | 4 | 0 | 4 | Bioinformatic (2 PhoP sites), Mass- Spectrometry (HNS, StpA) |
| (3, 0) | *araAB* | 0 | 3 | 0 | |
| | *xylA* | 0 | 3 | 0 | |
| | *yicI* | 3 | 0 | 0 | |
| (0, 3) | *ompR* | 0 | 3 | 0 | |
| | *ybjL* | 3 | 0 | 0 | |
| (0, 4) | *relBE* | 0 | 4 | 0 | Mass- Spectrometry (RelBE) |

will change the stoichiometry of the proteins produced by the operon. By investigating over a hundred promoter regions in this experiment it becomes feasible to target multiple promoters within an operon as we have done with *fdoGHI-fdhE*. We can then determine under what conditions an operon is internally regulated.

## In summary

By examining the over 100 promoters considered here, grown under 12 growth conditions, we have a total of more than 1000 information footprints and data sets. In this age of big data, methods to explore and draw insights from that data are crucial. To that end, as introduced in *Figure 10*, we have developed an online resource (see https://www.rpgroup.caltech.edu/RegSeq/interactive) that makes it possible for anyone who is interested to view our data and draw their own biological conclusions. Information footprints for any combination of gene and growth condition are displayed via drop down menus. Each identified transcription-factor-binding site is marked, and energy matrices for all transcription-factor-binding sites are displayed. In addition, for each gene, we feature a simple cartoon-level schematic that captures our now current, best understanding of the regulatory architecture and resulting mechanism.

The interactive figure in question was invaluable in identifying transcription factors, such as GlpR, whose binding properties vary depending on growth condition. As sigma factor availability also varies greatly depending on growth condition, studying the interactive figure identified many of the secondary RNAP sites present. The interactive figure provides a valuable resource both to those who are interested in the regulation of a particular gene and those who wish to look for patterns in gene regulation across multiple genes or across different growth conditions.

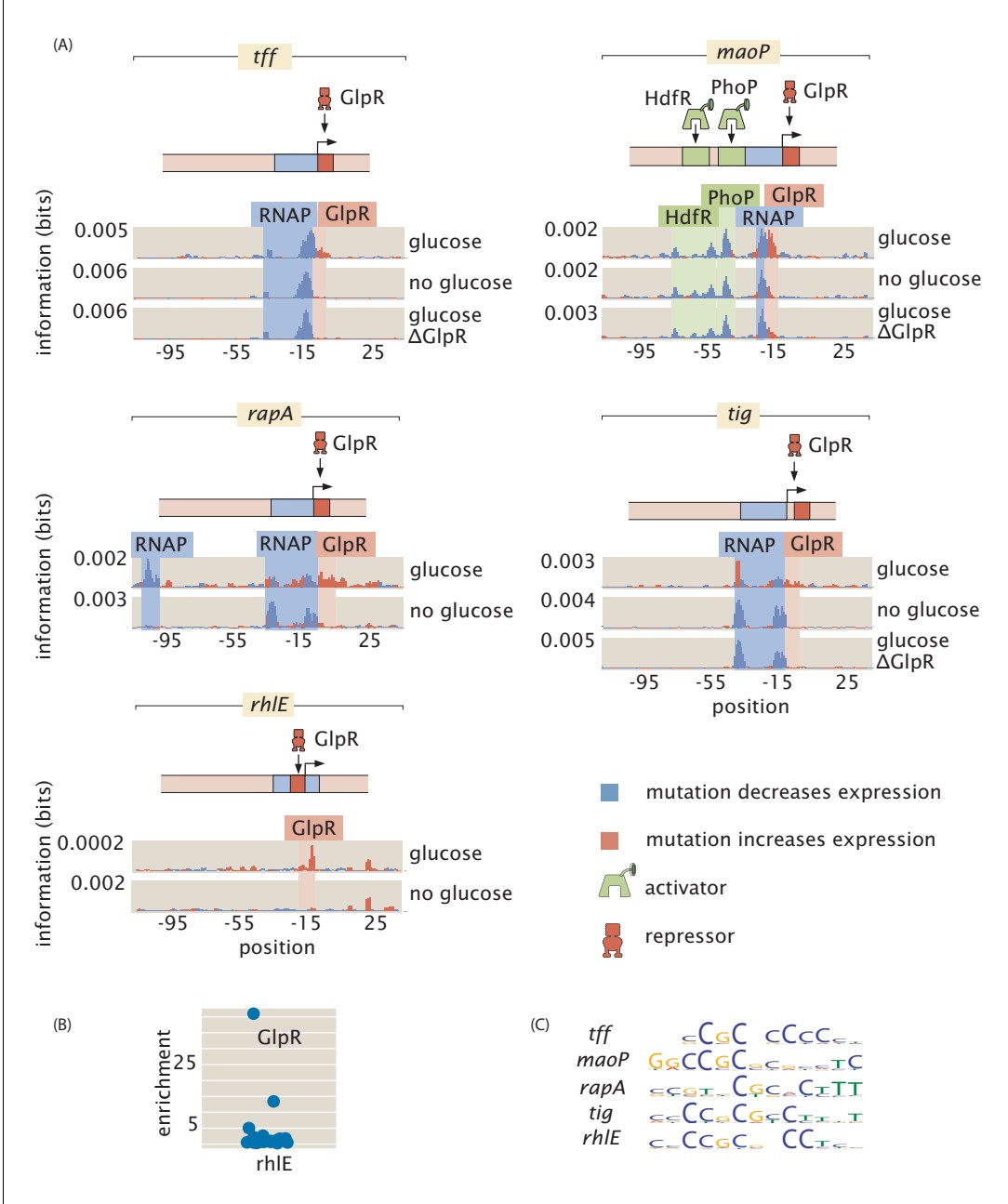

**Figure 7.** GlpR as a widely acting regulator. (A) Information footprints for the promoters which we found to be regulated by GlpR, all of which were previously unknown. Activator-binding regions are highlighted in green, repressor-binding regions in red, and RNAP-binding regions in blue. (B) GlpR was demonstrated to bind to *rhlE* by mass spectrometry. (C) Sequence logos for GlpR-binding sites. Binding sites in the promotes of *tff, tig, maoP, rhlE,* and *rapA* have similar DNA binding preferences as seen in the sequence logos and each transcription-factor-binding site binds strongly only in the presence of glucose (As shown in (A)). These similarities suggest that the same transcription factor binds to each site. To test this hypothesis, we knocked out GlpR and ran the Reg-Seq experiments for *tff, tig,* and *maoP*. In (A), we see that knocking out GlpR removes the binding signature of the transcription factor. Numeric values for the binding locations can be found in *Figure 7—source data 1*.
The online version of this article includes the following source data for figure 7:

**Source data 1.** Data for information footprints, PWMs, and mass spectrometry in *Figure 7*.

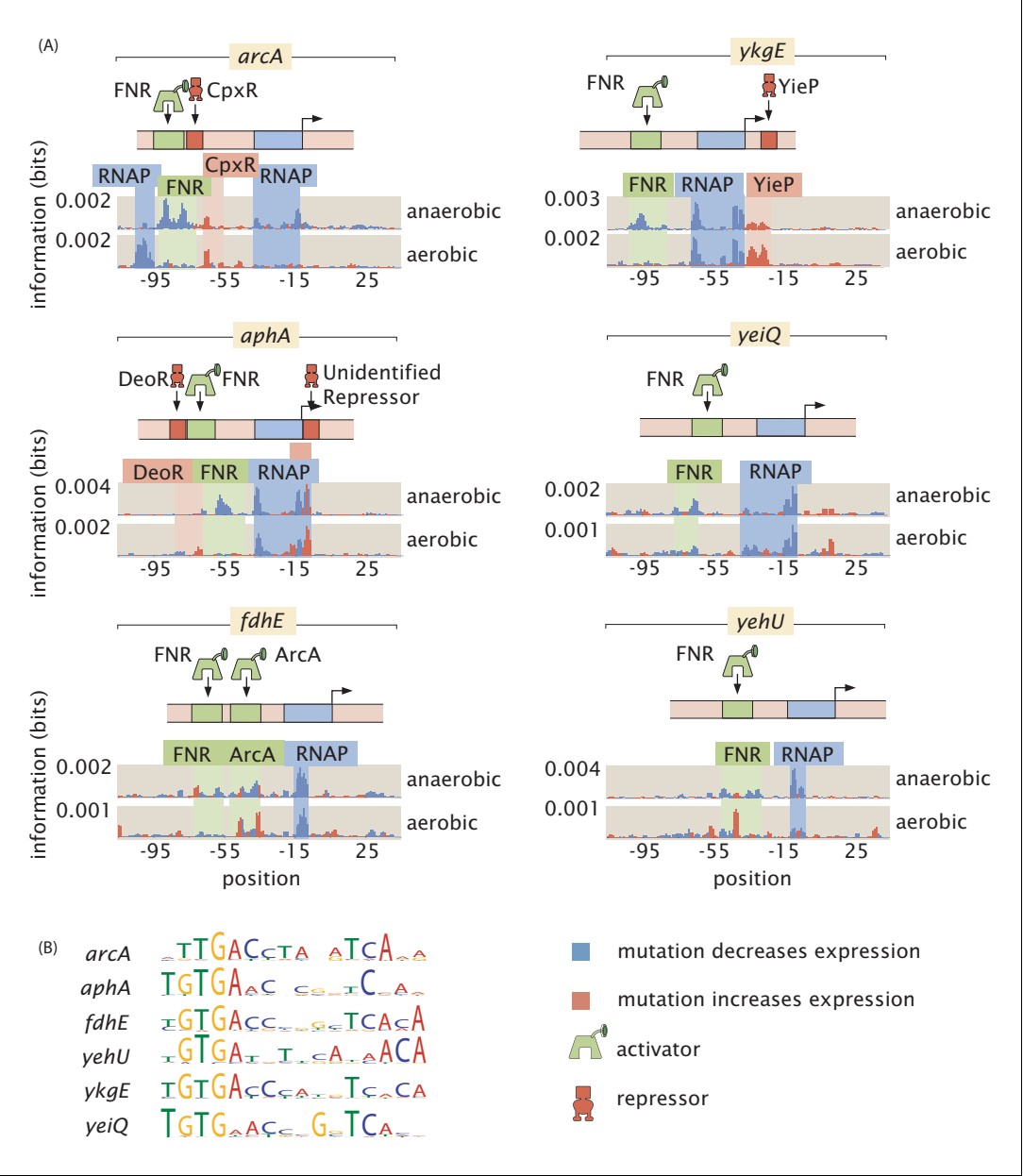

**Figure 8.** FNR as a global regulator. FNR is known to be upregulated in anaerobic growth, and here we found it to regulate a suite of six genes. In aerobic growth conditions, the putative FNR sites are weakened. (**A**) Information footprints for the six regulated promoters. Activator binding regions are highlighted in green, repressor-binding regions in red, and RNAP binding regions in blue. (**B**) Sequence logos for the FNR-binding sites displayed in (**A**). The DNA binding preference of the six sites are shown to be similar from their sequence logos. Numeric values for the binding locations can be found in *Figure 8—source data 1*.

The online version of this article includes the following source data for figure 8:

**Source data 1.** Data for information footprints and PWMs in *Figure 8*.

## Discussion

The study of gene regulation is one of the centerpieces of modern biology. As a result, it is surprising that in the genome era, our ignorance of the regulatory landscape in even the best-understood model organisms remains so vast. Despite understanding the regulation of transcription initiation in bacterial promoters (*Browning and Busby, 2016*), and how to tune their expression (*Barnes et al., 2019*), we lack an experimental framework to unravel understudied promoter architectures at scale.

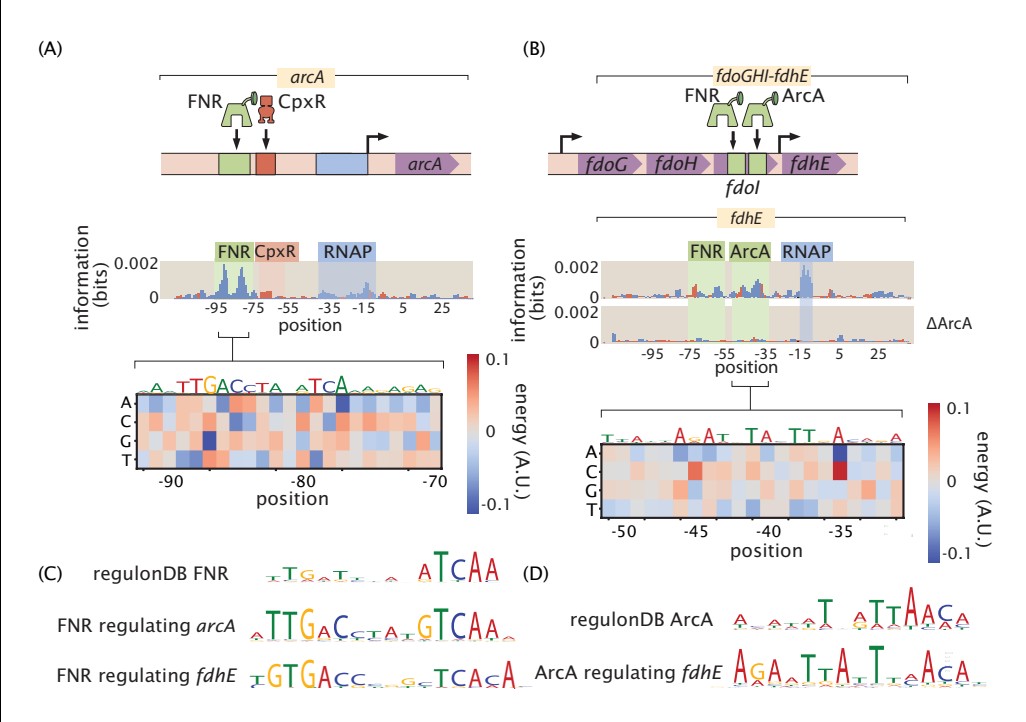

**Figure 9.** Inspection of a genetic circuit. (A) Here, the information footprint of the *arcA* promoter is displayed along with the energy matrix describing the discovered FNR-binding site. (B) Intra-operon regulation of *fdhE* by both FNR and ArcA. The information footprint of *fdhE* is displayed. The discovered sites for FNR and ArcA are highlighted and the energy matrix for ArcA is displayed. A TOMTOM (*Gupta et al., 2007*) search of the binding motif found that ArcA was the most likely candidate for the transcription factor. The displayed information footprint from a knockout of ArcA demonstrates that the binding signature of the site, and its associated RNAP site, are no longer determinants of gene expression. (C) Sequence logos for FNR generated from both the sites cataloged in RegulonDB, as well as the discovered sites regulating *arcA* and *fdhE*. (D) Sequence logos for ArcA from sites contained in RegulonDB and the ArcA site regulating *fdhE*. Numeric values for the binding locations can be found in *Figure 9—source data 1*.

The online version of this article includes the following source data for figure 9:

**Source data 1.** Data for information footprints, energy matrices, and PWMs in *Figure 9B*.

As such, in our view, one of the grand challenges of the genome era is the need to uncover the regulatory landscape for each and every organism with a known genome sequence. Given the ability to read and write DNA sequences at will, we are convinced that to make that reading of DNA sequence truly informative about biological function and to give that writing the full power and poetry of what Crick christened 'the two great polymer languages', we need a full accounting of how the genes of a given organism are regulated and how environmental signals communicate with the transcription factors that mediate that regulation – the so-called 'allosterome' problem (*Lindsley and Rutter, 2006*). The work presented here provides a general methodology for making progress on the former problem and also demonstrates that, by performing Reg-Seq in different growth conditions, we can make headway on the latter problem as well.

The advent of cheap DNA sequencing offers the promise of beginning to achieve this grand challenge in the form of MPRAs as reviewed in *Kinney and McCandlish, 2019*. A particular implementation of such methods was christened Sort-Seq (*Kinney et al., 2010*) and was demonstrated in the context of well understood regulatory architectures. A second generation of the Sort-Seq method (*Belliveau et al., 2018*) established a full protocol for regulatory dissection through the use of DNA-affinity chromatography and mass spectrometry which made it possible to identify the transcription factors that bind the putative binding sites discovered by Sort-Seq. However, there were critical shortcomings in the method, not least of which was that it lacked the scalability to uncover the regulatory genome in a more multiplexed manner.

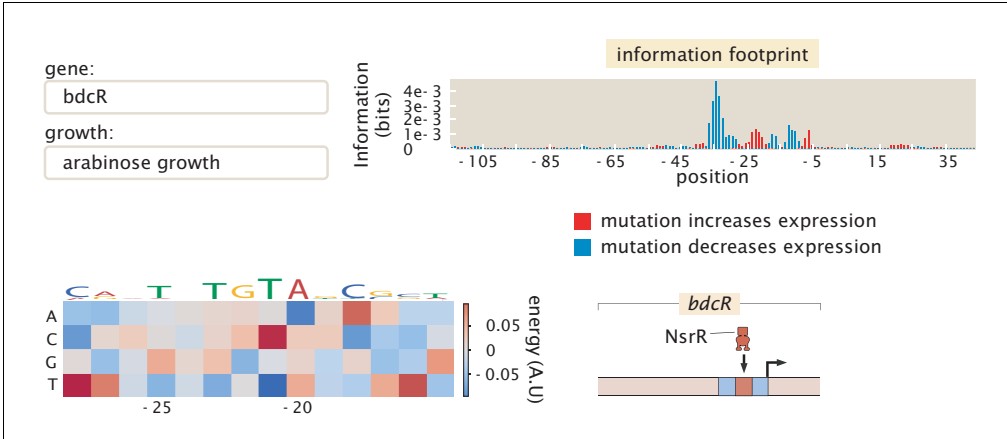

**Figure 10.** Representative view of the interactive figure that is available online. This interactive figure captures the entirety of our dataset. Each figure features a drop-down menu of genes and growth conditions. For each such gene and growth condition, there is a corresponding information footprint revealing putative binding sites, an energy matrix that shows the strength of binding of the relevant transcription factor to those binding sites and a cartoon that schematizes the newly-discovered regulatory architecture of that gene. Numeric values for the binding locations can be found in *Figure 10—source data 1*.

The online version of this article includes the following source data for figure 10:

**Source data 1.** Data for information footprints, energy matrices, and PWMs in *Figure 10*.

The work presented here builds on the foundations laid in previous studies by invoking RNA-Seq as a readout for the level of expression of the promoter mutant libraries needed to infer information footprints and their corresponding energy matrices and sequence logos. The original inference and hypothesis generation is followed by a combination of mass spectrometry, comparison of binding motifs, and gene knockouts to identify the transcription factors that bind those sites. The case studies described in the main text showcase the ability of the Reg-Seq method to deliver on the promise of beginning to uncover the regulatory genome systematically. The extensive online resources hint at a way of systematically reporting those insights in a way that can be used by the community at large to develop regulatory intuition for biological function and to design novel regulatory architectures using energy matrices.

However, several shortcomings remain in the approach introduced here. First, the current implementation of Reg-Seq is not fully automated for various aspects in the experimental pipeline; for example, manual examination of information footprints is used to generate testable regulatory hypotheses. As the method is scaled up further, this can limit throughput of the analysis. To address this for future work, we have created an automated methodology for identifying putative binding sites, which we describe in the Materials and methods section, that will simplify future scaled up efforts at identifying putative binding sites. All putative binding sites reported in this study either were identified through the automated methodology or have additional evidence for their presence such as mass spectrometry. In addition, these regulatory hypotheses can be converted into gene regulatory models using statistical physics (*Buchler et al., 2003*; *Bintu et al., 2005*). However, here too, as the complexity of the regulatory architectures increases, it will be of great interest to use automated model generation as suggested in a recent biophysically based neural network approach (*Tareen and Kinney, 2019*).

Another key challenge faced by the methods described here is that the mass spectrometry and the gene knockout confirmation aspects of the experimental pipeline remain low-throughput and, at times, inconclusive. Occasionally, we have found it challenging to observe weaker binding sites when multiple strong binding sites are also present. This was the case for the *marRAB* operon. To make our transcription factor identification methods more high-throughput, we have begun to explore a new generation of experiments such as in vitro binding assays that will make it possible to accomplish transcription factor identification in a multiplexed manner. Specifically, we are exploring multiplexed mass spectrometry measurements and multiplexed Reg-Seq on libraries of gene

knockouts as ways to break the identification bottleneck. Transcription factor identification using Reg-Seq is also complicated by the growth conditions that we can test; for the 18 genes that we tested and labeled as 'inactive' in this study, Reg-Seq did not reveal even an RNAP-binding site, suggesting that the proper growth condition to get high levels of expression was not used, or perhaps that the mutation window chosen for the gene does not capture a highly transcribing TSS. While information on the location of a TSS is available for 2500 of 2600 operons in *E. coli* (*Santos-Zavaleta et al., 2019*), this information does not guarantee those sites will have high transcription in the growth conditions studied. Similarly, many genes have multiple TSS that can be active under different growth conditions. In these cases, we are limited both by the finite set of growth conditions we test as well as by the length of the mutation window, as it cannot always capture all TSS.

Another shortcoming of the current implementation of the method is that it misses regulatory action at a distance. Indeed, our laboratory has invested a significant effort in exploring such long-distance regulatory action in the form of DNA looping in bacteria (*Johnson et al., 2012*; *Han et al., 2009*) and V(D)J recombination in jawed vertebrates (*Lovely et al., 2015*; *Hirokawa et al., 2020*). It is well known that transcriptional control through enhancers in eukaryotic regulation is central in contexts ranging from embryonic development to hematopoiesis (*Melnikov et al., 2012*). The current incarnation of the methods, as described here, have focused on contiguous regions in the vicinity of the transcription start site (within the 160 base pair mutagenized window). Clearly, to dissect the entire regulatory genome, these methods will have to be extended to non-contiguous regions of the genome.

Despite their limitations, the findings from this study provide a foundation for systematic, multiplexed regulatory dissections. We have developed a method to pass from complete regulatory ignorance to designable, regulatory architectures and we are hopeful that others will adopt these methods with the ambition of uncovering the regulatory architectures that preside over their organisms of interest.

## Materials and methods

Here, we provide an overview of the key methodological aspects of Reg-Seq. Extensive details of the methods used in this study can also be found on the GitHub Wiki associated with this work.

### Library design and construction

We selected 113 TSS from the *E. coli* K12 genome for experiments. The promoter regions analyzed in this study were each 160 base pairs in length, a region that includes 45 base pairs downstream and 115 base pairs upstream of each TSS. The general principles by which we selected each TSS were to first prioritize those TSS which have been extensively experimentally validated and catalogued in RegulonDB (*Santos-Zavaleta et al., 2019*) or EcoCyc (*Keseler et al., 2017*). Secondly, we selected those sites which had evidence of active transcription from RACE experiments (*Mendoza-Vargas et al., 2009*) and were listed in RegulonDB. If a TSS lacked both experimental evidence and active transcription as indicated by RACE experiments, we used the computationally predicted TSS as indicated on RegulonDB (*Santos-Zavaleta et al., 2019*) or EcoCyc (*Keseler et al., 2017*) and determined previously by *Huerta and Collado-Vides, 2003*. If there were multiple TSS located upstream of the gene in question, we selected the TSS closest to the gene start, unless selecting one further upstream would allow for multiple TSS to be contained in the 160 base pair mutated region analyzed for each promoter. Not all TSS locations are known, and many genes have multiple TSS. The exact start sites used, as well as the reasoning behind our selection of each TSS, are listed in *Supplementary file 1*.

Promoter variants were synthesized on a microarray (TWIST Bioscience, San Francisco, CA). The sequences were designed computationally such that each base in the 160 base pair promoter region has a 10% probability of being mutated. For each promoter's oligonucleotide library, we ensured that the mutation rate as averaged across all sequences was kept between 9.5% and 10.5%, otherwise the library was regenerated. There are an average of 2200 unique promoter sequences per gene (for an analysis of how our results depend upon number of unique promoter sequences see *Appendix 3—figure 1*). The library arrived lyophilized (76 pmol) and was resuspended in 100 μL of TE (pH 8.0). Of the resuspended oligonucleotide, 1 μL was amplified for 12 cycles with New England

Biolabs Q5 High-Fidelity 2x Master Mix (NEB, Ipswich, MA) to increase the quantity of DNA in the library. Unless otherwise stated, all amplifications were performed using this polymerase mixture.

The PCR product was then run on a 2% TAE agarose gel, and approximately 200 base pair amplicons were extracted using a Zymoclean Gel DNA Recovery Kit (Zymo Research, Irvine, CA). To add a random 20-nucleotide barcode to each oligonucleotide, 1 ng of the purified DNA library was amplified for 10 PCR cycles using primers containing random 20-nucleotide DNA overhangs. All primer sequences can be found in *Supplementary file 2*. After cleaning this PCR product using a Zymo Clean and Concentrator Kit (Zymo Research, Irvine, CA), the library was cloned into the plasmid backbone of pJK14 (SC101 origin) (*Kinney et al., 2010*) using Gibson Assembly. An illustration of this plasmid is displayed in *Appendix 1—figure 1*. Genetic constructs were electroporated into *E. coli* K-12 MG1655 (*Blattner et al., 1997*) and plated on LB plates with kanamycin. After 24 hr of growth on plates, libraries were scraped and inoculated into M9 media with 0.5% glucose in preparation for DNA sequencing.

All genetic barcodes were inserted 120 base pairs from the 5' end of the mRNA, containing 45 base pairs from the targeted regulatory region, 64 base pairs containing primer sites used in the construction of the plasmid, and 11 base pairs containing a three frame stop codon. Exact sequences of primers and spacer sequences for the constructs are listed in *Supplementary file 2*. Following each genetic barcode, there is an RBS, a GFP-coding region, and a terminator.

## Preparation of libraries for sequencing

To prepare cDNA libraries for sequencing, cells were grown to an optical density of 0.3 and RNA was stabilized using Qiagen RNA Protect (Qiagen, Hilden, Germany). Lysis was performed using lysozyme (Sigma Aldrich, Saint Louis, MO) and RNA isolated using the Qiagen RNA Mini Kit. Reverse transcription was preformed using Superscript IV (Invitrogen, Carlsbad, CA) with a specific primer for the labeled mRNA. qPCR was then performed in triplicate to check the level of DNA contamination. Any sample that had contaminating DNA at a level of 5% or more of the mRNA concentration was discarded. DNA libraries were prepared by growing cells to an optical density of 0.3 and isolating plasmid DNA with a spin miniprep kit (Qiagen, Hilden, Germany).

## Sequencing

After preparing the barcoded libraries, we used next-generation sequencing (NGS) to map promoters to their respective barcodes. Sequencing libraries (both cDNA and DNA) had unindexed illumina flow cell adaptors attached via PCR, using primers that amplified a 221 base pair region that included the random barcode. We limited PCR cycles to exponential amplification, as determined by qPCR. Specifically, when we performed qPCR to check for DNA contamination, we also determined the number of cycles at which each sample reached exponential amplification, and then repeated the PCR reactions with the determined number of cycles to limit bias. After amplification, libraries were cleaned using a Zymo Clean and Concentrator kit and analyzed on an Agilent 2100 Bioanalyzer (Agilent, Santa Clara, CA). Samples were submitted to NGX Bio (NGX Bio, South Plainfield, NJ) for 150 base pair paired-end sequencing on a Hi-Seq 2500 (Illumina, San Diego, CA). We typically acquired 250 million total reads for mapping of libraries. Further details of how we process the sequences can be found in Appendix 1 Section 'Sequencing Analysis' and the GitHub Wiki associated with this work.

To quantify relative gene expression values for each promoter mutant in our library, we next grew cells expressing the DNA libraries in various growth conditions to an OD600 of 0.3. DNA and cDNA libraries were prepared in the same way as stated previously, and were sequenced at the Millard and Muriel Jacobs Genetics and Genomics Laboratory at Caltech on a HiSeq 2500 with a 100 base pair single read flow cell. An average of five unique 20 base pair barcodes per variant promoter was used for the purpose of counting transcripts. Specifically, for each promoter variant the number of sequences from the DNA library and the number of sequences produced from mRNA are determined.

## Determination of energy matrices

Energy matrices are used to represent the binding energy contribution for each nucleotide in a DNA sequence. We use relative gene expression values, as determined by counting genetic barcodes

from NGS data for each mutated variant of a given regulatory sequence, and infer the energy contribution of each nucleotide by maximizing the mutual information between the rank-ordered binding strength predictions from the energy matrix and the gene expression data. We also perform this maximization using MCMC. Further discussion of how energy matrices are inferred can be found in Appendix 3 Section 'Energymatrix inference' and on the GitHub Wiki that accompanies this study.

In each energy matrix plot, a red box indicates that a mutation to a nucleotide in that position decreases the energy of transcription factor binding, while a blue box indicates that a mutation at a given nucleotide position increases transcription-factor-binding energy. Energy matrices are typically given in arbitrary units, but the method by which we can assign absolute units in $k_b T$ is covered in Appendix 3 Section 'Inference of scaling factors for energy matrices'.

## DNA-affinity chromatography and mass spectrometry

Upon identifying a putative transcription-factor-binding site, we used DNA-affinity chromatography, as performed in *Belliveau et al., 2018*, to isolate and enrich for the transcription factor of interest. In brief, we order biotinylated oligos of our binding site of interest (Integrated DNA Technologies, Coralville, IA) along with a control, 'scrambled' sequence, that we expect to have no specificity for the given transcription factor. We tether these oligos to magnetic streptavidin beads (Dynabeads MyOne T1; ThermoFisher, Waltham, MA), and incubate them overnight with whole cell lysate grown in the presences of either heavy (with $^{15}$N) or light (with $^{14}$N) lysine for the experimental and control sequences, respectively. The next day, proteins are recovered by digesting the DNA with the PtsI restriction enzyme (New England Biolabs, Ipswich, MA), whose cut site was incorporated into all designed oligos.

Protein samples were then prepared for mass spectrometry by either in-gel or in-solution digestion using the Lys-C protease (Wako Chemicals, Osaka, Japan). Liquid chromatography coupled mass spectrometry (LC-MS) was performed as previously described by *Belliveau et al., 2018*, and is further discussed in Appendix 3 Section 'Processing of mass spectrometry experiments'. SILAC labeling was performed by growing cells (Δ LysA) in either heavy isotope form of lysine or its natural form.

It is also important to note that while we utilized the SILAC method to identify the transcription factor identities, our approach does not require this specific technique. Specifically, our method only requires a way to contrast between the copy number of proteins bound to a target promoter in relation to a scrambled version of the promoter. In principle, one could use multiplexed proteomics based on isobaric mass tags (*Pappireddi et al., 2019*) to characterize up to 10 promoters in parallel. Isobaric tags are reagents used to covalently modify peptides by using the heavy-isotope distribution in the tag to encode different conditions. The most widely adopted methods for isobaric tagging are the isobaric tag for relative and absolute quantitation (iTRAQ) and the tandem mass tag (TMT). This multiplexed approach involves the fragmentation of peptide ions by colliding with an inert gas. The resulting ions are resolved in a second MS-MS scan (MS2).

Only a subset (13) of all transcription factor targets were identified by mass spectrometry due to limitations in scaling the technique to large numbers of targets. The transcription factors identified by this method are enriched more than any other DNA binding protein, with p<0.01 using the outlier detection method as outlined by *Cox and Mann, 2008*, with corrections for multiple hypothesis testing using the method proposed by *Benjamini and Hochberg, 1995*. Details on data processing can be found in Appendix 3 Section 'Processing of mass spectrometry experiments'. A detailed explanation of all experimental and computational steps can be found in the GitHub Wiki that accompanies this work.

## Construction of knockout strains

Conducting DNA-affinity chromatography followed by mass spectrometry on putative binding sites resulted in potential candidates for the transcription factors that bind to the target region. For some cases, to verify that a given transcription factor is, in fact, regulating a given promoter, we repeated the RNA sequencing experiments on strains in which the transcription factor of interest has been knocked out.

To construct the knockout strains, we ordered strains from the Keio collection (*Yamamoto et al., 2009*) from the Coli Genetic Stock Center. These knockouts were put in a MG1655 background via

phage P1 transduction and verified with Sanger sequencing. To remove the kanamycin resistance that comes with the strains from the Keio collection, we transformed in the pCP20 plasmid (*Datsenko and Wanner, 2000*), induced FLP recombinase, and then selected for colonies that no longer grew on either kanamycin or ampicillin, verifying both loss of the chromosomally integrated kanamycin resistance and the pCP20 plasmid which confers ampicillin resistance. Finally, we transformed our desired promoter libraries into the constructed knockout strains, allowing us to perform the RNA sequencing in the same context as the original experiments.

## Automated putative binding site algorithm

We introduce a systematized way of identifying the locations of binding sites to supplement manual curation (described in the Section 'Manual selection of binding sites'). As illustrated in *Figure 11*, for a given information footprint, we average over 15 base pair 'windows'. We then determine which base pairs are part of a regulatory region by setting an information threshold of $2.5 \times 10^{-4}$ bits. Threshold selection is described in Appendix 2 Section 'False positive and false negative rates'. All base pair positions that pass the information threshold were then joined into regulatory regions. We consider 'activator-like' (mutation decreases expression) and 'repressor-like' (mutation increases expression) base pairs separately. This means that it is possible to have overlapping repressor- and activator-binding sites identified. We join any base pair positions within four base pairs of each other into single regulatory regions. We then find the edges of the region by trimming off any base pairs at the edge that are below the information threshold (even if the 15 base pair average is above the threshold). While we can often resolve overlapping or nearby repressors from activators, a limitation of this method of identification is that is cannot resolve two activators or two repressors that are very close to each other or overlapping.

To identify RNAP-binding sites, we compare the sequence preference (through energy matrices and sequence logos) to experimentally validated examples of RNAP sites. We have examples of energy matrices for the $\sigma^{70}$ RNAP site from *Belliveau et al., 2018*. For energy matrices of other $\sigma$ factor binding sites, such as $\sigma^{32}$ and $\sigma^{28}$, we use energy matrices generated from within the Reg-Seq experiment itself. For a $\sigma^{32}$ binding site, for example, we used the example from the *hslU* gene. For a $\sigma^{28}$ binding site, we used the energy matrix generated from the *dnaE* gene. We 'scan' the example energy matrices across the mutated region. For each position in the region, we calculate the Pearson correlation coefficient between the example RNAP energy matrix and the inferred energy matrix at that position. We find RNAP-binding site locations by thresholding the Pearson correlation coefficients at a value of 0.45. When performing manual curation of binding sites, we visually compare the sequence logos of the example RNAP-binding sites to the sequence logos of putative binding sites. Further details of the method to create energy matrices and compare them to known motifs are given in Appendix 3 Section 'Energy matrix inference' and Appendix 3 Section 'TOMTOM motif comparison', respectively. Further, a detailed discussion of energy matrix construction is provided in the Sequencing Analysis GitHub Wiki page that accompanies this work.

## Manual selection of binding sites

Similarly to the automated method of locating putative binding regions, we look for regions of high mutual information in the information footprints. While there was no hard cut-off for mutual information values during manual curation, we select clusters of base pairs that have a similar average information value ($2.5 \times 10^{-4}$ bits) to that described in the Section 'Automated putative binding site algorithm'.

During manual curation of binding sites, we also disqualify any binding sites where there are only three or fewer base pairs with high values in the mutual information footprint. The logic behind this decision is that individual bases with very high mutual information can potentially indicate that a putative binding site is only active when a certain mutation occurs. In turn, the binding site would not be active in wild-type conditions. To explain why this is, consider that a typical binding site mutation, at any given base pair, will significantly *weaken* the binding site of interest. Therefore, each of those mutated base pairs is said to have a 'large effect' on expression. For a very poor binding site that is not active in the wild-type case, most mutations will further weaken a site which already will have only a minor effect on gene expression. However, for a small number of base pairs, a mutation can occur that makes the DNA bind more tightly to the transcription factor, making it relevant for

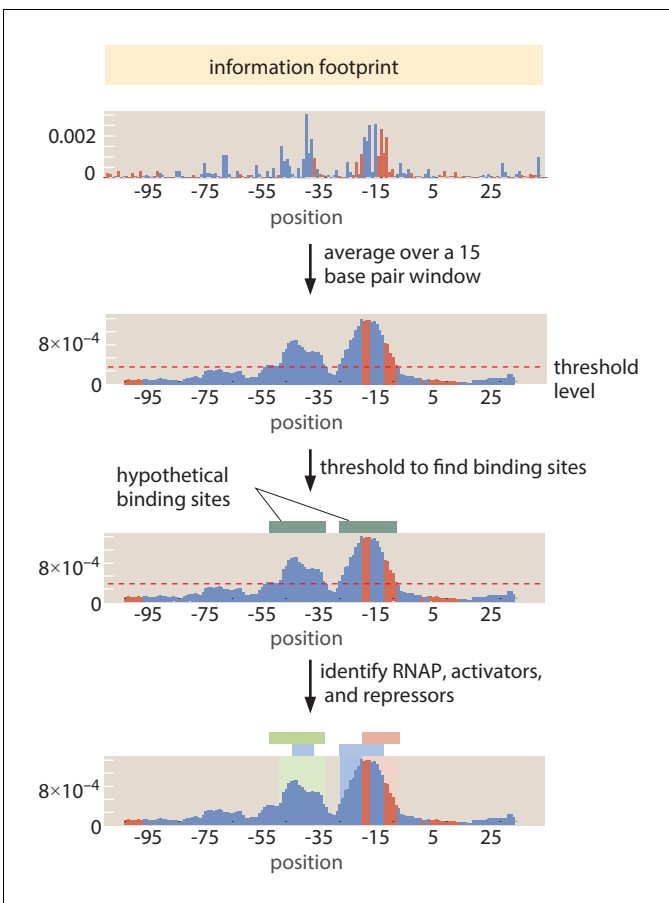

**Figure 11.** Procedure to identify binding site regions automatically. First, an information footprint is generated for the target region. Next, the information footprint is smoothed over a 15 base pair sliding window and a threshold of $2.5 \times 10^{-4}$ bits is applied to identify regions of interest. RNAP-binding sites are first identified (in blue), and the remainder of the regulatory regions are identified as repressor-binding sites (if they tend to increase expression on mutation from wild type) or activator-binding sites (if they tend to decrease expression upon mutation).

The online version of this article includes the following source data for figure 11:

**Source data 1.** Information footprint data displayed in *Figure 11*.

gene expression. Therefore, in the case of an extremely weak binding site that is not relevant in the wild type condition, there can still be a small number of highly informative bases. Initial hypothesis generation in Reg-Seq was done manually. However, all those sites that are reported in *Table 2* that do not have additional validation through mass spectrometry, gene knockouts, or bioinformatics appear in the set of putative binding sites generated by the method described in Section 'Automated putative binding site algorithm'.

## Code and data availability

An in-depth discussion of all experimental protocols and mathematical analysis used in this study can be found on the GitHub Wiki for this study (*Ireland, 2020* https://github.com/RPGroup-PBoC/RegSeq/wiki (copy archived at https://github.com/elifesciences-publications/RegSeq). All code used for processing data and plotting as well as the final processed data, plasmid sequences, and primer sequences can also be found on the GitHub repository(archived by Zenodo; https://doi.org/10.5281/zenodo.3966687). Energy matrices were generated using the MPAthic software (*Ireland and Kinney, 2016*). All raw sequencing data is available at the Sequence Read Archive (accession no. PRJNA599253 and PRJNA603368). All inferred information footprints and energy matrices can be found on the GitHub repository (archived by Zenodo; https://doi.org/10.5281/zenodo.3966687). All

mass spectrometry raw data is available on the CaltechData repository (https://doi.org/10.22002/d1.1336).

## Acknowledgements

We are grateful to Rachel Banks, Stephanie Barnes, Curt Callan, Griffin Chure, Ana Duarte, Vahe Galstyan, Hernan Garcia, Soichi Hirokawa, Thomas Lecuit, Heun Jin Lee, Madhav Mani, Muir Morrison, Steve Quake, Manuel Razo-Mejia, Gabe Salmon, and Guillaume Urtecho for useful discussion and feedback on the manuscript. Guillaume Urtecho and Sri Kosuri have been instrumental in providing key advice and protocols at various stages in the development of this work. We would like to thank Jost Vielmetter and Nina Budaeva for providing access to their Cell Disruptor. Brett Lomenick provided crucial help and advice with protein preparation. We also thank Igor Antoshechkin for his help with sequencing at the Caltech Genomics Facility.

Funding: We are deeply grateful for support from NIH Grants DP1 OD000217 (Director's Pioneer Award) and 1R35 GM118043-01 (Maximizing Investigators Research Award) which made it possible to undertake this multi-year project. NMB was supported by an HHMI International Student Research Fellowship. SMB was supported by the NIH Institutional National Research Service Award (5T32GM007616-38) provided through Caltech. The Proteome Exploration Laboratory is supported by, the Beckman Institute, and NIH 1S10OD02001301.

## Additional information

### Funding

| Funder | Grant reference number | Author |
| --- | --- | --- |
| National Institutes of Health | Director's Pioneer Award | Rob Phillips |
| National Institutes of Health | National Research Service Award | Suzannah M Beeler |
| National Institutes of Health | Maximizing Investigators Research Award | Rob Phillips |
| Howard Hughes Medical Institute | International Student Research Fellowship | Nathan M Belliveau |
| National Institutes of Health | 1S10OD02001301 | Annie Moradian Michael J Sweredoski |

The funders had no role in study design, data collection and interpretation, or the decision to submit the work for publication.

### Author contributions

William T Ireland, Conceptualization, Data curation, Software, Formal analysis, Validation, Investigation, Visualization, Methodology, Writing - original draft, Writing - review and editing; Suzannah M Beeler, Conceptualization, Formal analysis, Validation, Investigation, Visualization, Methodology, Writing - original draft, Writing - review and editing; Emanuel Flores-Bautista, Software, Formal analysis, Investigation, Visualization, Methodology, Writing - original draft, Writing - review and editing; Nicholas S McCarty, Resources, Software, Writing - review and editing; Tom Röschinger, Data curation, Software, Writing - review and editing; Nathan M Belliveau, Conceptualization, Methodology, Writing - review and editing; Michael J Sweredoski, Formal analysis, Methodology, Writing - review and editing; Annie Moradian, Methodology, Writing - review and editing; Justin B Kinney, Software, Methodology, Writing - review and editing; Rob Phillips, Conceptualization, Resources, Supervision, Funding acquisition, Validation, Visualization, Methodology, Writing - original draft, Project administration, Writing - review and editing

### Author ORCIDs

William T Ireland https://orcid.org/0000-0003-0971-2904
Suzannah M Beeler http://orcid.org/0000-0002-1930-4827

Tom Röschinger [iD] https://orcid.org/0000-0002-4900-3216
Nathan M Belliveau [iD] https://orcid.org/0000-0002-1536-1963
Michael J Sweredoski [iD] http://orcid.org/0000-0003-0878-3831
Annie Moradian [iD] http://orcid.org/0000-0002-0407-2031
Justin B Kinney [iD] http://orcid.org/0000-0003-1897-3778
Rob Phillips [iD] https://orcid.org/0000-0003-3082-2809

## Decision letter and Author response

Decision letter https://doi.org/10.7554/eLife.55308.sa1
Author response https://doi.org/10.7554/eLife.55308.sa2

# Additional files

## Supplementary files

• Supplementary file 1. This file contains the presumed location of the TSS for each promoter region in Reg-Seq. It additionally contains the logic behind the choice of TSS when there are multiple options.

• Supplementary file 2. This file contains all primers used in the Reg-Seq experiment. Additionally, it contains the flanking sequences of the mutated inserts and the barcodes used to label the growth conditions in the Reg-Seq experiment.

• Supplementary file 3. This file contains all transcription-factor-binding sites identified either through the automated binding site algorithm or which were identified manually and have additional evidence for binding. The starting and ending base pairs for each binding site, and whether the transcription factor acts as an activator or repressor are listed.

• Source code 1. This file contains custom python scripts used in the processing and analysis of sequencing data.

• Transparent reporting form

## Data availability

Sequencing data has been deposited in the SRA under accession no.PRJNA599253 and PRJNA603368 Mass spectrometry data is deposited in the CalTech data repository at https://doi.org/10.22002/d1.1336 Model files and inferred information footprints are deposited in the CalTech data repository at https://doi.org/10.22002/D1.1331 Processed sequencing data sets and analysis software are available in the GitHub repository available at https://doi.org/10.5281/zenodo.3953312.

The following datasets were generated:

| Author(s) | Year | Dataset title | Dataset URL | Database and Identifier |
|---|---|---|---|---|
| Ireland W, Beeler S, Flores-Bautista E, Belliveau N, Sweredoski M, Moradian A, Kinney J, Phillips R | 2019 | RNAseq data for the Reg-Seq project | https://www.ncbi.nlm.nih.gov/sra/PRJNA599253 | NCBI Sequence Read Archive, PRJNA599253 |
| Ireland W, Beeler S, Flores-Bautista E, Belliveau N, Sweredoski M, Moradian A, Kinney J, Phillips R | 2019 | Mass Spectrometry data for the Reg-Seq project | https://doi.org/10.22002/d1.1336 | CalTech Data, 10.22002/d1.1336 |
| Ireland W, Beeler S, Flores-Bautista E, Belliveau N, Sweredoski M, Moradian A, Kinney J, Phillips R | 2019 | Sequencing Data for mapping mutated constructs | http://www.ncbi.nlm.nih.gov/bioproject/603368 | NCBI BioProject, PRJNA603368 |

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

# Appendix 1

## Extended details of experimental design

### Choosing target genes

Genes in this study were chosen to cover several different categories. Twenty-nine genes had at least one transcription-factor-binding site listed in RegulonDB and were picked to validate our method under a number of conditions (15 with relevant high evidence sites). Thirty-seven were chosen because the work of *Schmidt et al., 2016* demonstrated that gene expression changed significantly under different growth conditions. A handful of genes such as *minC*, *maoP*, or *fdhE* were chosen because we found either their physiological significance interesting, as in the case of *minC*, whose product is crucial for cell division and proper partitioning of the cell into two equal sized daughters in *E. coli* (*Lutkenhaus, 2007*). Alternatively, for some cases we found the gene regulatory question interesting, such as for the intra-operon regulation demonstrated by *fdhE*. The remainder of the genes were chosen because they had no regulatory information, often had minimal information about the function of the gene, and had an annotated transcription start site (TSS) in RegulonDB. A list of all genes chosen can be found in *Supplementary file 1*.

### Sequencing analysis

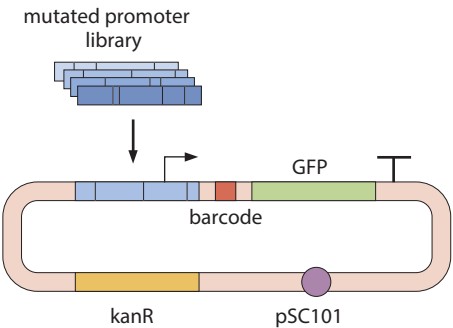

**Appendix 1—figure 1.** Schematic of the genetic construct used in this study. Mutated DNA libraries for each regulatory region were expressed from a pSC101 plasmid with kanamycin resistance (kanR). Each mutated sequence is 160 bp in length, which includes 45 bp downstream and 115 bp upstream of a given TSS. Each mutated sequence is flanked by primer-binding sites to facilitate cloning. The genetic construct also contains a random barcode, a ribosome-binding site (RBS), a GFP gene, and a terminator labeled with a large 'T'.

In this Appendix section, we provide further details associated with the analysis of next-generation sequencing (NGS) results, from both the 'mapping' experiment, in which each unique barcode is 'linked' to its corresponding mutated promoter region, and from the barcode sequencing experiments, in which the frequency of each barcode is counted and relative gene expression values determined. It is important to perform two sequencing experiments, in this manner, for a couple of reasons. Oligonucleotide libraries ordered from Twist Bioscience, which we use to construct promoter regions mutated at a 10% rate, are prone to random errors. This means that we do not fully know what is in the ordered library, and so it is necessary to sequence the full library and determine which mutations are present in each promoter region. The 'mapping' phase of experiments also serves to connect each random, genetic barcode (which is added via PCR with primer overhangs) to its corresponding, mutated promoter. By linking barcodes to promoters, we are able to build a 'codex' that enables us to count genetic barcodes and, in turn, understand the relative gene expression values for each mutant promoter sequence.

For the 'mapping' of genetic barcodes to their corresponding mutant promoter, we use paired-end sequencing, with 150 cycles for both Read 1 and Read 2, on a Hi-Seq 2500 machine. We acquired 250 million total reads for mapping of libraries.

In our analysis of FASTQ files, we removed any barcodes that were associated with a promoter variant which had insertions or deletions. Similarly, any genetic barcodes associated with multiple

promoter variants were removed from the analysis, as were any sequences which appeared only once (barcodes must appear at least two times to be analyzed, as the appearance of a single, unique barcode sequence could be attributed to a sequencing error). The paired end reads from this sequencing step were assembled using the FLASH tool (*Magoč and Salzberg, 2011*). Any sequence with a PHRED score less than 20 was then removed using the FastX toolkit (*Hannon, 2010*). The specific commands used for this step of our analysis are listed on the GitHub Wiki associated with this work.

To analyze the 'mapping' data and link each genetic barcode to its unique, mutagenized promoter region, we used a custom Python module, which can be found on the GitHub repository associated with this work. This module contains functions to check that sequences are the expected length, map unique barcodes to their corresponding promoter regions, and extract barcode sequences for subsequent sequencing experiments. We also provided a Jupyter notebook on the GitHub repository which provides a step-by-step walkthrough of the code used in processing sequencing data.

After mapping each barcode to its corresponding, mutated promoter region, we next 'count' barcodes, both DNA and cDNA, to determine the relative gene expression values for each mutated promoter. For barcode counting experiments, only the region containing the random, 20 bp barcode was sequenced. For each growth condition, each promoter library yielded 20,000 to 500,000 usable sequencing reads. If the dataset for a gene in a given growth condition did not have at least 20,000 reads, it was not analyzed further, as we consistently found that, below this threshold, we reached a regime wherein the inference reliability of MCMC was reduced.

When preparing DNA and cDNA for NGS, we add a 4nt barcode, via PCR, to the library isolated from each growth condition. These 4nt barcodes are used during data analysis both to map each library to its particular growth condition and to keep track of biological replicates, while the 20 bp barcodes can be used to identify each mutated promoter region. We performed all experiments with two biological replicates.

After collecting the FASTQ files, we perform quality filtering with FastX. We then perform barcode splitting with the FastX toolkit to separate each FASTQ file based on its growth condition, as well as separate the sequencing files based on whether they are derived from the DNA or cDNA library. Each experimental condition (both biological replicates, RNA vs. DNA, and growth conditions) receives a unique, 4nt barcode sequence, which enables us to identify where each library came from. Full details of our sequencing analysis methodologies, as well as all Python scripts, can be found on the GitHub repository associated with this work.

## Growth conditions

The growth conditions used in this study were inspired by *Schmidt et al., 2016*, a study which observed changes in the *E. coli* proteome under growth conditions similar to the ones presented. The growth conditions utilized in this study are tabulated in *Appendix 1—table 1*. The growth conditions explored here involved a range of environmental perturbations including altering the carbon source, inducing stress, or introducing trace metals. Unless otherwise noted in the caption of *Appendix 1—table 1*, the cells were grown in the medium at 37°C until reaching an OD of 0.3, at which point the cells were harvested and the RNA extracted. These growth conditions were chosen so as to span a wide range of growth rates, as well as to illuminate any carbon source specific regulators.

**Appendix 1—table 1.** All growth conditions used in the Reg-Seq study.

Heat shocked cells were exposed to 42℃ for 5 min upon reaching OD 0.3 as this is known to induce transcription by $\sigma^{32}$ (*Arsène et al., 2000*). Low oxygen growth cells were grown in a flask sealed with parafilm with minimal oxygen, although some was present as no anaerobic chamber was used. This level of oxygen stress was still sufficient to activate FNR binding, thus activating anaerobic metabolism. For cells grown with iron, upon reaching OD of 0.3 iron was added and cells were incubated for 10 min before harvesting RNA. Growth without cAMP was accomplished by the use of the JK10 strain (*Kinney et al., 2010*) which does not maintain its cAMP levels.

| Growth conditions |
| --- |
| M9 with glucose (0.5%) |
| M9 with acetate (0.5%) |
| M9 with arabinose (0.5%) |
| M9 with xylose (0.5%) and arabinose (0.5%) |
| M9 with succinate (0.5%) |
| M9 with trehalose (0.5%) |
| M9 with glucose (0.5%) and 5 mM sodium salycilate |
| LB |
| heat shock in M9 with glucose (0.5%) |
| LB in low oxygen |
| zinc, 5 mM ZnCl in M9 with glucose (0.5%) |
| iron, 5 mM FeCL in M9 with glucose (0.5%) |
| no cAMP in M9 with glucose (0.5%) |

All knockout experiment were performed in M9 with glucose except for the knockouts for *arcA*, *hdfR*, and *phoP* which were grown in LB.

## Appendix 2

### Validating Reg-Seq against previous methods and results

The work presented here is effectively a third-generation of the use of Sort-Seq methods for the discovery of regulatory architecture. The primary difference between the present work and previous generations (*Kinney et al., 2010*; *Belliveau et al., 2018*) is the use of RNA-Seq rather than fluorescence and cell sorting as a readout of the level of expression of our promoter libraries. As such, there are many important questions to be asked about the comparison between the earlier methods and this work. We attack that question in several ways. First, as shown in *Figure 3*, we have performed a head-to-head comparison of the two approaches to be described further in this section. Second, as shown in the next section, our list of candidate promoters included roughly 20% for which there is at least one experimentally validated transcription-factor-binding site. In these cases, we examined the extent to which our methods recover the known features of regulatory control about those promoters.

### Comparison between Reg-Seq by RNA-Seq and fluorescent sorting

As the basis for comparing the results of the fluorescence-based Sort-Seq approach with our RNA-Seq-based approach, we use information footprints and position weight matrices as our metrics.

When making these comparisons between the two methods, we compare the values of a position weight matrix (PWM), often displayed as a sequence logo, generated from the Sort-Seq and Reg-Seq methods. PWMs contain the probabilities that a given base will occur at a given position in the binding site. We calculate the Pearson correlation coefficient between the PWM values (represented as the height of the letters at each position) for the two methods. To compute the correlation coefficient, we use

$$r = \frac{\sum_{\alpha=1}^{4} \sum_{i=1}^{N} (x_{i,\alpha} - \bar{x})(y_{i,\alpha} - \bar{y})}{\sqrt{\sum_{\alpha=1}^{4} \sum_{i=1}^{N} (x_{i,\alpha} - \bar{x})^2} \sqrt{\sum_{\alpha=1}^{4} \sum_{i=1}^{N} (y_{i,\alpha} - \bar{y})^2}}, \tag{5}$$

where $x_{i,\alpha}$ and $y_{i,\alpha}$ are the entries of the PWM of nucleotide $\alpha$ at position $i$ obtained from Sort-Seq and Reg-Seq respectively, $N$ is the total length of the binding site, and $\bar{x}$ and $\bar{y}$ are the means of $x_{i,\alpha}$ and $y_{i,\alpha}$, respectively. As an example, consider the following sequence logo from a Sort-Seq experiment,

| Position | A | C | G | T |
|---|---|---|---|---|
| 1 | 0.01 | 0.01 | 0.03 | **0.95** |
| 2 | 0.04 | **0.83** | 0.06 | 0.07 |
| 3 | **0.70** | 0.17 | 0.11 | 0.02 |
| 4 | **0.86** | 0.01 | 0.10 | 0.03 |

and the same region resulting from a Reg-Seq experiment:

| Position | A | C | G | T |
|---|---|---|---|---|
| 1 | 0.01 | 0.04 | 0.03 | **0.92** |
| 2 | 0.05 | **0.85** | 0.07 | 0.03 |
| 3 | **0.74** | 0.14 | 0.09 | 0.03 |
| 4 | **0.81** | 0.02 | 0.13 | 0.04 |

We see that for both sequence logos, the preferred nucleotides from position 1 through 4 are T-C-A-A, as indicated by the values in bold. Plugging in these values into *Equation 5*, we get a Pearson correlation coefficient of $r = 0.997$, indicating substantial agreement between the Sort-Seq and Reg-Seq methods in this example. As a way to visualize similarity, for each position in the sequence logo we can plot the numerical value as resulting from the Sort-Seq experiment ($x_{i,\alpha}$) vs. the

corresponding value obtained from the Reg-Seq experiment ($y_{i,\alpha}$). Perfect correspondence between the methods would result in all the data lying on the $x = y$ line (*Appendix 2—figure 1*).

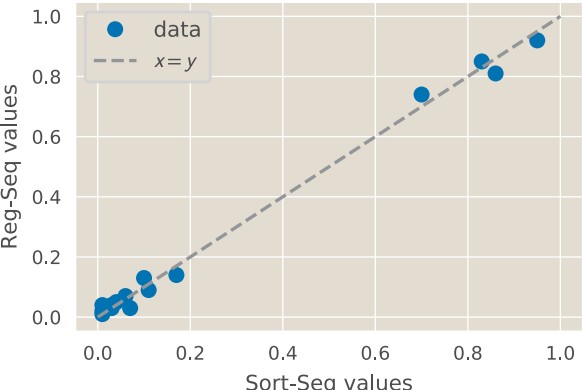

**Appendix 2—figure 1.** Mock data comparing Sort-Seq and Reg-Seq sequence logo values. These data have a Pearson correlation coefficient of $r = 0.997$. This high correlation is also indicated by the data deviating little from the $x = y$ line.

*Figure 3* shows examples of this comparison for four distinct genes of interest. *Figure 3(A)* shows the results of the two methods for the *lacZYA* promoter with special reference to the CRP-binding site. Both the information footprint and the the position weight matrices (displayed with sequence logos) identify the same binding site.

*Figure 3(B)* provides a similar analysis for the *dgoRKADT* promoter where the position weight matrices for the CRP-binding site from Reg-Seq and Sort-Seq have a correlation coefficient of r = 0.90. *Figure 3(C)* provides a quantitative dissection of the *relBE* promoter which is repressed by RelBE. Here, we use both information footprints and expression shifts as a way to quantify the significance of mutations to different binding sites across the promoter. Finally, *Figure 3(D)* shows a comparison of the two methods for the *marRAB* promoter. The two approaches both identify a MarR-binding site.

## False positive and false negative rates

We introduce a systematized way of identifying the locations of binding sites, as shown in *Figure 11*, that allows the false negative and false positive rate of binding site identification to be clearly assessed. For a given information footprint, we average over 15 base pair 'windows'. We then determine which base pairs are part of a regulatory region by setting an information threshold of $2.5 \times 10^{-4}$ bits, which is explained below. All base pair positions that pass the information threshold are then joined into 'regulatory regions', which we consider to be 'activator-like' (if a mutation decreases expression) or 'repressor-like' (if a mutation increases expression). This means that it is possible to identify overlapping repressor- and activator-binding sites. We join any base pair positions within 4 base pairs of each other into a single regulatory region. We then find the edges of each binding site region by trimming off any base pairs at the edge that are below the information threshold (even if the 15 base pair average is above the threshold). A limitation of this method of identification is that is cannot resolve transcription-factor-binding sites that are very close to each other. The primary reasons for this is that putative binding sites will overlap after the smoothing step. While the method could be tuned to avoid treating nearby regions as the same site, many transcription-factor-binding sites will have sections of base pairs within their site where base identity has little to no effect on gene expression. Helix-turn-helix type transcription factors like CRP (whose binding site can be observed in *Figure 3*) are common examples of this phenomenon.

To determine which information threshold to use as a cutoff for a putative binding site, as displayed in *Figure 11*, we selected a training set of genes which included two of our 'gold standard' genes with previously studied binding sites, DgoR (the upstream site from the *dgoR* promoter) and CRP (from the *araAB* promoter), two genes with only RNAP-binding sites, including *hslU* (under heat shock) and *poxB*, and several genes that we classified as inactive, wherein no RNAP-binding sites or other binding sites could be identified. These inactive genes included *hicB, mtgA, eco, hslU* (without

heat shock), and *yncD*. The growth condition (heat shock) is specified for the *hslU* promoter as transcription occurs from a $\sigma^{32}$ RNAP site, which will be inactive except during heat shock. We selected the threshold such that the RNAP sites and known binding sites were identified, while no binding sites were identified in the inactive regions.

We then determine a set of binding sites upon which to test this method and determine a false negative rate for the Reg-Seq experiment. In this set of binding sites, we include those sites which are 'high-evidence' according to EcoCyc. Such 'high evidence' binding sites have been validated experimentally with the binding of purified protein or through site mutation. Some 'high-evidence' sites are excluded because they are not included within our 160 base pair, mutagenized sequence, or because they are not active in any of the growth conditions that we tested. Justifications for those binding sites which were not included are now listed in a new appendix; Appendix 4 Section 'Explanation of included binding sites'. A full list of promoters and binding sites that *were* included in the set of genes used to validate our automated binding-site finding algorithms are also provided in *Appendix 2—table 1*.

**Appendix 2—table 1.** A suite of experimentally validated and high-evidence binding sites used to test our automated binding site finding algorithm.

Specifically, this list of genes was used to test the false negative rate of our Reg-Seq method by examining what fraction of high-evidence sites were also identified with Reg-Seq.

| Gene | Transcription factor | Transcription factor type |
| --- | --- | --- |
| *rspA* | CRP | activator |
| *rspA* | YdfH | repressor |
| *araAB* | AraC (two sites) | activator |
| *znuCB* | Zur | repressor |
| *xylA* | CRP | activator |
| *xylA* | XylR (two sites) | activator |
| *xylF* | XylR (two sites) | activator |
| *dicC* | DicA | repressor |
| *relBE* | RelBE | repressor |
| *ftsK* | LexA | repressor |
| *znuA* | Zur | repressor |
| *lac* | CRP | activator |
| *marR* | Fis | activator |
| *marR* | MarA | activator |
| *marR* | MarR (two sites) | repressor |
| *dgoR* | CRP | activator |
| *dgoR* | DgoR (right site) | repressor |
| *ompR* | IHF (three sites) | repressor |
| *ompR* | CRP | repressor |
| *dicA* | DicA | repressor |
| *araC* | AraC (two sites) | repressor |
| *araC* | AraC (two sites) | activator |
| *araC* | CRP | activator |
| *araC* | XylR (two sites) | repressor |

For each promoter contained in *Appendix 2—table 1*, we used the automated procedure outlined above and in *Figure 11* to identify the activator- and repressor-binding sites. A visual display of the expected binding sites, the information footprints for the promoters in *Appendix 2—table 1*, and the discovered binding sites are all displayed in *Appendix 2—figures 2* and *3*. To assess the

false negative rate, we compare the identified regulatory regions to the known binding sites from *Appendix 2—table 2*. At this stage, we did not consider the identities of the binding sites; we merely consider their presence or absence. Inferred binding sites are declared to 'match' the known binding site if the automated identification procedure classifies at least half of the base pairs reported in EcoCyc as belonging to a transcription-factor-binding site and correctly determines whether the binding site belongs to an activator or repressor.

We do not require exact matching of the edges of the binding sites for several reasons. One such reason is that, in some cases, the sequence of half of a binding site (for example, corresponding to one half of a helix-turn-helix binding motif) can contribute relatively little to gene expression, and so will not have high mutual information values in the corresponding information footprint for that binding site. While this may appear unintuitive for transcription factors where both sections of the binding site are bound by identical halves of a dimer, we see several examples of this in our Reg-Seq experiment results, including for CRP-binding sites of the *rspA* promoter studied during our analysis of false negative rates. We can see in *Appendix 2—figures 2* and *3* that the downstream half of the binding site is not identified as important for gene expression. If we examine the wild type sequence of the *rspA* promoter, we also see that, for the upstream half of the sequence, the wild type matches the five most conserved bases of the consensus sequence (TGTGA) perfectly. The downstream half of the sequence, however, has three mismatches out of five bases. The downstream half of the binding site already binds to its target transcription factor poorly, so further mutations have little effect. While it is true that CRP binds to that sequence region, it is also true that CRP binds only extremely weakly to that section of the region. A similar effect can be seen in previous work from *Belliveau et al., 2018* where a mutation in the downstream half of a CRP-binding site in the *xylE* promoter had more than a 10-fold greater effect on binding energy than mutation in the upstream half of the binding site. As such, we are lenient when evaluating the successes of our algorithm in this regard. Furthermore, the methods that have been used to determine the presence of 'high evidence' binding sites in the past, such as ChIP-Seq, do not typically have base pair resolution with which to precisely determine the edges of binding sites (*Skene and Henikoff, 2015*).

Lastly, a known weakness of our algorithmic approach is that binding sites that are extremely close or overlapping cannot be distinguished from each other initially. For example, the XylR sites in the *xylF* promoter are only separated by three bases according to RegulonDB. While the sites can be distinguished upon later investigation through gene knockouts, mass spectrometry, or motif comparison, our initial algorithm joins the sites into one large site. While this is a weakness of the algorithm, for our purposes it does not constitute a false negative, as the important regions for regulation are still discovered. All regions for all promoters that are classified as regulatory regions, their identities as activators, repressors, or RNAP binding sites, as well as their starting and ending base pairs, can be found in *Supplementary file 3*. Furthermore, we summarize the success and failures of the method at each binding site in *Appendix 2—table 2* below.

**Appendix 2—table 2.** The results of the comparison between experimentally verified, high-evidence binding sites and Reg-Seq-binding sites.
A visual illustration of the comparison can be found in *Appendix 2—figures 2* and *3*.

| Gene | Transcription factor | Was the region classified correctly? |
|------|---------------------|--------------------------------------|
| *rspA* | CRP | Yes |
| *rspA* | YdfH | Yes |
| *araAB* | AraC (two sites) | Yes |
| *znuCB* | Zur | Yes |
| *xylA* | CRP | Yes |
| *xylA* | XylR (two sites) | Yes |
| *xylF* | XylR (two sites) | Yes |
| *dicC* | DicA | Yes |
| *relBE* | RelBE | Yes |
| *ftsK* | LexA | Yes |

*Continued on next page*

*Appendix 2—table 2 continued*

| Gene | Transcription factor | Was the region classified correctly? |
| --- | --- | --- |
| *znuA* | Zur | Yes |
| *lac* | CRP | Yes |
| *marR* | Fis | No |
| *marR* | MarA | Yes |
| *marR* | MarR (two sites) | Yes |
| *dgoR* | CRP | Yes |
| *dgoR* | DgoR (right site) | No |
| *ompR* | IHF (three sites) | Yes |
| *ompR* | CRP | No |
| *dicA* | DicA | No |
| *araC* | AraC (four sites) | one site identified |
| *araC* | CRP | No |
| *araC* | XylR (two sites) | No |

We see in *Appendix 2—table 2* that 11 of the 15 promoter regions included in *Appendix 2—table 1* have all transcription-factor-binding sites classified as putative transcription factors, two have the majority of sites correctly classified, and two do not have any of their binding sites correctly classified as regulatory elements. We can see the information footprints used in the correct identifications in *Appendix 2—figures 2* and *3*. We could alternatively consider that 23 out of 33 binding sites are correctly classified. However, we argue that the false negative rate should be considered on a per promoter basis, rather than on the basis of individual binding sites. The reason for this argument can be seen in the two 'worst' cases of correct binding site identification; namely, for the *araC* and *dicA* promoters.

The *araC* promoter is repressed by multiple repressor-binding sites in all growth conditions tested. *araC* only has high expression transiently after addition of arabinose (*Johnson and Schleif, 1995*), and while growth in arabinose is utilized in this experiment, RNA was not collected during the window of high expression. The case study shows that Reg-Seq does not perform well when many repressor sites regulate the promoter. Reg-Seq relies on mapping the effect on expression of mutating a particular site, and when many strong repressor sites are present, expression change will be minimal unless all repressor sites are weakened through mutation. Additionally, in this highly repressed case, the RNAP-binding site we observe in the mutagenized region is not the documented RNAP site in RegulonDB, indicating that we are seeing transcription primarily from an alternative TSS. Different RNAP sites are often regulated differently, and in this case, the presence of an alternative and dominant RNAP-binding site (in the repressed case), likely contributes to a failure to observe six of the seven binding sites in the *araC* promoter. Similarly, in the *dicA* promoter, we did not find an RNAP-binding site in the studied region, which would make it very unlikely for any transcription-factor-binding sites to be identifiable.

In order to determine false positive rates, we test against promoters for which we are certain there are not additional, unannotated binding sites. Most known binding sites were not determined using a method like Reg-Seq, which looks for regulatory elements across an entire promoter region at base pair resolution. Rather, many efforts to pinpoint transcription-factor-binding site locations use assays like ChIP-Seq, which prioritizes looking for all binding sites of a given transcription factor across the entire genome. For those promoters studied with Reg-Seq, there are five promoters for which we have reason to believe that there are no undiscovered binding sites. There is evidence that the *zupT* promoter is constitutive (*Grass et al., 2005*), and the *marR*, *relBE*, *dgoR*, and *lacZYA* promoters have all been examined for binding sites at base pair resolution previously (in the Sort-Seq experiment [*Belliveau et al., 2018*; *Kinney et al., 2010*]).

To evaluate false positive rates, we examine the putative activator and repressor binding sites as identified using our automated methodology (described previously), and compare any known

binding sites to the known binding sites for the target promoters. We also classify any putative regulatory regions that are outside of known transcription-factor-binding sites as false positives. Similarly, any identified RNAP-binding sites which were outside of the known RNAP binding locations were classified as false positives. In the *zupT* promoter, only the correctly placed RNAP site was identified. There were similarly no false positives identified in the *marR*, *relBE*, *dgoR*, or *lacZYA* promoters.

We additionally compare the energy matrices from putative regulatory regions to known binding site motifs. The known motifs are obtained either from RegulonDB or are generated from data from our prior Sort-Seq experiments (see *Belliveau et al., 2018*). We utilize the TOMTOM motif comparison software from *Gupta et al., 2007* to perform these comparisons. TOMTOM generates a p-value under the null hypothesis that the two compared motifs are drawn independently from the same underlying probability distribution. We test 95 motifs against each target motif that we are attempting to identify. The 95 resulting p-values (for each target) generated by TOMTOM are displayed in *Appendix 2—figure 4*. A full discussion of TOMTOM can be found in Appendix 3 Section 'TOMTOM motif comparison'. We only included those transcription factors that either have over 50 known binding sites in RegulonDB or have experimental measurements of binding site preference, such as in Sort-Seq (*Belliveau et al., 2018*). As such, we used TOMTOM on the XylR, CRP, MarA, MarR, and RelBE sites in *Appendix 2—table 2*. We utilized a p-value cutoff of 0.05, corrected for multiple hypothesis testing. 95 motifs were tested against each target, and using the Bonferroni correction leads to a p-value cutoff of $\frac{0.05}{95} = 5 \times 10^{-4}$. In *Appendix 2—figure 4* we show that the correct transcription factor falls below the p-value threshold in all cases. For the CRP-binding site in the *lacZYA* promoter, FNR also falls below the cutoff, but CRP has a calculated p-value that is $\approx$ 6 orders of magnitude lower, and so is clearly identified as the correct binding site. The results show that motif comparisons can be used reliably in those cases where we have high-quality energy matrices for comparison.

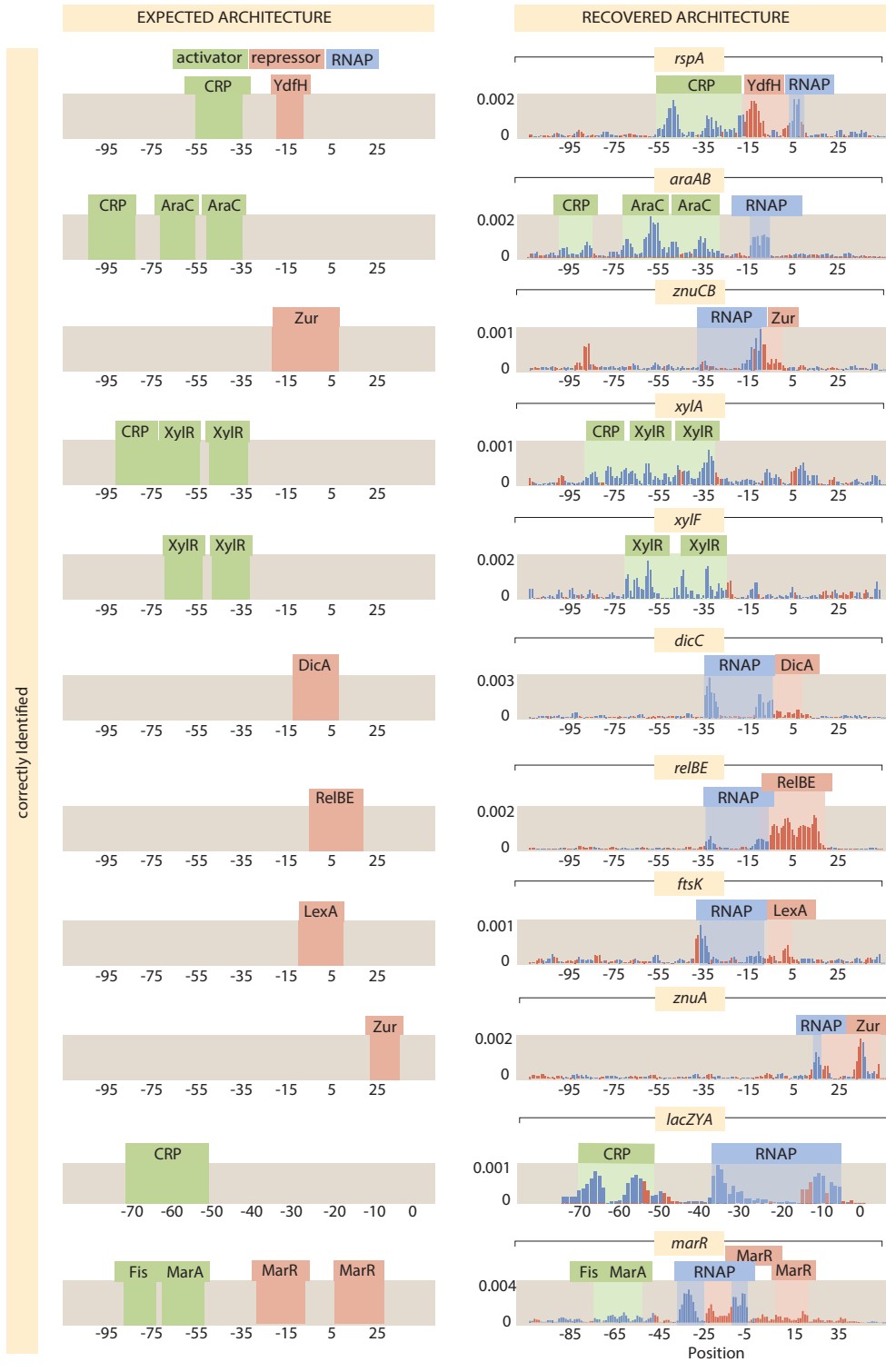

**Appendix 2—figure 2.** A visual comparison of the literature binding sites (left panel) and the extent of the binding sites discovered by our algorithmic approach (right panel). RNAP-binding sites are also labeled in the right panel, but RNAP-binding sites are not included in the false positive analysis. Numeric values for the displayed data can be found in *Appendix 2—figure 2—source data 1*.

The online version of this article includes the following source data is available for figure 2:

**Appendix 2—figure 2—source data 1.** Data for information footprints and identified regions in *Appendix 2—figure 2*.

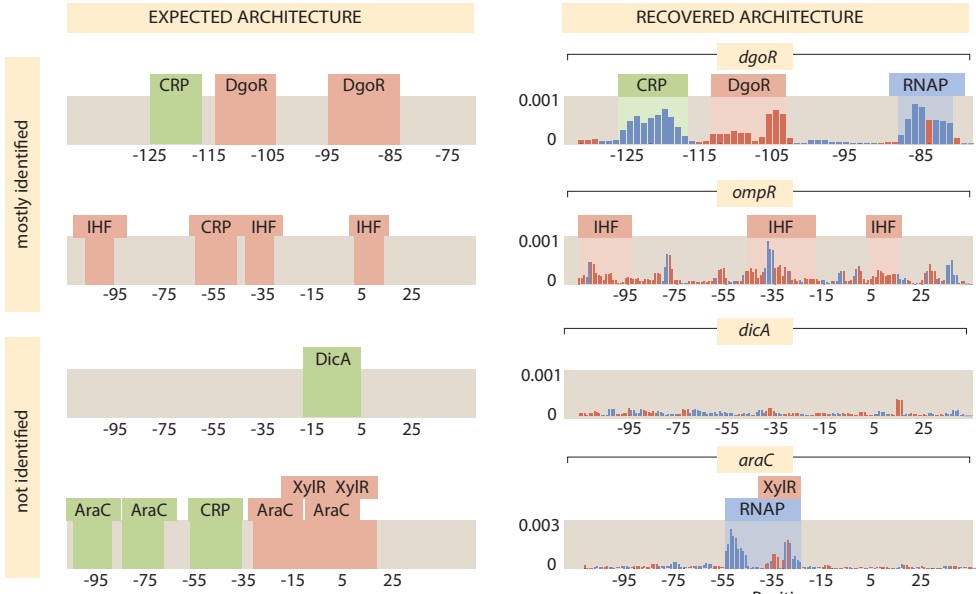

**Appendix 2—figure 3.** A continuation of the visual comparison of the literature binding sites (left panel) and the binding sites discovered by our algorithmic approach (right panel) begun in *Appendix 2—figure 2*.

The online version of this article includes the following source data is available for figure 3:

**Appendix 2—figure 3—source data 1.** Data for information footprints and identified regions in *Appendix 2—figure 3*.

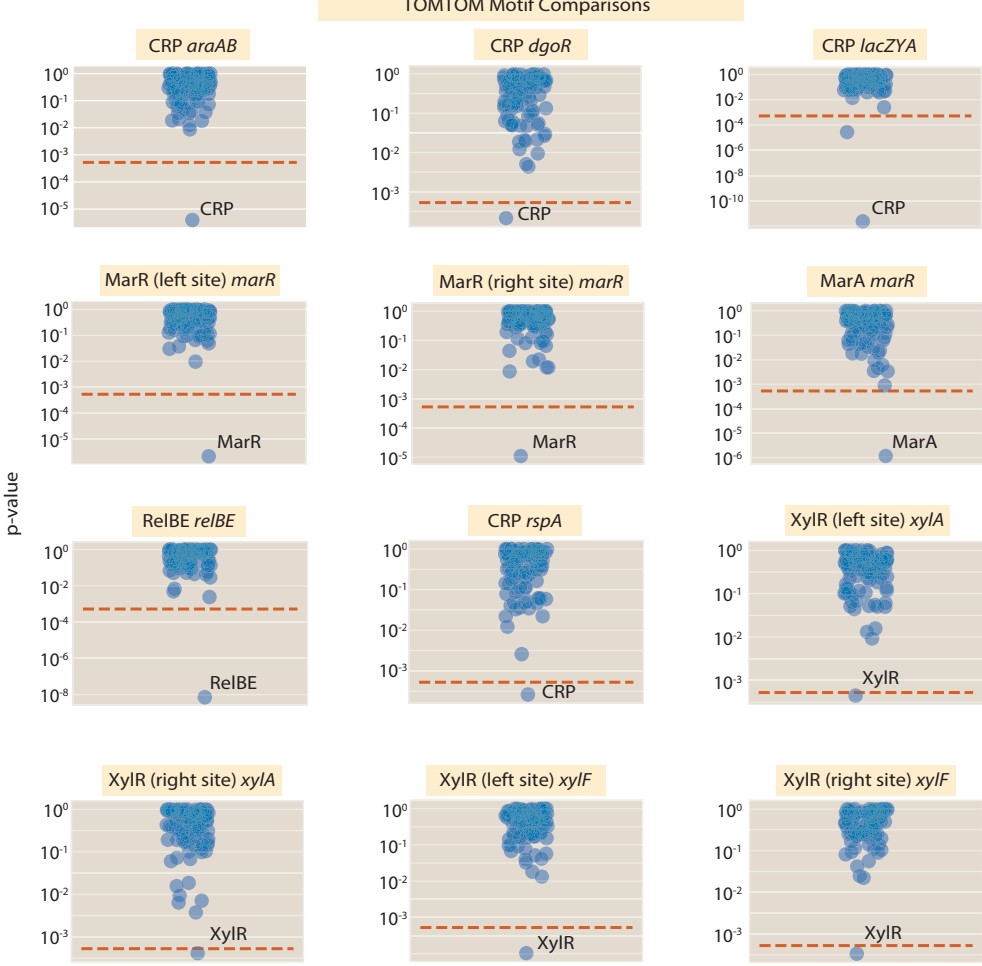

**Appendix 2—figure 4.** A visual display of the results of the TOMTOM motif comparison between the discovered binding sites and known sequence motifs from RegulonDB and our prior Sort-Seq experiment (*Belliveau et al., 2018*). Each dot in a given panel represents a comparison between the target position weight matrix (given in the figure title) and a position weight matrix for a given transcription factor. The p-value is calculated using the null hypothesis, that both motifs are drawn independently from the same underlying probability distribution. The red dotted line is displayed at a p-value of $5 \times 10^{-4}$. The line represents a p-value threshold of 0.05 that has been corrected for multiple hypothesis testing using the Bonferroni correction (95 motifs were compared against the target for a p-value threshold of $\frac{0.05}{95} = 5 \times 10^{-4}$). Numeric values for the displayed data can be found in *Appendix 2—figure 4-source data 1*.

The online version of this article includes the following source data is available for figure 4:

**Appendix 2—figure 4—source data 1.** All p-values displayed in *Appendix 2—figure 4*.

## Appendix 3

### Extended details of analysis methods

Information footprints

We favor the use of information footprints as a tool for hypothesis generation to identify regions which may contain transcription-factor-binding sites. In general, a mutation within a transcription factor site is likely to weaken that site. We look for groups of positions where mutation away from wild type has a large effect on gene expression. Our datasets consist of nucleotide sequences, the number of times we sequenced a given, specific mutated promoter in the plasmid library, and the number of times we sequenced its corresponding mRNA. A simplified illustrative dataset on a hypothetical 4 nucleotide sequence is shown in *Appendix 3—table 1*.

*Appendix 3—table 1.* Example dataset of four nucleotide sequences, and the corresponding counts from the plasmid library and mRNAs.

| Sequence | Library sequencing counts | mRNA counts |
| --- | --- | --- |
| ACTA | 5 | 23 |
| ATTA | 5 | 3 |
| CCTG | 11 | 11 |
| TAGA | 12 | 3 |
| GTGC | 2 | 0 |
| CACA | 8 | 7 |
| AGGC | 7 | 3 |

One strategy to measure the impact of a given mutation on expression is to take all sequences which have base $b$ at position $i$ and determine the number of mRNAs produced per read in the sequencing library. By comparing the values for different bases we can determine how large of an effect a mutation has on gene expression. For example in *Appendix 3—table 1*, for the second position ($i = 2$) those sequences that contain the wild type base A ($b = \text{A}$) have 20 sequencing counts out of 50 ($23 + 3 + 11 + 3 + 0 + 7 + 3 = 50$) from the DNA library and 10 sequencing counts from the 50 ($5 + 5 + 11 + 12 + 2 + 8 + 7 = 50$) mRNA reads. For all other sequences ($b = \text{C}, \text{G}$, or T), there are 30 sequencing counts from the DNA library and 40 sequencing counts from mRNA. A measure of the effect of mutation on expression would be to compare the ratios $\frac{\text{mRNA counts}/\text{total mRNA counts}}{\text{library counts}/\text{total library counts}}$ between mutated and wild-type sequences. For the data in *Appendix 3—table 1*, sequences with a wild type base at position 2 will have a ratio of purple $(10/50)/(20/50) = 0.5$ and sequences with a mutated base at position 2 will have a ratio of $(40/50)/(30/50) \approx 1.3$.

While directly comparing ratios is one way to measure the effect on gene expression, we use mutual information to quantify the effect of mutation, as *Kinney et al., 2010* demonstrated could be done successfully. In *Appendix 3—table 1*, the frequency of the nucleotide A in the DNA library at position 2 is 0.4, as 20 out of 50 sequencing counts have an A at position 2. Similarly, the other frequencies at position 2 are 0.32 for C, 0.14 for G and 0.14 for T. In the observed mRNA sequence counts, we find C at 34 of of 50 total mRNA counts, which gives a frequency of 0.68, indicating that Cytosine is enriched in the mRNA transcripts compared to the DNA library. The frequencies for the other bases are 0.2 for A, 0.06 for T and 0.06 for G. Large enrichment of a base compared to others in mRNA sequencing counts occurs when base identity is important for gene expression.

We are classifying bases as either wild type ($m = 0$) or mutated ($m = 1$). A discussion of this assumption can be found at the end of this section. We compute mutual information at position $i$ as

$$I_i = \sum_{m=0}^{1} \sum_{\mu=0}^{1} p(m,\mu) \log_2 \left( \frac{p(m,\mu)}{p_{mut}(m)p_{expr}(\mu)} \right), \tag{6}$$

where $p_{expr}(\mu)$ is the ratio of the number of DNA ($\mu = 0$) or mRNA ($\mu = 1$) sequencing counts to the total number of counts,

$$p_{expr}(\mu) = \begin{cases} \sum (\text{mRNA counts})/(\text{total counts}) & \text{if } \mu=1 \\ \sum (\text{Library Sequencing counts})/(\text{total counts}), & \text{if } \mu=0. \end{cases} \tag{7}$$

From the example data in **Appendix 3—table 1** we can calculate $p_{expr}(\mu)$. To do so, we sum up DNA counts and mRNA counts from all sequences and divide by the total number of counts $(50 + 50 = 100)$ to obtain

$$p_{expr}(\mu) = \begin{cases} 0.5, & \text{if } \mu=1 \\ 0.5, & \text{if } \mu=0. \end{cases} \tag{8}$$

In addition, $p_{mut}(m)$ is the fraction of the total counts that either have a mutation ($m = 1$) at the given position or the fraction that have a wild-type base ($m = 0$) at the position. $p_{mut}$ has to be computed for each position individually. For position 1, the wild type base is an A, and we see that there are a total of 100 sequencing counts, of which 46 counts (DNA and mRNA combined) contain an A at position 1. Therefore, $p(m)$ can be calculated for position 1 as

$$p_{mut}(m) = \begin{cases} 0.46, & \text{if } m=0 \\ 0.54, & \text{if } m=1. \end{cases} \tag{9}$$

Lastly, the joint distribution $p(m, \mu)$ is the probability that a given sequencing read in the dataset will have expression level $\mu$ and mutation status $m$. $p(m, \mu)$ is calculated by dividing the number of sequencing reads at the chosen position with mutation status $m$ and expression status $\mu$ by the total number of sequencing reads. In the case of the example dataset in **Appendix 3—table 1** and for $m = 0$ and $\mu = 0$, we sum the sequencing reads that are wild type at position 1 and also are in the DNA library. As there are 17 sequences that fit the criteria out of 100 total sequences, $p(m = 0, \mu = 0) = 0.17$. The other values of $p(m, \mu)$ can be calculated to be

$$p(m, \mu) = \begin{cases} 0.17, & \text{if } m=0 \text{ (wild type base) and } \mu=0 \text{ (DNA)} \\ 0.21, & \text{if } m=1 \text{ (mutated base) and } \mu=1 \text{ (RNA)} \\ 0.33, & \text{if } m=1 \text{ and } \mu=0 \\ 0.29, & \text{if } m=0 \text{ and } \mu=1. \end{cases} \tag{10}$$

The marginal distributions $p_{expr}$ and $p_{mut}$ can be obtained by summing over one of the two variables, that is,

$$p_{expr}(\mu) = \sum_m p(m, \mu), \tag{11}$$

$$p_{mut}(m) = \sum_\mu p(m, \mu). \tag{12}$$

Plugging the values calculated above into **Equation (6)** yields a mutual information value of 0.06 bits at position 1. The unit is bits because the mutual information is computed with a logarithm of base 2. Other bases can be chosen, however, that results in different units for the mutual information.

Mutual information is a measurement that quantifies how much the measurement of one of two variables reduces uncertainty of the other variable. For example, very low mutual information means that by knowing one variable one gains no information about the other variable, while on the other hand high mutual information means that by knowing one variable our knowledge about the others increases. At a position where base identity matters little for expression level, there would be little difference in the frequency distributions for the library and mRNA transcripts. The entropy of the distribution would decrease only by a small amount when considering the two types of sequencing reads separately.

We seek to determine the effect on gene expression of mutating a given base. However, if mutation rates at each position are not fully independent such that $p(m_i, m_{i'}) \neq p(m_i)p(m_{i'})$, then the information value calculated in **Equation (6)** will also encode the effect of mutation at correlated

positions. For instance, if position $i$ is part of an activator-binding site, mutating it will have a large effect on gene expression. If position $i'$ is not within the activator site, then mutating position $i'$ will have minimal true effect on gene expression. However, if mutations at the two bases are correlated, mutating position $i'$ will make it more likely for $i$, and therefore the activator-binding site, to be mutated. Knowledge that $i'$ is mutated is predictive of overall expression, and so position $i'$ will have high mutual information according to *Equation (6)*, even though that position has no regulatory function. In our experiment we designed sequences to be synthesized such that each position had a probability of mutation that was independent of mutation at any other position. However, due to errors in the oligonucleotide synthesis process, additional mutations in the ordered sequences were introduced. Sequencing our DNA libraries reveals that mutation at a given base pair can make mutation at another base pair more likely by up to 10%, where neighboring base pairs are the most likely to have correlations between mutations. This is enough to cloud the signature of most transcription factors in an information footprint calculated using *Equation (6)*.

We need to determine values for $p_i(m|\mu)$ when mutations are independent, and to do this we need to fit these quantities from our data. We assert that

$$\langle C_{\mathrm{mRNA}} \rangle \propto e^{-\beta E_{eff}} \tag{13}$$

is a reasonable approximation to make, which we will justify by considering a number of possible regulatory scenarios. $\langle C_{\mathrm{mRNA}} \rangle$ is the average number of mRNAs produced and $E_{eff}$ is an effective binding energy for the sequence that can be determined by summing contributions from each position in the sequence independently. There are many possible underlying regulatory architectures, and those that have been discovered with Reg-Seq are summarized in *Table 1*. While we will show that under reasonable assumptions this approach is useful for any of these regulatory architectures, let us first consider the simple case where there is only an RNAP site in the region under study. We can write down an expression for average gene expression per cell as

$$\langle C_{\mathrm{mRNA}} \rangle \propto p_{bound} \propto \frac{\frac{P}{N_{NS}} e^{-\beta E_P}}{1 + \frac{P}{N_{NS}} e^{-\beta E_P}}, \tag{14}$$

where $p_{bound}$ is the probability that the RNAP is bound to DNA and is known to be proportional to gene expression in *E. coli* (*Ackers et al., 1982*; *Buchler et al., 2003*; *Garcia and Phillips, 2011*), $E_P$ is the energy of RNAP binding, $N_{NS}$ is the number of nonspecific DNA binding sites, and $P$ is the number of RNAP. If RNAP binds weakly then $\frac{P}{N_{NS}} e^{-\beta E_P} \ll 1$, and we can simplify *Equation (14)* to

$$\langle C_{\mathrm{mRNA}} \rangle \propto e^{-\beta E_P}. \tag{15}$$

Using this relation, we can compute the ratio of average mRNA counts in wild type $\langle C_{\mathrm{mRNA}}^{\mathrm{WT}_i} \rangle$ to average mRNA counts in a mutant $\langle C_{\mathrm{mRNA}}^{\mathrm{Mut}_i} \rangle$ as

$$\frac{\langle C_{\mathrm{mRNA}}^{\mathrm{WT}_i} \rangle}{\langle C_{\mathrm{mRNA}}^{\mathrm{Mut}_i} \rangle} = \frac{e^{-\beta E_{P_{\mathrm{WT}_i}}}}{e^{-\beta E_{P_{\mathrm{Mut}_i}}}}, \tag{16}$$

$$\frac{\langle C_{\mathrm{mRNA}}^{\mathrm{WT}_i} \rangle}{\langle C_{\mathrm{mRNA}}^{\mathrm{Mut}_i} \rangle} = e^{-\beta \left( E_{P_{\mathrm{WT}_i}} - E_{P_{\mathrm{Mut}_i}} \right)}, \tag{17}$$

where $E_{P_{\mathrm{WT}_i}}$ is the binding energy of RNAP to the wild-type binding site and $E_{P_{\mathrm{Mut}_i}}$ is the binding energy of RNAP to the mutant-binding site. Using the assumption that each position contributes independently to the binding energy, we can simplify the differences in energies to $E_{P_{\mathrm{WT}_i}} - E_{P_{\mathrm{Mut}_i}} = \Delta E_{P_i}$. We can now calculate the probability of finding a specific base in the expressed sequences. If the probability of finding a wild type base at position $i$ in the DNA library is $p_i(m=0|\mu=0)$, then

$$p_i(m=0|\mu=1) = \frac{p_i(m=0|\mu=0)\frac{\langle C_{\mathrm{mRNA}}^{\mathrm{WT}_i}\rangle}{\langle C_{\mathrm{mRNA}}^{\mathrm{Mut}_i}\rangle}}{p_i(m=1|\mu=0) + p_i(m=0|\mu=0)\frac{\langle C_{\mathrm{mRNA}}^{\mathrm{WT}_i}\rangle}{\langle C_{\mathrm{mRNA}}^{\mathrm{Mut}_i}\rangle}}, \tag{18}$$

$$p_i(m=0|\mu=1) = \frac{p_i(m=0|\mu=0)e^{-\beta \Delta E_{P_i}}}{p_i(m=1|\mu=0) + p_i(m=0|\mu=0)e^{-\beta \Delta E_{P_i}}}. \tag{19}$$

Under certain conditions, we can also infer a value for $p_i(m|\mu=1)$ using a linear model when there are any number of activator or repressor-binding sites. We will demonstrate this in the case of a single activator and a single repressor, although a similar analysis can be done when there are greater numbers of transcription factors. Define $p = \frac{P}{N_{NS}}e^{-\beta E_P}$ and $a = \frac{A}{N_{NS}}e^{-\beta E_A}$ where $A$ is the number of activators, and $E_A$ is the binding energy of the activator. Also define $r = \frac{R}{N_{NS}}e^{-\beta E_R}$ where $R$ is the number of repressors and $E_R$ is the binding energy of the repressor. Then we can compute the average number of produced mRNA as

$$\langle C_{\mathrm{mRNA}}\rangle \propto p_{bound} \propto \frac{p + pae^{-\beta \epsilon_{AP}}}{1 + a + p + r + pae^{-\beta \epsilon_{AP}}}, \tag{20}$$

where $\epsilon_{AP}$ is the interaction energy of activators and the RNAP. One assumption we make is that activators and RNAP bind weakly to their binding sites ($a<<1$ and $p<<1$) but interact strongly ($pae^{-\beta \epsilon_{AP}}>>p$). Under this assumption, RNAP and associated activators are much more likely to bind DNA as a unit than separately. The binding energy measurements by *Forcier et al., 2018* support this assumption in the case of CRP in the *lac* operon. The DNA-protein binding energy of CRP is measured to be -3.18 $k_BT$ and the interaction energy between CRP and RNAP is measured to be $\epsilon_{AP} = -6.56 k_BT$. The copy number of CRP is $A \approx 4000$ (*Schmidt et al., 2016*), the copy number of RNAP is $P \approx 2000$ in slowly growing cells (*Bremer and Dennis, 1996*), and the RNAP binding energy for the wild type *lac* promoter is $E_P \approx -5.2\, k_BT$ (*Brewster et al., 2012*). As $N_{NS} \approx 4.6 \times 10^6$, the value of $a$ can be calculated to be $a \approx \frac{4000}{4.6 \times 10^6}e^{3.18} \approx 0.02$. Similarly $p$ can be calculated to be $p \approx \frac{2000}{4.6 \times 10^6}e^{5.2} \approx 0.08$. Lastly, we can calculate $pae^{-\beta \epsilon_{AP}} \approx pae^{6.56} \approx 1$. We can see that these numbers satisfy the assumptions $a<<1$, $p<<1$, and $pae^{-\epsilon_{AP}}>>p$. We can simplify *Equation (20)* to

$$\langle C_{\mathrm{mRNA}}\rangle \propto p_{bound} \propto \frac{pae^{-\beta \epsilon_{AP}}}{1 + r + pae^{-\beta \epsilon_{AP}}}. \tag{21}$$

The last assumption we make is that repressors bind very strongly ($r>>1$ and $r>>pae^{-\epsilon_{AP}}$). To justify this assumption, we once again look to the lac operon. Wild-type LacI copy number is $R \approx 10$ and the wild-type binding energy for the O1 operator is $E_R \approx -16 k_BT$ (*Garcia and Phillips, 2011*). We can use these values to compute $r \approx \frac{10}{4.6 \times 10^6}e^{16} \approx 20$. We can simplify *Equation (21)* to

$$\langle C_{\mathrm{mRNA}}\rangle \propto \frac{pae^{-\beta \epsilon_{AP}}}{r} \tag{22}$$

$$\langle C_{\mathrm{mRNA}}\rangle \propto e^{-\beta(-E_P - E_A + E_R)} \tag{23}$$

As we typically assume that RNAP binding energy, activator binding energy, and repressor binding can all be represented as sums of contributions from their constituent bases, the combination of the energies can be written as a total effective energy $E_{eff}$ which is a sum of independent contributions from all positions within the binding sites.

We fit the parameters for each base using Markov Chain Monte Carlo Method (MCMC). Two MCMC runs are conducted using randomly generated initial conditions. We require both chains to reach sufficiently similar distributions to prove the convergence of the chains or we repeat the runs. During the analysis, we artificially treat mutation rates at all positions as equal, as we do not wish for mutation rate to play a role in mutual information calculations. The information values are smoothed by averaging with neighboring values.

By only considering wild type or mutated energy contributions to the total effective binding energy rather than having separate values for energy contributions from all four base pairs, our methods will not be accurate in the case of calculating mutual information at locations with degenerate base pairs. However, the information footprints are intended to be hypothesis generation tools that can identify transcription-factor-binding sites. As such, the most important test for the assumption that we can approximate effective energy contributions from all 4 bases as contributions from only wild type or mutated bases is to assess whether the approximation has any effect on determining binding site locations. We re-ran the false positive and false negative assessments discussed in Appendix 2 Section 'False positive and false negative rates', but instead calculated the effective energy parameters for producing information footprints as a sum of contributions from all four bases. We find that the literature binding sites that were properly identified, as summarized in *Appendix 2—table 2*, are identically identified. Specifically, any site which was identified using the previous method is still identified and any site that failed to be identified is still not observed. Similarly, when we only fit effective energy parameters for mutated or wild type bases there are no false positives identified in the promoters for *marR*, *relBE*, *dgoR*, *zupT*, or *lacZYA*. There are also no false positives when repeating the procedure while considering all 4 bases in the effective energy fits, implying that the simplification to only considering mutated or wild type bases does not have an effect on our ability to identify binding sites.

## Processing of mass spectrometry experiments

Mass spectrometry results were processed using MaxQuant (*Cox and Mann, 2008*; *Cox et al., 2009*). Spectra were searched against the UniProt *E. coli* K-12 database as well as a contaminant database (256 sequences). LysC was specified as the digestion enzyme. Proteins were considered if they were known to be transcription factors, or were predicted to bind DNA (using gene ontology term GO:0003677, for DNA-binding in BioCyc).

## Uncertainty due to number of independent sequences

1400 promoter variants were ordered from TWIST Bioscience for each promoter studied. Due to errors in oligonucleotide synthesis, additional mutations are randomly introduced into the ordered oligos. We have found that, as a result of these random, additional errors, the final number of variants received was an average of 2200 per promoter.

To demonstrate that our results are not strongly dependent on the number of sequences in each promoter library, and also to assess how a reduction in the number of sequences per promoter library could facilitate larger scale experiments in the future, we generated examples of smaller data sets by computationally sub-sampling the Reg-Seq experimental data from seven mutated promoter libraries (*maoP*, *hslU*, *rpsA*, *leuABCD*, *aphA*, *araC*, and *tig*). These promoters are representative of a large cross-section of the variety of regulation we see in our study, as they include promoters with constitutive expression (*hslU*), simple repression(*leuABCD*, *tig*), simple activation (*aphA*), as well as more complicated regulatory architectures (*maoP*, *rspA*, *araC*). Each sub-sampling was done three times, and we then use the Pearson correlation coefficient (Appendix 2 Section 'Comparison between Reg-Seq by RNA-Seq and fluorescent sorting') as a comparison metric between the inference based on the full data set and the computationally sub-sampled data sets.

Based on our analysis, the results of which are displayed in *Appendix 3—figure 1*, we find that there is only a small effect on the resulting sequence logo until the library has been reduced to approximately 500 promoter variants. We could, therefore, reasonably lower the resolution of the experiment to approximately 1000 or fewer unique sequences before large deviations in the inference are experienced. Decreasing the number of unique sequences can give modest boosts to the number of genes that can be studied, but will not be able to give order of magnitude increases in the number of genes that can be explored.

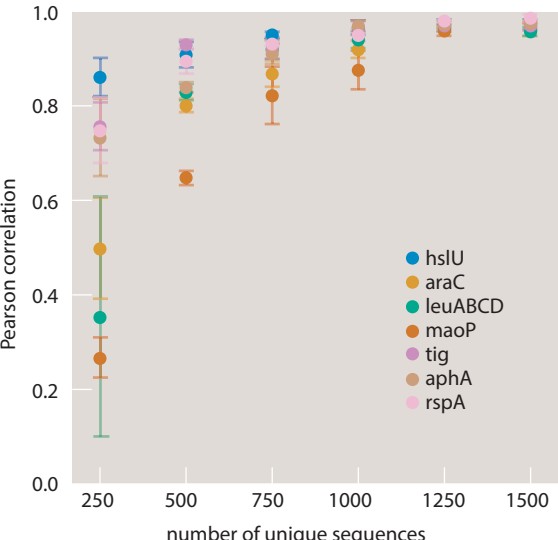

**Appendix 3—figure 1.** Pearson correlation as a function of the number of unique DNA sequences (as explained in Appendix 2 Section 'Comparison between Reg-Seq by RNA-Seq and 2uorescent sorting'). For seven different genes, we studied how the number of mutated DNA sequences affects the reproducibility of our MCMC inference models. As the number of unique sequences increases, so too does the Pearson correlation value, approaching 1.0. Numeric values for the displayed data can be found in *Appendix 3—figure 1—source data 1*.

The online version of this article includes the following source data is available for figure 1:

**Appendix 3—figure 1—source data 1.** Pearson correlation values for *Appendix 3—figure 1*.

## Effect on calculated energy matrices when transcription factor copy number ≈ plasmid copy number

Throughout this study, we utilize plasmids to express GFP from mutated promoters, and then use the ratio of mRNA/DNA, based on sequencing results, to handle the effect of variability in plasmid copy number between cells. It is necessary, however, to consider the situation wherein the plasmid copy number is of a similar magnitude to the transcription factor copy number, and whether this can impact the calculated energy matrices and binding energies. The genetic expression levels are determined not only by the binding energy, but also by the transcription factor availability, and so it is necessary to consider whether, for those cases where transcription factor copy number ≈ plasmid copy number, there is a corresponding under-estimation of binding energies. Prior work from our laboratory was precisely aimed at rigorously predicting and measuring this effect (*Weinert et al., 2014*). In that study, we demonstrated how to control this effect, wherein transcription factor copy number ≈ plasmid copy number, in a parameter-free manner. However, to mitigate this effect in future studies, we plan to use genome-integrated libraries, rather than plasmid-based expression.

The plasmid used in our experiments is derived from pUA66, which contains a pSC101 origin of replication (*Zaslaver et al., 2006*). The copy number of plasmids with a pSC101 origin is, in log phase, approximately 3 or 4 (*Lutz and Bujard, 1997*). We have not independently assessed the copy number of the plasmid used in this study.

The absolute copy number of thousands of proteins in *E. coli* have been determined using whole-proteome LC-MS. Specifically, a 2016 study that provides the absolute quantification for roughly 55 percent of predicted proteins in the *E. coli* K12 proteome (see Supplementary Table S6) (*Schmidt et al., 2016*). For those transcription factors that were quantified in that study, and also show up in our Reg-Seq experiments, we provide their absolute quantification in *E. coli* K12 for both glucose and LB growth media in *Appendix 3—table 2*.

**Appendix 3—table 2.** Global, absolute quantification for most transcription factors identified in this study, as determined for *E. coli* K12 grown in both glucose (5 g/L concentration in M9 minimal media) and LB medias.

The values in this table are reprinted from *Schmidt et al., 2016* Supplemental Table S6.

| Transcription factor name | Glucose | LB |
|---|---|---|
| FNR | 609 | 1101 |
| YieP | 158 | 261 |
| YciT | 82 | 104 |
| NsrR | 872 | 136 |
| LexA | 560 | 1027 |
| DeoR | 26 | 34 |
| CRP | 2048 | 3450 |
| YdfH | 96 | 154 |
| ArcA | 3367 | 5464 |
| Zur | 70 | 130 |
| GlpR | 75 | 145 |
| PhoP | 2967 | 3132 |
| HNS | 22541 | 47133 |
| StpA | 6863 | 5241 |
| DicA | 20 | 25 |
| YgbI | 2 | 6 |
| XylR | 1 | 8 |

For most transcription factors, the copy number as determined by LC-MS is much greater than the expected, low copy number of the plasmid used in this study, thus mitigating the concern that the limited availability of a transcription factor could impact gene expression.

There are a few transcription factors that have copy number on the order of the plasmid copy number, however, including XylR, DicA, and YgbI. Prior work from our group (**Weinert et al., 2014**) has explored how gene expression behaves in the regime where transcription factor copy number is ≈ plasmid copy number. Here, we will discuss the case of simple repression to demonstrate how the relationship between transcription factor and plasmid copy number could impact our results. The standard thermodynamic model for gene expression under simple repression with a weak promoter, as described by **Bintu et al., 2005**, is

$$C \propto p_{bound} = \frac{\frac{P}{N_{NS}} e^{-\beta \Delta \varepsilon_P}}{1 + \frac{R}{N_{NS}} e^{-\beta \Delta \varepsilon_R}}, \tag{24}$$

where $C$ is a measurement for gene expression level, $N_{NS}$ is the number of nonspecific DNA-binding sites, $P$ is the number of RNAP, and $R$ is the number of repressors. $\Delta \varepsilon_R$ and $\Delta \varepsilon_P$ represent the difference in the repressor-binding energy and RNAP-binding energy between the specific binding site and the averaged nonspecific genomic background respectively. **Weinert et al., 2014** demonstrated experimentally that, in the presence of multiple target binding sites, such as from a multi-copy plasmid, the gene expression level can be described by a very similar functional form to **Equation (24)**, namely,

$$C \propto p_{bound} = \frac{\lambda_P e^{-\beta \Delta \varepsilon_P}}{1 + \lambda_R e^{-\beta \Delta \varepsilon_R}}, \tag{25}$$

where $\lambda_P$ and $\lambda_R$ are the fugacity of RNAP and the repressor and describe the relative availability of RNAP or repressor as a function of plasmid copy number, transcription factor copy number, and binding site strength. The presence of additional plasmid copies does weaken the effect of repressor binding when the repressor copy number is ≈ plasmid copy number. Thus, our information footprint calculations will be affected and the information signature of binding sites such as YgbI, DicA, or XylR will be decreased.

For transcription-factor-binding site interactions that are sufficiently weak, together with a low transcription factor copy number, the effect of having multiple plasmids expressed in a cell could

cause us to have a false negative, and thus miss the presence of a binding site. However, the Reg-Seq method does not claim to capture every regulatory feature for a given promoter, as the activity of some transcription factors is induced only in certain growth conditions, we use a finite, 160 bp mutation window that may miss 'regulation at a distance', and the presence of extremely weak and nonspecific binding sites may cause Reg-Seq to 'miss' some transcription factors (indeed, for the *bdcR* promoter, the GlaR-binding site is outside of the mutagenized region and so is not observed). The effect of additional plasmids within the cellular confines will always decrease the fugacity in *Equation (25)*, as an increase in the number of sites competing for a limited pool of transcription factors will decrease the relative availability of those transcription factors. As a result, the effect on gene expression of a given transcription factor will always lessen in the presence of additional plasmids. This means that, while multi-copy plasmids can introduce false negatives into Reg-Seq, it will not introduce false positives. Additionally, we see empirically that, even for the lowest copy transcription factor for which we have a measurement, XylR ($\approx$ 1 copy per cell), we can identify its transcription-factor-binding site. In *Appendix 2—figures 2* and *3*, 2 (previously known) XylR sites are identified for the *xylA* promoter, and 2 (previously known) XylR sites are identified in the *xylF* promoter.

Finally, the energy matrices, which are a quantitative output of the Reg-Seq experiment, will be unaffected by the presence of multi-copy plasmids. As discussed in Appendix 3 Section 'Energy matrix inference', energy matrix inference relies on calculating the mutual information between the energy predictions of the model and the experimental data. Mutual information is invariant under transformations to the input variables that do not affect their rank order. While the presence of multiple plasmid copies will affect the fugacity in *Equation (25)*, and so will also affect any quantitative prediction of gene expression, a weaker repressor-binding site will still be predicted to have higher gene expression than a stronger repressor-binding site, regardless of the relative availability of the transcription factor. The rank-order is always preserved and so the presence of a multi-copy plasmid will not change the mutual information between model predictions and experimental data. As a result, the final inference of energy matrices will remain the same.

## Energy matrix inference

Energy matrices in this experiment are of the form shown in *Appendix 3—table 3*,

**Appendix 3—table 3.** Example energy matrix.
This matrix is in arbitrary units, and the process to obtain absolute units (in $k_B T$) is described in Appendix 3 Section 'Inference of scaling factors for energy matrices'.

| Pos | A | C | G | T |
|-----|---------|---------|---------|---------|
| 0 | −0.01 | −0.01 | −0.01 | 0.03 |
| 1 | 0.002 | 0.05 | −0.06 | 0.008 |
| 2 | −0.0002 | −0.04 | 0.008 | 0.03 |
| 3 | −0.02 | 0.02 | −0.01 | 0.01 |

where each entry gives the energy contribution from a base pair at a given location. As an example from *Appendix 3—table 3*, an A at position 1 would give a total energy contribution of −0.01 (A. U.). All energy matrices used in our analysis are linear energy matrices, where the total energy is the sum of contributions from each base pair. As a result, total binding energy is

$$\text{binding energy} = \sum_{i=1}^{L}\sum_{j=A}^{T} \theta_{ij} \cdot \delta_{ij}, \tag{26}$$

where $\delta_{ij}$ is the Kronecker delta, which takes on a value of 1 if the base at position $i$ is equal to $j$ and is 0 otherwise, $L$ is the length of the binding site, and $\theta_{ij}$ is the energy contribution of nucleotide $j$ and position $i$ in arbitrary units. To infer the parameters $\theta_{ij}$ in *Equation (26)* from the experimental data, we perform Bayesian inference using a MCMC method, which requires us to calculate the

likelihood of the model given the experimental data. The likelihood function is difficult to determine, but *Kinney et al., 2010* found that, given a large amount of data, the likelihood function is related to the mutual information between energy predictions and data by the equation

$$L(D|\theta) \propto 2^{NI(\mu,E)},\tag{27}$$

where $N$ is the total number of independent sequences, $D$ is the data consisting of sequences and measured sequencing counts, $I$ is the mutual information between gene expression label $\mu$ and energy predictions $E$. $\mu$ is a discrete variable that characterizes the gene expression level as described in *Equation (3)* in the main text. We can calculate mutual information using the formula for mutual information between a continuous and a discrete variable, namely,

$$I(\mu,E) = \int_{-\infty}^{\infty} dE \sum_{\mu=0}^{1} p(\mu,E) \log_2 \left( \frac{p(\mu,E)}{p(E)p(\mu)} \right).\tag{28}$$

While *Equation (28)* is used for continuous energy predictions, there are only $N$ total sequences, and so only $N$ discrete energy predictions. For a simple example of calculating the joint probability distribution $p(\mu,E)$, consider the hypothetical dataset of only four nucleotides in *Appendix 3—table 1*. We first predict the binding energy of each of the example sequences, shown in *Appendix 3—table 4*.

**Appendix 3—table 4.** Example dataset with energy predictions.
Energy predictions are made by applying the example energy matrix in *Appendix 3—table 3* to the example dataset in *Appendix 3—table 1* according to *Equation (26)*.

| $\mu = 0$ | $\mu = 1$ | Energy ($k_B T$ |
|---|---|---|
| 5 | 23 | 0.05 |
| 5 | 3 | 0.008 |
| 11 | 11 | 0.09 |
| 12 | 3 | −0.03 |
| 2 | 0 | 0.03 |
| 8 | 7 | −0.07 |
| 7 | 3 | −0.04 |

We use kernel density estimation with kernel width of 4% to estimate the true joint distribution $p(\mu, E_{smooth})$ from the data contained in the joint distribution in the matrix in *Appendix 3—table 4*. This process estimates an underlying continuous distribution from a discrete set of energy predictions. The details of kernel density estimation can be found in *Hastie et al., 2009*. We can do the final calculation of the mutual information by splitting the smoothed joint distribution into 500 energy 'bins' $z$ and calculating

$$I(\mu,E) = \sum_{z=1}^{500} \sum_{\mu=0}^{1} p(\mu,E_z) \log_2 \left( \frac{p(\mu,E_z)}{p(E_z)p(\mu)} \right).\tag{29}$$

With the ability to calculate the likelihood of an energy matrix model, MCMC can be used to infer the posterior distribution for our model. First a random matrix model is generated. The model is perturbed and the new model is accepted or rejected based on the Metropolis-Hastings algorithm (*Patil et al., 2010*). After an initial burn in period of 60,000 steps, iterations are saved every 60 steps. A total of 600,000 iterations are performed. This procedure is performed twice for each model, and if inferred models do not have a Pearson correlation coefficient of 0.99 or higher they are discarded and computed again. A complete overview of the computational pipeline can be found at the GitHub wiki page.

## Inference of scaling factors for energy matrices

For the majority of energy matrices reported in our work, the results are given in arbitrary units. This is a direct result of using the method of *Kinney et al., 2010* to infer our matrices. The method appeals to information theory to write an 'error-model-averaged' likelihood function for a given model. The likelihood function is given in *Equation (27)*. A property of mutual information is that it is invariant to changes in the input variables as long as those transformations do not affect the rank-order of those variables. As a result, we can scale the energy predictions by any constant without changing the likelihood of the model, which means that in the case of simple linear models for transcription factor binding we cannot assign absolute units to energy matrix values. When we widen our view to considering promoter regions rather than single binding sites we can overcome this drawback. Using thermodynamic modeling as outlined in *Bintu et al., 2005*, we can predict the gene expression from any given transcriptional architecture. In the case a thermodynamic model of simple repression the expression is given by

$$C \propto p_{bound} = \frac{\frac{P}{N_{NS}} e^{-\beta \Delta \varepsilon_P}}{1 + \frac{P}{N_{NS}} e^{-\beta \Delta \varepsilon_P} + \frac{R}{N_{NS}} e^{-\beta \Delta \varepsilon_R}}, \tag{30}$$

where $C$ is a measurement for expression, $P$ is the number of RNAP, $R$ is the number of repressors and $N_{NS}$ is the number of nonspecific binding sites. $\Delta \varepsilon_R$ and $\Delta \varepsilon_P$ represent the difference in the repressor binding energy and RNAP-binding energy between the specific binding site and the averaged nonspecific genomic background respectively. As we use linear energy matrix models as described in Appendix 3 Section 'Energy matrix inference', $\Delta \varepsilon_R$ and $\Delta \varepsilon_P$ will be given by *Equation (26)*. In these cases the overall rank order of gene expression predictions will change if you scale the energy matrix, and so the absolute units can be determined (*Kinney and Atwal, 2014*). *Equation (30)* is a more complicated and non-linear functional form for predicting $C$ than a simple linear binding model, and has a correspondingly more difficult to sample posterior. To address complications in the inference, we first only use the non-linear fits to fix overall scale and wild-type energy for energy matrices rather than fit all parameters in this way. In other words, we use the standard fitting procedure to find the $\theta_{ij}$ in the *Equation (26)* using the standard MCMC procedure.

The binding energy matrices can be written $A \cdot \theta_{ij} + B$ where $A$ is a constant that scales the matrix from arbitrary units to absolute units ($k_B T$) and $B$ is an additive constant that relates to the wild-type energy. We fit the constants $A$ and $B$ for the transcription factor binding energy using the thermodynamic model in *Equation (30)*.

While we can in principle fit thermodynamic models to any given architecture, these models are non-linear and, due to numerical difficulties, unreliable for sufficiently complex models. We only use this method on examples of simple repression or activation without more than one prominent RNAP model, whose transcription-factor-binding site does not overlap significantly with RNAP −10 or −35 sites. The scaling factors we discovered are given in *Appendix 3—table 5*.

**Appendix 3—table 5.** A table showing scaling factors to convert arbitrary units to absolute units in $k_B T$.

Growth conditions indicate the energy matrix and dataset used in the fit. In some growth condition additional regulatory features will be present, meaning specify condition is important.

| Gene | Growth | Scaling factor A |
|---|---|---|
| *tff-rpsB-tsf* | Heat shock | −8.1 $k_B T$ |
| *tig* | Heat shock | −26.3 $k_B T$ |
| *yjjJ* | Heat shock | −11.3 $k_B T$ |
| *bdcR* | Heat shock | −9.9 $k_B T$ |
| *fdhE* | Anaerobic growth | −6.34 $k_B T$ |
| *ykgE* | Arabinose | −12.1 $k_B T$ |
| *dicC* | Arabinose | −15.1 $k_B T$ |
| *rspA* | Arabinose | −5.5 $k_B T$ |

We perform the inference using parallel tempering MCMC, where multiple chains are run in parallel with different 'temperatures'. High temperature chains widely explore parameter space, escaping any local optima, while low temperature chains optimize locally. The current parameter values of the chains are exchanged periodically. The fitting procedure is done using the emcee ensemble sampler (*Goodman and Weare, 2010*) with 10 temperatures ranging from 1 to 10,000 on a log scale.

## Examination of promoters for which no RNAP site was found

We failed to find an RNAP site for 18 promoter sequences. In order to understand these sequences in more detail we examine the sequences within 50 bases of the TSS for the 18 genes in question for sequences which resemble the known consensus RNAP-binding site. For this comparison, we use the $\sigma^{70}$ consensus binding sequence $^{-35}$TTGACA - spacer sequence - TGNTATAAT$^{-7}$ (where the superscripts $^{-35}$ and $^{-7}$ indicate the position relative to the TSS). The consensus sequence we use for comparison contains the extended −10 element, consisting of a TG at bases −15 and −14 as we have found those to be important for gene expression in our study. The spacer length is between 15 and 13 bases (the typically reported spacer length is between 18 and 16 but this does not include the extended minus 10 element). The consensus sequence for the heat shock $\sigma$ factor was used for the promoter *yajL*.

Previously, to analyze RNAP sites, we have examined energy matrices produced by Reg-Seq. Now we add an examination of wild type sequences. For each promoter, we found the best match to the consensus site, namely the sequence with the fewest mutations compared to the consensus sequence. We use the number of mutations as a measure of how well the site resemble consensus. We find that 16 out of 18 promoters have at least five mutations in the sequence that most closely resembles RNAP, one promoter has four mutations, and the last has three mutations. To put these numbers into context, *Brewster et al., 2012* measured the RNAP binding energies of several RNAP-binding site mutants. Mutations away from the strongest sequence tested (*lac*UV5, which is two mutations away from consensus) yields a change in binding energy of $\approx 1 - 2k_B T$. If the promoters are constitutive, then (in the weak promoter approximation) expression level will be proportional to $e^{-\beta \Delta \epsilon_P}$ where $\Delta \epsilon_P$ is the RNAP binding energy relative to the nonspecific background. Therefore, as an approximation, a sequence with three mutations would be predicted to be 3−10 fold weaker than a 'strong' RNAP site, and as such could be said to show a resemblance to the consensus RNAP site. However, 16 out of 18 of these promoter regions have, at best, extremely weak RNAP sites. It is important to note however, that even extremely weak RNAP sites often transcribe, especially when aided by activators. We do not intend to claim that RNAP does not bind to these promoter regions, merely that we do not detect it in our experiment. In fact, while the RNAP sites are weak, there is experimental evidence in EcoCyc of some level of transcription for 9 out of 18 promoters.

## TOMTOM motif comparison

In some cases, we used an alternative approach to mass spectrometry to discover the transcription factor identity regulating a given promoter based on sequence analysis using a motif comparison tool. TOMTOM (*Gupta et al., 2007*) is a tool that uses a statistical method to infer if a putative motif resembles any previously discovered motif in a database. It accounts for all possible offsets between the motifs. Moreover, it uses a suite of metrics to compare between motifs such as Kullback-Leibler divergence, Pearson correlation, Euclidean distance, among others. All TOMTOM analyses in Reg-Seq utilize Euclidean distance. The method calculates a p-value under the null hypothesis that the two compared motifs are independently drawn from the same underlying distribution probability distribution.

We performed comparisons of the motifs generated from our energy matrices to those generated from all known transcription-factor-binding sites in RegulonDB. *Appendix 3—figure 2* shows a result of TOMTOM, where we compared the motif derived from the -35 region of the *ybjX* promoter and found a good match with the motif of PhoP from RegulonDB.

The information derived from this approach was then used to guide some of the transcription factor knockout experiments, in order to validate its interaction with a target promoter characterized by the loss of the information footprint. Furthermore, we also used TOMTOM to search for

similarities between our own database of motifs, in order to generate regulatory hypotheses in tandem. This was particularly useful when looking at the group of GlpR-binding sites found in this experiment.

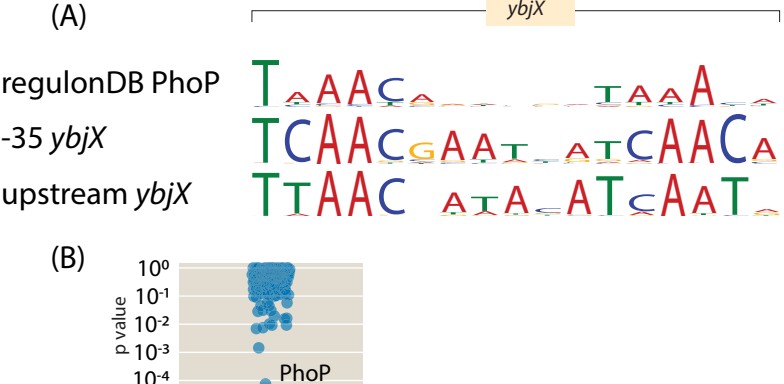

**Appendix 3—figure 2.** Motif comparison using TOMTOM for the two PhoP-binding sites in the *ybjX* promoter. Searching our energy motifs against the RegulonDB database using TOMTOM allowed us to guide our transcription factor knockout experiments. Here, we show the sequence logos of the PhoP transcription factor from RegulonDB (top) and the ones generated from the *ybjX* promoter energy matrix. E-value = 0.01 using Euclidean distance as a similarity matrix.

## Appendix 4

### Additional results
#### Binding sites regulating divergent operons

In addition to discovering new binding sites, we have discovered additional functions of known binding sites. In particular, in the case of *bdcR*, the repressor for the *bdcA* gene, which is transcribed from the same promoter in the opposite direction of transcription (*Partridge et al., 2009*), is also shown to repress *bdcR* in *Appendix 4—figure 1(A)*. Similarly in *Appendix 4—figure 1(B)* IvlY is shown to repress *ilvC* in the absence of inducer. Divergently (transcription in opposite directions from the same promoter) transcribed operons that share regulatory regions are plentiful in *E. coli*, and although there are already many known examples of transcription-factor-binding sites regulating several different operons, there are almost certainly many examples of this type of transcription that have yet to be discovered.

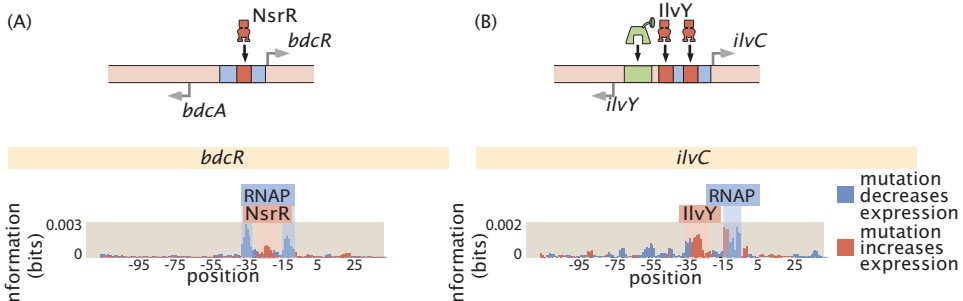

**Appendix 4—figure 1.** Two cases in which we see transcription-factor-binding sites that we have found to regulate both of the two divergently transcribed genes. (**A**) An information footprint and regulatory cartoon for the divergently transcribed *bdcA* and *bdcR* promoters. A single NsrR site regulates both promoters. (**B**) An information footprint and regulatory cartoon for the *ilvC* and *ilvY* promoters. Both promoters are repressed by IlvY when grown without acetolactate. Only the IlvY site is labeled on the information footprint.

In the case of *ilvC*, IlvY is known to activate *ilvC* in the presence of inducer. However, we now see that it also represses the promoter in the absence of that inducer. The production of *ilvC* is known to increase by approximately a factor of 100 in the presence of inducer (*Rhee et al., 1998*). The magnitude of the change is attributed to the cooperative binding of two IlvY-binding sites, but the lowered expression of the promoter due to IlvY repression in the absence of inducer is also a factor.

### Comparison of results to RegulonDB

One area in which our work can be compared to current repositories of regulatory information such as RegulonDB is in comparing the prevalence of different regulatory architectures in the database to Reg-Seq. *Appendix 4—figure 2* shows the prevalence of each type of architecture (not including architectures more complex than 2 activators and 2 repressors) and shows how simpler architectures are more common in both cases.

Another point of comparison between RegulonDB and Reg-Seq can be found in comparing sequence motifs from Reg-Seq to those generated from binding sites in RegulonDB. This can often produce useful results, such as in Appendix 3 Section 'TOMTOM motif comparison'. For other cases, the data used to generate the RegulonDB motifs can be lacking. We believe the GlpR motif in RegulonDB highlights some of the issues with using the reported motifs in RegulonDB to predict binding preference. First, there are only four promoters regulated by GlpR, with a total of 17 binding sites for GlpR in RegulonDB. Nine of these binding sites differ by nine mutations or more from the consensus site (out of 22 total base pairs). Nine mutations is more than even the weak O3 operator for LacI. We do believe that a relatively low number of weakly conserved binding sites likely do not reveal quality sequence logos for a binding site, especially as compared to Reg-Seq which constructs

sequence logos from over a thousand promoter variants. Generation of such sequence motifs is a point on which we believe Reg-Seq can improve the current status of regulatory knowledge.

## Explanation of included binding sites

This section is intended to clarify cases in which the regulatory cartoon or the displayed 'expected' binding sites differs from what can be found in RegulonDB or EcoCyc. The primary reason for these discrepancies is that our experiment only targets a 160 base pair mutation window. Some known binding sites will be outside of this window. Additionally, while some genes are known to be regulated by a specific transcription factor, the exact location of that transcription factor's binding site is unknown and so we cannot be certain during the design of the 160 base pair mutagenized window whether or not the transcription-factor-binding site will be present in our experiment. The locations of the TSS selected in this experiment can be found in *Supplementary file 1*. Additionally, some transcription factors are known to only be active under certain growth conditions. Information footprints are depictions of the regulatory information for a specific growth condition; accordingly, not all transcription-factor-binding sites can be identified using a single growth condition. Throughout the main text and SI, however, we depict regulatory cartoons with their full milieu of transcription factors (based on experiments performed in multiple growth conditions).

When devising this study, we sought to test the reliability of the Reg-Seq method by testing experimentally-validated transcription-factor-binding sites, as reported by EcoCyc or RegulonDB, to assess our ability to recapitulate prior experiments. EcoCyc labels some transcription-factor-binding sites as 'low-evidence' in their database, most of which were identified via sequence motif matching. We have repeatedly observed that transcription-factor-binding sites identified from sequence matching are unreliable in relation to the empirical data collected in our experiment, and so we choose not to include them in the set of 'gold standard' genes which were used for this purpose of assessing Reg-Seq's accuracy.

All of our 'gold standards' are genes for which there is high quality experimental evidence of their transcriptional regulation and the location of related transcription-factor-binding sites and, again, they were used to evaluate the false negative rates of our experiment. In those cases where the binding sites are either 'low-evidence' according to EcoCyc, the location of a binding site is not known, a gene is only actively transcribed in certain or unknown growth conditions, or the binding site location is outside of the 160 bp mutagenized region, we do not include them in the list of sites we use to test our method even though they appear as binding sites in RegulonDB or EcoCyc. Regulatory features that are not transcription factors, including regulatory RNAs, are also not labeled in our reported results.

Accordingly, in some cases, the regulatory cartoons or architectures we present in this study may appear to be incomplete relative to previous reports of promoter architectures. For each gene below, we explain these discrepancies. This section is intended to explain why annotations on information footprints or regulatory cartoons do not match what is seen in RegulonDB or EcoCyc.

### sdiA

*sdiA* is known to be regulated by both Nac as well as CsrA (which has two binding sites), the CsrA sites are downstream of the mutated region and the location of the Nac-binding site is unknown. Thus, none of these binding sites are reported in our regulatory architectures for this gene.

### yqhC

*yqhC* is known to be regulated by GlaR, but the location of this binding site is unknown. As a result, we were unable to identify this binding site in our analysis, and the architecture for *yqhC* is listed in this study as (0,0).

### bdcR

*bdcR* is known to be regulated by GlaR, but this binding site is outside of the targeted mutation window of 160 bp. A known binding site for NsrR is included within the 160 bp region, but it was not

previously known to regulate *bdcR*; the binding site for NsrR is included as a new discovery as shown in Section 'Binding sites regulating divergent operons'.

### aegA

*aegA* has a predicted CRP-binding site, but the location of this binding site is unknown and it is also listed as low-evidence in EcoCyc. As a result, the site is not included within this study's analysis.

### hicB

The CRP site associated with *hicB* is cited as low-evidence in EcoCyc and the HicB-binding site is outside of the 160 base pair mutated region. As a result, neither site is included in this study.

### rplKAJL-rpoBC

The known RplA-binding site for this operon is outside of the targeted, 160 base pair mutation window. As a result, the RplA site is not included in this study.

### tff-rpsB-tsf

RpsB is not contained in the mutated region. Additionally, the nearby predicted Mar-Sox-Rob-binding site is listed as low-evidence in EcoCyc and is also not directly predicted to regulate *tff-rpsB-tsf*, even though it may be present within the region. As a result, neither site is included in this study.

### yodB

GlaR is known to regulate *yodB*. However, the location of this binding site is unknown. As a result, we do not include the GlaR-binding site in our reported regulatory architecture for this gene.

### maoP

HdfR is known to regulate *maoP*. However, the location of the binding site is unknown. Additionally, the HdfR site is listed as low-evidence in EcoCyc. During the Reg-Seq experiment, however, we confirmed the presence of the low-evidence HdfR site with a gene knockout and located the binding site position. Thus, we include it in all regulatory cartoons and report the HdfR site in our discoveries.

### poxB

MarA and Sox have low-evidence binding sites in the mutagenized region. There is also a low-evidence site for Cra with an unknown binding location. As a result, neither site is included in the reported regulatory architectures in this study.

### mscM

While there is a known CpxR-binding site for *mscM*, the binding site exists outside of the mutagenized region. As a result, it is not included in the reported regulatory architectures in this study.

### tar

There is a low-evidence FNR site for *tar*. Its location is unknown. For both of these reasons, we do not include the binding site in our reported regulatory architectures for this gene.

### dpiBA

While there are 10 total binding sites for *dpiBA*, including an FNR site. However, the only ones that are known to regulate the particular TSS we chose (at position 652172 in *E. coli*) are 2 DcuR sites and a (low-evidence) NarL site. DcuR is induced by growth conditions like succinate or fumarate, neither of which were tested in this study. As a result, none of the sites are included in this study.

### araAB

There are a total of five AraC-binding sites and one CRP-binding site that regulate *araAB*. However, the three furthest upstream AraC-binding sites are outside of the 160 bp mutagenized region, and so only two AraC sites and one CRP site is included in the reported regulatory architecture in this study.

### xylF

There are two XylR sites, as well as three low-evidence Fis sites that regulate *xylF* in the mutagenized region. There is also a low-evidence CRP site outside the mutagenized region. Only the two XylR sites are included in the reported regulatory architectures, as the remaining sites are low-evidence or outside the mutagenized region.

### xylA

There are two XylR sites, two AraC, and a CRP site that regulates xylA. In our analysis, we utilize a growth condition containing xylose and arabinose. Under growth with xylose, XylR will bind DNA and activate expression. Under growth with arabinose, AraC will not bind DNA. We would only expect to see two XylR sites and a CRP site under growth in xylose and arabinose, so we only include these sites in our study.

### dicB

DicA has a low-evidence repressor-binding site for *dicB*. Additionally the binding location is unknown, and so we do not include the binding site in the reported regulatory architecture.

### xapAB

XapR has two low-evidence binding sites. The binding site furthest upstream is outside of the 160 bp mutagenized region. As the remaining site is low-evidence, it is not included in our reported regulatory architectures.

### ilvC

There are two IlvY-binding sites for *ilvC*. IlvY is known to be induced by acetolactate and activated in its presence . We do not utilize this growth condition in this experiment, however, nor do we include the two IlvY-binding sites in our 'gold standard' experimental analysis. We find that IlvY acts as a repressor when grown in other growth conditions. As repressor activity at these sites was not previously reported, we include this in our list of new discoveries.

### asnA

There are four low-evidence AsnC-binding sites in the mutated region. As they are low-evidence, however, we do not include these binding sites in the reported regulatory architectures for this gene.

### idnK

While there are three GntR sites, a CRP site, and one IdnR site, they are all low-evidence. As a result, we do not include any of these sites in our reported regulatory architectures.

### dinJ

While *dinJ* is regulated by DinJ-YafQ and LexA, they are both outside of the mutagenized window. As a result, neither are included in our reported regulatory architectures.

### yjiY

*yjiY* is regulated by both BtsR and CRP. However, CRP is outside of the mutagenized window and so CRP is not included in our reported regulatory architectures.

### cra

*cra* is regulated by a low-evidence binding site of PhoB. The location of the binding site is not known, however. As a result, the site is not included in the reported regulatory architecture.

### uvrD

*uvrD* is regulated by a low-evidence binding site for LexA. This binding site is not included in the reported regulatory architectures for this study.

### znuCB

There are binding sites for Zur and OxyR in the mutagenized region for *znuCB*. OxyR is known to act as an activator under oxidative stress. As we do not utilize an oxidative stress growth condition in this study, we do not include this binding site in the reported regulatory architectures for this study.

### znuA

There are binding sites for Zur and OxyR in the mutagenized region for *znuA*. The OxyR-binding site is outside of the mutagenized region. Only the Zur-binding site is included our reported regulatory architectures.

### pitA

There is a low-evidence binding site for FNR in the mutagenized region. The location of this binding site, however, is unknown. Thus, this binding site is not included in our reported regulatory architectures.

### ecnB

There is a low-evidence OmpR-binding site for *ecnB*. The binding site is not included in our reported regulatory architectures.

### lacZYA

The mutagenized region extends from the TSS (the primary TSS p1) to 75 base pairs upstream of the TSS. The location of the mutagenized region excludes the LacI sites, while including a single CRP-binding site, a MarA-binding site, and two HNS-binding sites. The expression from *marA* is expected to be low, as we do not grow the cells in the presence salicylate or antibiotic stress and so we do not expect to observe the MarA site. In fact, the precursor of the Reg-Seq experiment, Sort-Seq, mutagenized and studied the same 75 base pair region, and only observed binding by CRP (*Kinney et al., 2010*). As such, we only include CRP in *Table 2*, the regulatory cartoons, or the analysis of false positives and false negatives.

### leuABCD

There is a binding site for LeuO regulating *leuABCD*. The site is low-evidence and also has no known binding location. As a result, the site is not included in our reported regulatory architectures.

### arcA

There is a binding site for FNR within the mutagenized region listed as 'low-evidence' in EcoCyc. We find substantial additional evidence for the presence of the FNR-binding site. As such, we include the site in *Table 2* as an 'Identified Binding Site'.

### relBE

The *relBE* promoter contains four RelBE-binding sites and two RelB-binding sites in EcoCyc and RegulonDB. While the all four RelBE sites are listed as high evidence, *Belliveau et al., 2018* mutagenized the RelBE promoter and did not identify binding in the furthest downstream or furthest upstream binding sites. Also, the original identification of the RelBE-binding sites presented (*Li et al., 2008*), claims that the furthest upstream and downstream sites are only identified by similarity to consensus sequence. As a result only two of the RelBE and two of the RelB sites are included in this study.

### marR

The *marR* promoter contains a CpxR, CRP, Cra, and AcrR in EcoCyc that are not included in the 'gold standard' analysis or *Table 2*. *Belliveau et al., 2018* performed mutagenesis experiments on the *marR* promoter and did not identify these additional sites and so they have been excluded.

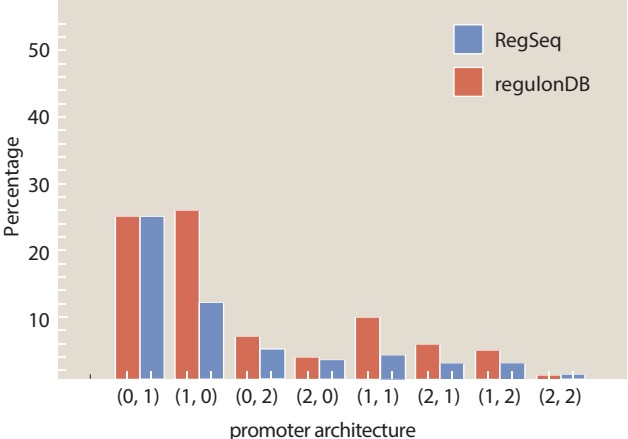

**Appendix 4—figure 2.** A comparison of the types of architectures found in RegulonDB (*Santos-Zavaleta et al., 2019*) to the architectures with newly discovered binding sites found in the Reg-Seq study. For each type of architecture, labeled as (number of activators, number of repressors), the fraction that architecture comprises of the total number of operons is given both for the data found in RegulonDB and from the results of the Reg-Seq experiment. Numeric values for the displayed data can be found in *Appendix 4—figure 2—source data 1*.

The online version of this article includes the following source data is available for figure 2:

**Appendix 4—figure 2—source data 1.** Source data for the percentage composition of regulatory architectures.

# Appendix 5

## Resource Table

**Appendix 5—key resources table**

| Reagent type (species) or resource | Designation | Source or reference | Identifiers | Additional information |
|---|---|---|---|---|
| Cell line (*Escherichia coli*) | *E. coli* K12 | *E. coli* Stock Center | | |
| Cell line (*Escherichia coli*) | *E. coli* ΔYieP | *E. coli* Stock Center | | |
| Cell line (*Escherichia coli*) | *E. coli* ΔGlpR | *E. coli* Stock Center | | |
| Cell line (*Escherichia coli*) | *E. coli* ΔArcA | *E. coli* Stock Center | | |
| Cell line (*Escherichia coli*) | *E. coli* ΔLrhA | *E. coli* Stock Center | | |
| Cell line (*Escherichia coli*) | *E. coli* ΔPhoP | *E. coli* Stock Center | | |
| Cell line (*Escherichia coli*) | *E. coli* ΔHdfR | *E. coli* Stock Center | | |
| Strain, strain background (*Escherichia coli*) | *E. coli* ΔGlpR in K12 strain | This paper | | Knockout transferred to *E. coli* K12 |
| Strain, strain background (*Escherichia coli*) | *E. coli* ΔArcA in K12 strain | This paper | | Knockout transferred to *E. coli* K12 |
| Strain, strain background (*Escherichia coli*) | *E. coli* ΔLrhA in K12 strain | This paper | | Knockout transferred to *E. coli* K12 |
| Strain, strain background (*Escherichia coli*) | *E. coli* ΔPhoP in K12 strain | This paper | | Knockout transferred to *E. coli* K12 |
| Strain, strain background (*Escherichia coli*) | *E. coli* ΔHdfR in K12 strain | This paper | | Knockout transferred to *E. coli* K12 |
| Chemical compound, drug | Q5 Polymerase | Qiagen | Cat. : M0491L | |
| Chemical compound, drug | qPCR master mix | QuantaBio | Cat. : 101414–166 | |
| Chemical compound, drug | Lysyl Endopeptidase | Wako Chemicals | Cat. : 125–05061 | |

*Continued on next page*

*Appendix 5—key resources table continued*

| Reagent type (species) or resource | Designation | Source or reference | Identifiers | Additional information |
|---|---|---|---|---|
| Commercial assay or kit | RNEasy Mini kit | Qiagen | Cat. : 74104 | |
| Chemical compound, drug | RNAprotect bacteria reagent | Qiagen | Cat. : 76506 | |
| Software, algorithm | mpathic | Kinney Lab *Ireland and Kinney, 2016* | | |
| Software, algorithm | FastX | Hannon Lab | RRID:SCR_005534 | |
| Software, algorithm | FLASH | CBCB | RRID:SCR_005531 | |
| Other | Oligo Pool | Twist Bioscience | | |
| Sequence-based reagent | fwd oligo 101 | IDT | | TTCGTCTTCACCT CGAGCACGCTTATT CGTGCCGTG TTAT |
| Sequence-based reagent | fwd oligo 102 | IDT | | TTCGTCTTCACCTC GAGCACTTTGCTT CAGTCAGA TTCGC |
| Sequence-based reagent | fwd oligo 103 | IDT | | TTCGTCTTCACCT CGAGCACGTCGAGT CCTATG TAACCGT |
| Sequence-based reagent | fwd oligo 104 | IDT | | TTCGTCTTCACCT CGAGCACGTAAGAT GGAAGCCGGGATA |
| Sequence-based reagent | fwd oligo 105 | IDT | | TTCGTCTTCACCT CGAGCACGGTGTCGC AACATGA TCTAC |
| Sequence-based reagent | fwd oligo 106 | IDT | | TTCGTCTTCACCT CGAGCACGTGCTAAG TCACACTG TTGG |
| Sequence-based reagent | fwd oligo 107 | IDT | | TTCGTCTTCACCT CGAGCACTCTAAACA G TTAGGCCCAGG |
| Sequence-based reagent | fwd oligo 108 | IDT | | TTCGTCTTCACCT CGAGCACGTCTTTAT ACTTGCC TGCCG |
| Sequence-based reagent | fwd oligo 109 | IDT | | TTCGTCTTCACCT CGAGCACCACCGCGA TCAA TACAACTT |
| Sequence-based reagent | fwd oligo 110 | IDT | | TTCGTCTTCACCT CGAGCACTTCGGATA GAC TCAGGAAGC |
| Sequence-based reagent | fwd oligo 111 | IDT | | TTCGTCTTCACCT CGAGCACCCATTGAT AGATTCGC TCGC |
| Sequence-based reagent | fwd oligo 112 | IDT | | TTCGTCTTCACCT CGAGCACTTTTCTAC TTTCCGGC TTGC |
| Sequence-based reagent | fwd oligo 113 | IDT | | TTCGTCTTCACCT CGAGCACATGACTAT TGGGGTCG TACC |

*Continued on next page*

*Appendix 5—key resources table continued*

| Reagent type (species) or resource | Designation | Source or reference | Identifiers | Additional information |
|---|---|---|---|---|
| Sequence-based reagent | fwd oligo 114 | IDT | | TTCGTCTTCACCT CGAGCACTCGACAAT AG TTGAGCCCTT |
| Sequence-based reagent | fwd oligo 115 | IDT | | TTCGTCTTCACCT CGAGCACGAGCCATG TGAAATG TGTGT |
| Sequence-based reagent | fwd oligo 116 | IDT | | TTCGTCTTCACCT CGAGCACCGTATACG TAAGGG TTCCGA |
| Sequence-based reagent | fwd oligo 117 | IDT | | TTCGTCTTCACCT CGAGCACTTATGATG TCCGGA TACCCG |
| Sequence-based reagent | fwd oligo 118 | IDT | | TTCGTCTTCACCT CGAGCACTCTTAGAA A TCCACGGGTCC |
| Sequence-based reagent | rev oligo 101 | IDT | | TGTAAAACGACGG CCAGTGACTAGCGC TGAGGAGAAGCCT AATAGGGCACAGC AATCAAAAG TA |
| Sequence-based reagent | rev oligo 102 | IDT | | TGTAAAACGACG GCCAGTGAGGAGCGC TGAGGAGAAGCC TAATACCGGGATT CAGTGA TTGAAC |
| Sequence-based reagent | rev oligo 103 | IDT | | TGTAAAACGACG GCCAGTGAGTCCC GC TGAGGAGAAG CCTAATATGAAGAT ATGACGACCCC TG |
| Sequence-based reagent | rev oligo 104 | IDT | | TGTAAAACGACGG CCAGTGACCGACGCT GAGGAGAAGCCTAA TATTCCACAGCTC TATGAGG TG |
| Sequence-based reagent | rev oligo 105 | IDT | | TGTAAAACGACGG CCAGTGATTGGCGCT GAGGAGAAGCCTA ATAGCAAACATGA C TAGGAACCG |
| Sequence-based reagent | rev oligo 106 | IDT | | TGTAAAACGACGG CCAGTGAGATACGC TGAGGAGAAGCC TAATACCGGGACG AGATTAG TACAA |
| Sequence-based reagent | rev oligo 107 | IDT | | TGTAAAACGACGGC CAGTGAACTCCGCT GAGGAGAAGCCTA ATACACGCCAGTT GTGAACA TAA |
| Sequence-based reagent | rev oligo 108 | IDT | | TGTAAAACGACG GCCAGTGATACTCGC TGAGGAGAAGC CTAATACAAAGGC CAAATCAG TTCCA |
| Sequence-based reagent | rev oligo 109 | IDT | | TGTAAAACGACGGC CAGTGACCAACGCT GAGGAGAAGCCT AATAGGTGCATGGG AGGAACTA TA |
| Sequence-based reagent | rev oligo 110 | IDT | | TGTAAAACGACG GCCAGTGAAGGCCGC TGAGGAGAAGCCT AATATGCATGGGT CTGTCTATTG T |
| Sequence-based reagent | rev oligo 111 | IDT | | TGTAAAACGACGGC CAGTGAAATTCGC TGAGGAGAAGCCT AATACTCCTATGCT AGCTCGAC TC |
| Sequence-based reagent | rev oligo 112 | IDT | | TGTAAAACGACG GCCAGTGATTGT CGC TGAGGAGAAG CCTAATAATGGTA AGAAGC TCCCACAA |
| Sequence-based reagent | rev oligo 113 | IDT | | TGTAAAACGACGGC CAGTGATTTACGCT GAGGAGAAGCCTA ATACTATGGTCA TTCCCG TACGA |

*Continued on next page*

*Appendix 5—key resources table continued*

| Reagent type (species) or resource | Designation | Source or reference | Identifiers | Additional information |
|---|---|---|---|---|
| Sequence-based reagent | rev oligo 114 | IDT | | TGTAAAACGACGGC CAGTGAACCGCGCT GAGGAGAAGCCTA ATATAATCGGCT ACGTTGTGTCT |
| Sequence-based reagent | rev oligo 115 | IDT | | TGTAAAACGACGGC CAGTGATGGCCGC TGAGGAGAAGC CTAATATGACTCGA TCCTTTAG TCCG |
| Sequence-based reagent | rev oligo 116 | IDT | | TGTAAAACGACGG CCAGTGAGGCCCGC TGAGGAGAAGC CTAATAACGCTTT GTGTTATCCGA TG |
| Sequence-based reagent | rev oligo 117 | IDT | | TGTAAAACGACGG CCAGTGAGGTGCG C TGAGGAGAAG CCTAATAACCACG GTGGAGTATACA TC |
| Sequence-based reagent | rev oligo 118 | IDT | | TGTAAAACGACG GCCAGTGACAATCG C TGAGGAGAAGC CTAATAGGCACCA GGTACATATC TCA |
| Sequence-based reagent | mRNA rev | IDT | | GCAGGGGATAA TATTGCCCA |
| Sequence-based reagent | fwd sequencing 94 | IDT | | AATGATACGGCGACCAC CGAGATCT ACACTCTT TCCCTACACGACGC TCTTCCGATCTGACC TA TTAGGCTT CTCCTCAGCG |
| Sequence-based reagent | fwd sequencing 95 | IDT | | AATGATACGGCGACCAC CGAGATCT ACACTCTT TCCCTACACGACGC TCTTCCGATCTCAGT TA TTAGGCTT CTCCTCAGCG |
| Sequence-based reagent | fwd sequencing 96 | IDT | | AATGATACGGCGACCAC CGAGATCT ACACTCTT TCCCTACACGACGC TCTTCCGATCTTCTA TA TTAGGCTT CTCCTCAGCG |
| Sequence-based reagent | fwd sequencing 97 | IDT | | AATGATACGGCGACCAC CGAGATCT ACACTCTT TCCCTACACGACGC TCTTCCGATCTAGAG TA TTAGGCTT CTCCTCAGCG |
| Sequence-based reagent | fwd sequencing 98 | IDT | | AATGATACGGCGACCAC CGAGATCT ACACTCTT TCCCTACACGACGC TCTTCCGATCTGCAT TA TTAGGCTT CTCCTCAGCG |
| Sequence-based reagent | fwd sequencing 99 | IDT | | AATGATACGGCGACCAC CGAGATCT ACACTCTT TCCCTACACGACGC TCTTCCGATCTCTTA TA TTAGGCTT CTCCTCAGCG |
| Sequence-based reagent | fwd sequencing 100 | IDT | | AATGATACGGCGACCAC CGAGATCT ACACTCTT TCCCTACACGACGC TCTTCCGATCTTAGC TA TTAGGCTT CTCCTCAGCG |
| Sequence-based reagent | fwd sequencing 101 | IDT | | AATGATACGGCGACCAC CGAGATCT ACACTCTT TCCCTACACGACGC TCTTCCGATCTCAAG TA TTAGGCTT CTCCTCAGCG |
| Sequence-based reagent | fwd sequencing 102 | IDT | | AATGATACGGCGACCAC CGAGATCT ACACTCTT TCCCTACACGACGC TCTTCCGATCTGTAC TA TTAGGCTT CTCCTCAGCG |
| Sequence-based reagent | fwd sequencing 103 | IDT | | AATGATACGGCGACCAC CGAGATCT ACACTCTT TCCCTACACGACGC TCTTCCGATCTTGAA TA TTAGGCTT CTCCTCAGCG |
| Sequence-based reagent | fwd sequencing 104 | IDT | | AATGATACGGCGACCAC CGAGATCT ACACTCTT TCCCTACACGACGC TCTTCCGATCTTCGT TA TTAGGCTT CTCCTCAGCG |
| Sequence-based reagent | fwd sequencing 105 | IDT | | AATGATACGGCGACCAC CGAGATCT ACACTCTT TCCCTACACGACGC TCTTCCGATCTATGC TA TTAGGCTT CTCCTCAGCG |

*Continued on next page*

*Appendix 5—key resources table continued*

| Reagent type (species) or resource | Designation | Source or reference | Identifiers | Additional information |
|---|---|---|---|---|
| Sequence-based reagent | fwd sequencing 106 | IDT | | AATGATACGGCGACCAC CGAGATCT ACACTCTT TCCCTACACGACGC TCTTCCGATCTGTCA TA TTAGGCTT CTCCTCAGCG |
| Sequence-based reagent | fwd sequencing 107 | IDT | | AATGATACGGCGACCAC CGAGATCT ACACTCTT TCCCTACACGACGC TCTTCCGATCTCTCA TA TTAGGCTT CTCCTCAGCG |
| Sequence-based reagent | fwd sequencing 108 | IDT | | AATGATACGGCGACCAC CGAGATCT ACACTCTT TCCCTACACGACGC TCTTCCGATCTAGTA TA TTAGGCTT CTCCTCAGCG |
| Sequence-based reagent | rev sequencing | IDT | | AAGCAGAAGACGGCAT ACGAGATCGGT CTCG GCA TTCCTGCTGAACC GCTCTTCCGATCTCAAA GCAGGGGATAA TATTGCCCA |
| Other | Streptavin coated dynabeads | Thermo Fisher | Cat. : 65601 | |
| Database | RegulonDB | | RRID:SCR_ 003499 | |
| Database | EcoCyc | | RRID:SCR_ 002433 | |

