## [Decision Letter]

**Acceptance summary:**

The manuscript presents a new approach, Reg-Seq, to study regulatory logic in hundreds of promoters at a time. To do so, the authors use massively parallel reporter assays and mass spectroscopy and build an information theoretical model to characterize binding energies in regulatory promoters at the base-pair resolution level. This approach can significantly advance our understanding of gene regulation in microbes and opens new avenues towards genome-wide quantification of regulatory logic and discovery of new transcription factors.

**Decision letter after peer review:**

Thank you for submitting your article "Deciphering the regulatory genome of *Escherichia coli*, one hundred promoters at a time" for consideration by *eLife*. Your article has been reviewed by three peer reviewers, and the evaluation has been overseen by a Reviewing Editor and Naama Barkai as the Senior Editor. The following individual involved in review of your submission has agreed to reveal their identity: Tamar Friedlander (Reviewer #2).

The reviewers have discussed the reviews with one another and the Reviewing Editor has drafted this decision to help you prepare a revised submission.

Comments:

This manuscript introduces a new method "Reg-Seq" for identifying regulatory sites in promoters using RNA sequencing for cell sorting. The method generates impressive quantitative data for many genes in *E. coli*, which corroborates many previous results and starts to give us hints about the overall trends of regulatory architectures in *E. coli*. Although the experimental set up is very compelling and potentially promising, there are still some major concerns regarding presentation of the data and the underlying methods and analyses, which we would like to see addressed.

The main concerns about the manuscript can be summarized as: 1) Experimental method should be better discussed in the main text, its limitations should be explicitly pointed out and its advantages should be compared more clearly against previous techniques. 2) Statistical methods (including inference based on mutual information profiles) should be described more clearly and with more quantitative details throughout the text and the significance of the conclusions should be more explicitly expressed. 3) Analysis of *E. coli* data, comparison with previously identified regulatory regions and binding sites and the significance of these findings should be better discussed.

The reviewers strongly believe that the manuscript should be reworked to focus less on cherry-picked examples and more on presenting the method. The manuscript needs a significant rearrangement to better explain the procedures applied and fill in various missing points as detailed below.

Essential revisions:

1) The approach based on the mutual information profile to identify the binding sites within a promoter and to classify them as activator or repressors should be better explained:

1.1) Since this is such a key step, the authors need to provide more details about how the manual choice of activator/ repressors works. Overall, it is unclear how one goes from mutual information profiles to deciding whether known sites are or are not identified, how to decide which regions are putative sites, whether they fit known motifs, whether they are RNAP sites, and how to construct an energy matrix for each putative site.

1.2) It is mentioned (subsection “Analysis of sequencing results”, first paragraph) that, to confirm these manual choices, authors computationally identify regions of activators / repressors by assessing the fold change in gene expression due to mutations. It is unclear how the authors are assessing the significance of the changes, i.e., what the underlying null model is. If this computational approach is statistically sound, why the need for the manual approach? These manual choices are worrisome and raise concerns about the feasibility to scale-up this method, as intended by the authors.

2) It is difficult to assess the false positive/negative rate of identifying binding sites: This is partly due to the lack of clarity about the authors' approach in deciding whether a known site was identified or not, partly because the annotation of known sites is unclear and incomplete, and partly because the statistics are not transparently reported in the manuscript. It is therefore important that the authors present an objective method for assessing what fraction of known sites is not recovered, for what fraction of predicted novel sites a motif can be made, and for what fraction of those the mass-spec successfully recovers a binding TF.

As a suggestion: The authors should start from a unambiguously defined set of sites (e.g. from RegulonDB and EcoCyc) and then have an unambiguous procedure for:

i) Calling segments where a site exists (for example based on the summed mutual information within the segment).

ii) Calling the motif by comparing the profile with PSWMs or energy matrices for known motifs.

After that, they should clearly report what fractions of sites are recovered and for what fraction of sites the correct motif is predicted.

3) Energy matrices:

3.1) How are the energy matrices constructed? Details are missing from the manuscript.

3.2) How are the p-values estimated to assign significance to energy parameters? It appears that the authors have used MCMC to sample from a likelihood function but the details are very vague. To define an energy, it is assumed that the average expression can be written as the negative exponential of some effective energy to parametrize the average effects of mutations on expression- this should be better justified in the text. However, it seems that instead of 4 different energy parameters at each position, the model only assumes a “mutant” and a “wild type” energy. This may not be a good approximation given that we know nucleotides within TF binding sites have highly varying degeneracies. Moreover, the likelihood function for the observed DNA and RNA reads should be expressed as a function of the energy parameters at all positions. However, this is not the approach taken by the authors and instead, equations (8) and (9) only look at single positions. So, it is not clear what kind of likelihood function the MCMC is sampling from, or how p-values and confidence intervals are determined. Also, if the authors are indeed using MCMC with some likelihood function to sample the space of possible parameter values, the resulting variation in the energy parameters should correspond to posterior probability intervals and not confidence intervals.

3.3) Could the authors present the accuracy of the inferred energy matrices as a function of the number of genomic variants used? A particular example is currently given in Appendix 3. Can the authors use a simulation for that purpose? This is required to understand the validity of the results and the ability to scale-up such an experiment (if for example a lower resolution is sufficient, can larger regulatory regions be analyzed in future experiments?)

4) The step for assigning regulatory logic to promoters is unclear. Do authors always assume OR-gate-like interactions between the TFs, which is what Equation 14 suggests? If so, what is the justification for that assumption? Is it just the most basic statistical mechanical assumption, or is there strong empirical support?

5) The description of the method and analysis is hard to follow and unclear about many assumptions and limitations. To give a few examples:

5.1) Appendix 3 outlines how to calculate the mutual information footprint from the read count data but a more explicit discussion would be helpful: For example, how should the quantities in Equation 5 (*p(m,* 𝜇*), p(m), p(*𝜇)) be calculate from the preceding example data (subsection “Information footprints”)?

5.2) The methods rely on both DNA and RNA sequencing of the same sample and using ratios of RNA/DNA reads from the same reporter to estimate expression. The description of the sequencing protocol is insufficient. Importantly, the DNA sequencing is not even mentioned in the methods and it is not explained how precisely this expression quantification is done.

5.3) The methods mention correcting for correlated mutations, MCMC to assign p-values, and energy matrix reconstruction, but it is completely unclear how either of these are done.

6) The discussion of the methods in the main text is mostly qualitative and crucial details are scattered throughout the manuscript, in the appendices and in Materials and methods.

6.1) For example, it requires digging into the appendices to understand that the authors study a region of 160 bp mostly upstream TSS, and so, is this method inadequate if the regulatory region is larger or if its location is unknown? A clear and concise explanation of the crucial details (e.g. size of the region, number of variants per promoter, etc) should be given in the main text.

6.2) To further clarify the methods, it would be helpful to include a picture of the reporter construct, which based on the descriptions includes:

-promoter including 45 bp downstream of annotated TSS.

-then 64bp with primers for plasmid construction

-then 11bp with stop codons in 3 frames.

-then barcodes?

-then a RBS

-then GFP mRNA

7) Authors used TSS information to decide which promoters to pursue. How limiting is this for scaling up the procedure? It would be helpful to discuss briefly what fraction of promoters in *E. coli* have good TSS information and how valid is to assume that at the genome-wide level, each operon is dominated by a single TSS?

8) The authors rely on expression from plasmids and use mRNA/DNA ratio to handle the effect of variability in plasmid copy number between cells. However, if the plasmid copy number is of a similar order of magnitude as the transcription factor copy number, then the expression level measured (to calculate the energy matrices) is determined not only by the binding energy, but also by the TF availability leading to under-estimation of the binding energies. The authors should comment on this in the manuscript, and if they have the data available, show the measurements for plasmid and TF copy numbers to address this point. At this point we do not see a necessity for additional experiments.

9) The limitations of the method should be more concisely explained. Currently, limitations are scattered throughout the text.

Revisions required for Figures and Tables:

Figure 1: This figure is hard to read. It is difficult to distinguish the individual tick marks around the genome, because there are too many, they are too densely packed, and the colors are too mixed. Also, the caption describes the color of some ticks as "red," but in the printed figure they look more brown (They appeared closer to red on the screen.)

Figure 2: You might want to clarify that the blue region likely corresponds to the σ factor binding site.

Figure 3: The number of (0,0) promoters should be shown as well. Moreover, it seems that the counts in Figure 3 add up to about 50 and the text mentions that there are 32 promoters with no sites found, i.e. type (0,0). Adding this in the sum still seems much less than 113 (i.e., the total number of promoters as indicated in the manuscript). Related to this, it seems that Table 2 has clearly less than 113 promoters in it. Where are the remaining ones?

Figure 4: The authors state about the results in Figure 4: "In each of the cases shown in the figure, prior to the work presented here, these promoters had no regulatory information in relevant databases such as EcoCyc (Keseler et al., 2016) and RegulonDB (Santos- 318 Zavaleta et al., 2019)."

This is simply not true. If you check EcoCyc for these genes you find:

i) *idnK* regulated by CRP, IdnR and GtnR.

ii) *leuABCD* regulated by LeuO, slyA, LrhA, and RcsB-BgU.

iii) *maoP* regulated by hdfR.

iv) *rspA* regulated by CRP and YdfH.

Only for *yjjJ*, *aphA*, and *ybjX* nothing was previously reported. It should also be noted that except for the *rspA* promoter, the previously reported regulatory interactions were generally not identified by the method.

Finally, it seems that there is no mass-spec data for *yjjJ*. Is the identification of marA as the regulator based on motif matching then?

Table 2: This table is unclear. Does the "architecture" of a promoter (first column) correspond to what is inferred in this study, to what is known in the literature, or to a combination of both? For the newly found sites it should be listed whether the binding TF was identified and (if so) what it is. For the fourth column, “literature binding sites”, it is not clear whether this is the total number of known literature sites or only the number of known sites that were successfully identified here – both should be listed. It is not clear what the fifth column “identified binding sites” refers to. The evidence column is also unclear. Is this evidence for the newly found sites or the literature sites?

Just to illustrate the confusion: consider the well-known lac operon. In Table 2 the *lac* operon (called lac as opposed to *lacZYA*, which is used elsewhere in the manuscript) is reported to have only 1 known literature site, whereas it is well established that it has a site for CRP and multiple sites for LacI. In the final column it is claimed that there is mass-spec evidence for LacI binding. However, on the website, the *lac* operon does not occur at all and in Appendix 2—figure 1 only the CRP site is reported. What makes this all even more confusing is that none of the experimental conditions included lactose or another inducer of the operon. As such, the lac operon should be highly repressed and so we expect very little effect of mutating the CRP site, and strong effects of abolishing the LacI site.

In multiple cases in the table, less literature information is reported than it actually exists: e.g. tar is a reported target of FNR, *uvrD* is a reported target of lexA, *cra* is a reported target of PhoB, and so on. The authors should make clear what literature information is shown here.

The authors should clear up this information so as to make it unambiguous and consistent across tables, figures, and website.

---

## [Author Response]

Essential revisions:1) The approach based on the mutual information profile to identify the binding sites within a promoter and to classify them as activator or repressors should be better explained:1.1) Since this is such a key step, the authors need to provide more details about how the manual choice of activator/ repressors works. Overall, it is unclear how one goes from mutual information profiles to deciding whether known sites are or are not identified, how to decide which regions are putative sites, whether they fit known motifs, whether they are RNAP sites, and how to construct an energy matrix for each putative site.

We thank the reviewers for this comment and completely agree with both its spirit and content. As such, we have taken this opportunity to expand and clarify our discussion of the data analysis pipeline and how it leads to regulatory hypotheses. First, it is important to note that the information footprints used in this paper are tools for hypothesis generation, and the Reg-Seq approach is further complemented by the use of expression shifts (a signed quantity which provides other helpful hints) about possible binding site locations. Deciding which regions are putative binding sites depends on the value of the mutual information of the base pairs within the putative binding sites. The process by which we calculate mutual information can be found in Appendix 3 subsection “Information footprints”. In this part of the response, we focus on manual curation, the process by which we identify potential transcription factor binding sites by hand, and later take up the subject of automated identification of transcription factor binding sites as suggested by the reviewers.

In regards to manual curation, we used our experience in looking at gold-standard genes – genes with well-studied, experimentally-determined regulatory architectures (as indicated in EcoCyc or RegulonDB) – from a previous study conducted by our group (Belliveau et al., 2018). Our analysis of these genes, which include *lacZYA* and other well-studied promoters, provided us with a sense of the “threshold value” of the mutual information that indicates the presence of a binding site. As a result, as noted at the beginning of this response, a key additional tool that we have introduced and that has been a critical part of the manual method of identifying putative binding sites is the use of expression shifts, a signed version of the information footprint that is generally associated with increases in gene expression (corresponding to the presence of a repressor binding site) and decreases in gene expression (corresponding to presence of an activator or RNAP binding site). Because expression shifts are a signed quantity, boundaries between binding sites are more easily identified. For future iterations of the Reg-Seq method, we have every intention to explore automated curation of expression shifts as a tool to identify candidate binding sites as we highlight below.

During manual curation of binding sites, we also disqualify any binding sites where there are only 3 or fewer base pairs with high values in the mutual information footprint. The logic behind this decision is that individual bases with very high mutual information can potentially indicate that a putative binding site is only active when a certain mutation occurs. In turn, the binding site would not be active in wild-type conditions. To explain why this is, consider that a typical binding site mutation, at any given base pair, will significantly *weaken* the binding site of interest. Therefore, each of those mutated base pairs is said to have a “large effect” on expression. For a very poor binding site that is not active in the wild-type case, most mutations will further weaken a site which already will have only a minor effect on gene expression. However, for a small number of base pairs, a mutation can occur that makes the DNA bind more tightly to the TF, making it relevant for gene expression. Therefore, in the case of an extremely weak binding site that is not relevant in the wild type condition, there can still be a small number of highly informative bases. We integrate these expanded details into a new subsection in the Materials and methods section of the main text that is entitled “Manual selection of binding sites”.

While there was no hard cut-off for mutual information values during manual curation, inspired by the reviewer comments, we have undertaken another round of automation, a schematic for which is presented in Figure 11 and is to be explained in more detail in another reviewer comment. To automatically identify putative binding sites from information footprints, we developed a strategy, which is as follows: we first take a 15 base-pair window where the average mutual information value is greater than 2.5×10−4 bits. The cut-off value was chosen based on examination of a test set of genes, a fuller discussion of which can be found in another reviewer comment. Any cluster of 15 bp that meets the information value threshold is then classified as a putative regulatory region, and can then be used to identify putative RNAP, activator, and repressor binding sites. We classify binding sites as those for activators or repressors based on whether mutation of bases at a given site causes expression to increase (indicating a repressor binding site) or decrease (denoting an activator/RNAP binding site).

Specifically, to identify RNAP binding sites, we compare the sequence preference (through energy matrices and sequence logos) to experimentally validated examples of RNAP sites. We have examples of energy matrices for the *_σ_*^70^ RNAP site from Belliveau et al., 2018. For energy matrices of other *_σ_* factor binding sites, such as *_σ_*^32^ and *_σ_*^28^, we use energy matrices generated from within the Reg-Seq experiment itself. For a *_σ_*^32^ binding site, for example, we used the example from the *hslU* gene. For a *_σ_*^28^ binding site, we used the energy matrix generated from the *dnaE* gene. When performing manual curation of binding sites, we visually compare the sequence logos of the example RNAP binding sites to the sequence logos of putative binding sites. We have also included a discussion of the method to create energy matrices and compare them to known motifs in Appendix 3 subsection “Energy matrix inference” and Appendix 3 subsection “TOMTOM motif comparison”, respectively. Further, a detailed discussion of energy matrix construction is provided in the GitHub Wiki that accompanies this work.

We really hope that the reviewers understand our goals. The information footprints should not be viewed as the key outcome of this work; we see them, rather, as a useful visualization tool that helps us generate hypotheses about the regulatory architecture for a given promoter. These hypotheses are allied with, and bolstered by, regulatory cartoons and energy matrices, which serve as a jumping off point for the kind of quantitative dissections we have carried out using chemical master equations and thermodynamic models in conjunction with careful promoter by promoter analyses during previous studies in our group (see Laxhuber et al., 2020; Amit et al., 2011; Phillips et al., 2019; Razo-Mejia et al., 2018; Rydenfelt et al., 2014a; Weinert et al., 2014). We share the reviewer’s concern in regards to scaling up the Reg-Seq methodology, but believe that it is not enough to produce single base pair resolution graphs such as the information footprints (and their analogs in the work of others), and are striving instead to produce a systematic approach to parlay those plots into actionable, regulatory hypotheses.

1.2) It is mentioned (subsection “Analysis of sequencing results”, first paragraph) that, to confirm these manual choices, authors computationally identify regions of activators / repressors by assessing the fold change in gene expression due to mutations. It is unclear how the authors are assessing the significance of the changes, i.e., what the underlying null model is. If this computational approach is statistically sound, why the need for the manual approach? These manual choices are worrisome and raise concerns about the feasibility to scale-up this method, as intended by the authors.

We appreciate the reviewer’s suggestions and the opportunity to improve our computational methods in this resubmitted manuscript. We hope that the reviewers will appreciate that we have been engaged in a staged scale-up of our approach to dissect the regulatory genome. During each generation of experiments, we have been upgrading all of our methodologies in an effort to develop tools that, eventually, will help us fully elucidate the so-called “regulome”. We detail the improvements that we have made below.

In light of the excellent suggestions of the reviewers, we have implemented automated motif finding, which replaces the appeal to p-values as explained in this study. The use of automated motif finding replaces our initial focus on p-values, replacing it with an appeal to our ability to recover the regulatory architectures of certain “gold-standard” promoters as a metric of the soundness of our results. The details of this new automated method are discussed in the Materials and methods subsection entitled “Automated putative binding site algorithm”.

Despite the advancements (and shortcomings) of the Reg-Seq method, we still hope that the reviewers are aware of the state of the art in the field of bacterial regulation and the important advances that we have made even without a genome-scale effort. During each round of new experiments, we want to materially alter the completeness of the EcoCyc and RegulonDB databases such that researchers interested in particular promoters have a way forward to design and perturb those promoters. Indeed, the group of Collado-Vides (see Santos-Zavaleta et al., 2019,) in Mexico, which manages the RegulonDB database, has already taken the results of the unpublished work in this manuscript that appeared on BioRxiv, contacted us, and updated their database accordingly. Our hope in the coming few years is that each and every promoter will have a full regulatory cartoon and corresponding energy matrix but, for the moment, we are excited about two successive 10-fold scaleups in our work, and hope that the reviewers acknowledge this as significant progress (Belliveau et al., 2018).

2) It is difficult to assess the false positive/negative rate of identifying binding sites: This is partly due to the lack of clarity about the authors' approach in deciding whether a known site was identified or not, partly because the annotation of known sites is unclear and incomplete, and partly because the statistics are not transparently reported in the manuscript. It is therefore important that the authors present an objective method for assessing what fraction of known sites is not recovered, for what fraction of predicted novel sites a motif can be made, and for what fraction of those the mass-spec successfully recovers a binding TF.As a suggestion: The authors should start from a unambiguously defined set of sites (e.g. from RegulonDB and EcoCyc) and then have an unambiguous procedure for:i) Calling segments where a site exists (for example based on the summed mutual information within the segment).ii) Calling the motif by comparing the profile with PSWMs or energy matrices for known motifs.After that, they should clearly report what fractions of sites are recovered and for what fraction of sites the correct motif is predicted.

We thank the reviewer for this comment, which is an extremely serious consideration that we hope to fully address in this resubmitted manuscript. We note that, with each successive generation of our experimental efforts, we are expanding our ability to scale-up the methods. The manual curation of binding sites has been used as a tool for hypothesis generation that we have followed by independent confirmation of these hypotheses. That said, we are very interested in developing robust, scalable approaches.

In this resubmitted manuscript, we introduce a systematized way of identifying the locations of binding sites, as shown in Figure 11, that allows the false negative and false positive rate of binding site identification to be clearly assessed. This new approach also replaces the previous arguments based on p-values and offers an alternative method to assess the accuracy of our method. Thanks to the reviewer’s suggestions, we find this new treatment more intuitive and transparent, and think it has substantially improved this manuscript.

For a given information footprint, we average over 15 base pair “windows”. We then determine which base pairs are part of a regulatory region by setting an information threshold of 2.5×10−4 bits (the rationale for this threshold is explained below). All base pair positions that pass the information threshold are then joined into “regulatory regions”, which we consider to be “activator-like” (if a mutation decreases expression) or “repressor-like” (if a mutation increases expression). This means that it is possible to identify overlapping repressor and activator binding sites. We join any base pair positions within 4 base pairs of each other into a single regulatory region. We then find the edges of each binding site region by trimming off any base pairs at the edge that are below the information threshold (even if the 15 base pair average is above the threshold). While we can often resolve overlapping or nearby repressors from activators, a limitation of this method of identification is that is cannot resolve two activators or two repressors that are very close to each other or overlapping.

To determine which information threshold to use as a cutoff for a putative binding site, as displayed in Figure 11, we selected a training set of genes which included two of our ”gold standard” genes with transcription factor binding sites which have experimental validation, DgoR (the upstream site from the *dgoR* promoter) and CRP (from the *araAB* promoter), two genes with only RNAP binding sites, including *hslU* (under heat shock) and *poxB*, and several genes that we classified as inactive, wherein no RNAP binding sites or other binding sites could be identified. These inactive genes included *hicB, mtgA, eco, hslU* (without heat shock), and *yncD*. The growth condition (heat shock) is specified for the *hslU* promoter as transcription occurs from a *_σ_*^32^ RNAP site, which will be inactive except during heat shock. We selected the threshold such that the RNAP sites and known binding sites were identified, while no binding sites were identified in the inactive regions.

We then determine a set of binding sites upon which to test this method and determine a false negative rate for the Reg-Seq experiment. In this set of binding sites, we include those sites which are “high-evidence” according to EcoCyc. Such “high evidence” binding sites have been validated experimentally with the binding of purified protein or through site mutation. Some “high-evidence” sites are excluded because they are not included within our 160-base pair, mutagenized sequence, or because they are not active in any of the growth conditions that we tested. Justifications for those binding sites which were not included are now listed in a new appendix; Appendix 4 subsection “Explanation of included binding sites”. A full list of promoters and binding sites that *were* included in the set of genes used to validate our automated binding-site finding algorithms are also provided in Appendix 2—table 1.

For each promoter contained in Appendix 2—table 1, we used the automated procedure outlined above and in Figure 11 to identify the activator and repressor binding sites. A visual display of the expected binding sites, the information footprints for the promoters in Appendix 2—table 1, and the discovered binding sites are all displayed in Appendix 2—figure 2 and 3. To assess the false negative rate, we summarize the identified regulatory regions to the known binding sites from Appendix 2—table 2. At this stage, we did not consider the identities of the binding sites; we merely consider their presence or absence. Inferred binding sites are declared to “match” the known binding site if the automated identification procedure classifies at least half of the base pairs reported in EcoCyc as belonging to a transcription factor binding site and correctly determines whether the binding site belongs to an activator or repressor.

We do not require exact matching of the edges of the binding sites for several reasons. One such reason is that, in some cases, the sequence of half of a binding site (for example, corresponding to one half of a helix-turn-helix binding motif) can contribute relatively little to gene expression, and so will not have high mutual information values in the corresponding information footprint for that binding site. While this may appear unintuitive for TFs where both sections of the binding site are bound by identical halves of a dimer, we see several examples of this in our Reg-Seq experiment results, including for CRP binding sites of the rspA promoter studied during our analysis of false negative rates. We can see in Appendix 2—figures 2 and 3 that the downstream half of the binding site is not identified as important for gene expression. If we examine the wild type sequence of the rspA promoter, we also see that, for the upstream half of the sequence, the wild type matches the five most conserved bases of the consensus sequence (TGTGA) perfectly. The downstream half of the sequence, however, has 4 mismatches out of 5 bases (CGGCT compared to the consensus sequence of TCACA). The downstream half of the binding site already binds to its target transcription factor poorly, so further mutations have little effect. While it is true that CRP binds to that sequence region, it is also true that CRP binds only extremely weakly to that section of the region. A similar effect can be seen in previous work from Belliveau et al., 2018, where a mutation in the downstream half of a CRP binding site in the xylE promoter had more than a 10-fold greater effect on binding energy than mutation in the upstream half of the binding site. As such, we are lenient when evaluating the successes of our algorithm in this regard. Furthermore, the methods that have been used to determine the presence of “high evidence” binding sites in the past, such as ChIP-seq, do not typically have base pair resolution with which to precisely determine the edges of binding sites (Skene and Henikoff, 2015).

Lastly, a known weakness of our algorithmic approach is that binding sites that are extremely close or overlapping cannot be distinguished from each other initially. For example, the XylR sites in the xylF promoter are only separated by 3 bases according to RegulonDB. While the sites can be distinguished upon later investigation through gene knockouts, mass spectrometry, or motif comparison, our initial algorithm joins the sites into one large site. While this is a weakness of the algorithm, for our purposes it does not constitute a false negative, as the important regions for regulation are still discovered. All regions for all promoters that are classified as regulatory regions, their identities as activators, repressors, or RNAP binding sites, as well as their starting and ending base pairs, can be found in Supplementary file 3. Furthermore, we summarize the success and failures of the method at each binding site in Appendix 2—table 2.

We see in Appendix 2—table 2 that 11 of the 15 promoter regions included in Appendix 2—table 1 have all TF binding sites classified as putative transcription factors, two have the majority of sites correctly classified, and two do not have any of their binding sites correctly classified as regulatory elements. We can see the information footprints used in the correct identifications in Appendix 2—figures 2 and 3. Considering binding sites, we find that 23 out of 33 binding sites are correctly classified. However, we argue that the false negative rate should be considered on a per promoter basis, rather than on the basis of individual binding sites. The reason for this argument can be seen in the two “worst” cases of correct binding site identification; namely, for the *araC* and *dicA* promoters.

The *araC* promoter is repressed by multiple repressor binding sites in all growth conditions tested. *araC* only has high expression transiently after addition of arabinose (Johnson and Schleif, 1995), and while growth in arabinose is utilized in this experiment, RNA was not collected during the window of high expression. The case study shows that Reg-Seq does not perform well when many repressor sites regulate the promoter. Reg-Seq relies on mapping the effect on expression of mutating a particular site, and when many strong repressor sites are present, expression change will be minimal unless all repressor sites are weakened through mutation. Additionally, in this highly repressed case, the RNAP binding site we observe in the mutagenized region is not the documented RNAP site in RegulonDB, indicating that we are seeing transcription primarily from an alternative TSS. Different RNAP sites are often regulated differently, and in this case, the presence of an alternative and dominant RNAP binding site (in the repressed case), likely contributes to a failure to observe six of the seven binding sites in the *araC* promoter. Similarly, in the *dicA* promoter, we did not find an RNAP binding site in the studied region, which would make it very unlikely for any transcription factor binding sites to be identifiable.

We next take the reviewer’s suggestion to compare the energy matrices from putative regulatory regions to known binding site motifs. The known motifs are obtained either from RegulonDB or are generated from data from prior Sort-Seq experiments (see Belliveau et al., 2018). We utilize the TOMTOM motif comparison software (Gupta et al., 2007) to perform these comparisons. TOMTOM generates a p-value under the null hypothesis that the two compared motifs are drawn independently from the same underlying probability distribution. We test 95 motifs against each target motif that we are attempting to identify. The 95 resulting p-values (for each target) generated by TOMTOM are displayed in Appendix 2—figure 4. A full discussion of TOMTOM can be found in Appendix 3 subsection “TOMTOM motif comparison”. We only included those transcription factors that either have over 50 known binding sites in RegulonDB or have experimental measurements of binding site preference, such as in Sort-Seq (Belliveau et al., 2018). As such, we used TOMTOM on the XylR, CRP, MarA, MarR, and RelBE sites in Appendix 2—table 2. We utilized a p-value cutoff of 0.05, corrected for multiple hypothesis testing. 95 motifs were tested against each target and using the Bonferroni correction leads to a p-value cutoff of ^0.05^_95_ = 5 × 10^−4^. In Appendix 2—figure 4 we show that the correct transcription factor falls below the p-value threshold in all cases. For the CRP binding site in the *lacZYA* promoter, FNR also falls below the cut-off, but CRP has a calculated p-value that is ≈ 6 orders of magnitude lower, and so is clearly identified as the correct binding site. The results show that motif comparisons can be used reliably in those cases where we have high-quality energy matrices for comparison.

In order to determine false positive rates, we test against promoters for which we are certain there are not additional, unannotated binding sites. Most known binding sites were not determined using a method like Reg-Seq, which looks for regulatory elements across an entire promoter region at base pair resolution. Rather, many efforts to pinpoint TF binding site locations use assays like ChIP-seq, which prioritizes looking for all binding sites of a given TF across the entire genome. For those promoters studied with Reg-Seq, there are five promoters for which we have reason to believe that there are no undiscovered binding sites. There is evidence that the *zupT* promoter is constitutive (Grass et al., 2005), and the *marR*, *relBE*, *dgoR*, and *lacZYA* promoters have all been examined for binding sites at base pair resolution previously in the Sort-Seq experiments (Belliveau et al., 2018;Kinney et al., 2010).

To evaluate false positive rates, we examine the putative activator and repressor binding sites as identified using our automated methodology (described previously), and compare any known binding sites to the known binding sites for the target promoters. We also classify any putative regulatory regions that are outside of known TF binding sites as false positives. Similarly, any identified RNAP binding sites which were outside of the known RNAP binding locations were classified as false positives. In the *zupT* promoter, only the correctly placed RNAP site was identified. There were similarly no false positives identified in the *marR*, *relBE*, *dgoR*, or *lacZYA* promoters. We have added this discussion to Appendix 2 in a new subsection entitled “False positive and false negative rates”.

3) Energy matrices:3.1) How are the energy matrices constructed? Details are missing from the manuscript.

We thank the reviewer for this comment and concur that additional information on our construction of energy matrices is necessary. We have added a general overview of our method for constructing energy matrices to the Materials and methods section of this manuscript and further detail can be found in Appendix 3 subsection “Energy matrix inference”. To ensure that readers of this work are able to pick up and use our methods in their own laboratories, we have also paid considerable attention, in this resubmitted manuscript, to the construction of a full, detailed set of experimental, mathematical, and computational discussions. Detailed instructions to build energy matrices can be found on the GitHub Wiki for this work, while associated code for building energy matrices can be found in our Jupyter notebooks.

To address the reviewer’s concern and clearly delineate how we built energy matrices in this work, we note that the plotting of an energy matrix for a regulatory region of interest begins by analyzing the next-generation sequencing data for each 160 bp mutagenized region included in our DNA libraries. In particular, we deduce the relative gene expression value for each mutated regulatory region by counting genetic barcodes in RNA and cDNA form. This approach, whereby genetic barcodes serve as a semi-quantitative metric for gene expression, is frequently used in a type of experiments known as “massively parallel reporter assays” (MPRAs). We count barcodes as a relative measure of gene expression values for every mutant sequence and infer the energy matrix that best describes the data by maximizing the mutual information between the expression predictions from a given energy matrix and its expression data. We do this by starting from a random energy matrix and using a Markov Chain Monte Carlo method following the procedure outlined in Kinney et al., 2010. An energy matrix with high mutual information would predict (if the binding site corresponds to a repressor) that the sequences with low expression would have strong binding of said repressor, while sequences with high expression would have weak binding by the repressor.

The output from our MCMC inference is a table of the form:

**Author response table 1. resptable1:** 

**pos**	**A**	**C**	**G**	**T**
0	-0.005387	-0.011758	-0.010176	0.027322
1	0.002338	0.049826	-0.058030	0.005866
2	-0.000259	-0.037224	0.008021	0.029461
3	-0.017494	0.015760	-0.012184	0.013918
…	…	…	…	…

In Author response table 1, each value represents the binding energy contribution for a specific nucleotide: A, C, G, or T. A negative value indicates that the nucleotide in that specific position confers a stronger binding energy with the transcription factor, while a positive value indicates a weaker binding energy. We next visualize these nucleotide-specific values using a simple Python script executed in Jupyter notebooks. The notebook and all code used for energy matrix visualization is available in our GitHub repository. Again, we are grateful to the reviewer for raising this concern and agree that our methods needed to be made clearer. We also note that we have updated our Materials and methods section, in the main text, to immediately state that “the reader is urged to visit our detailed GitHub repository and associated methods on our GitHub Wiki”.

3.2) How are the p-values estimated to assign significance to energy parameters? It appears that the authors have used MCMC to sample from a likelihood function, but the details are very vague. To define an energy, it is assumed that the average expression can be written as the negative exponential of some effective energy to parametrize the average effects of mutations on expression- this should be better justified in the text. However, it seems that instead of 4 different energy parameters at each position, the model only assumes a “mutant” and a “wild type” energy. This may not be a good approximation given that we know nucleotides within TF binding sites have highly varying degeneracies. Moreover, the likelihood function for the observed DNA and RNA reads should be expressed as a function of the energy parameters at all positions. However, this is not the approach taken by the authors and instead, equations (8) and (9) only look at single positions. So, it is not clear what kind of likelihood function the MCMC is sampling from, or how p-values and confidence intervals are determined. Also, if the authors are indeed using MCMC with some likelihood function to sample the space of possible parameter values, the resulting variation in the energy parameters should correspond to posterior probability intervals and not confidence intervals.

We thank the reviewer for the comment. We are afraid our p-value analysis was confusing since there is no such analysis associated with energy matrices. The information footprints, for which we calculate effective energies and p-values, and the energy matrices involve separate calculations. Furthermore, we would like to note that the quantitative measures created during Reg-Seq, the energy matrices of transcription factors or RNAP sites, do in fact consider all four bases separately for each position. This is again in contrast to the calculations of effective energies for information footprints, which are based on calculating parameters for only mutated or wild type bases.

We acknowledge that, by only considering wild type or mutated energy contributions to the total effective binding energy rather than having separate values for energy contributions from all four base pairs, our methods will not be accurate in the case of calculating mutual information at locations with degenerate base pairs. However, the information footprints, which are created from the fits to effective energy of wild type or mutated base pairs referred to in the reviewer comment, are intended to be hypothesis generation tools that can identify transcription factor binding sites, rather than a quantitative tool like the energy matrices for transcription factors that we generate. As such, the most important test for the assumption that we can approximate effective energy contributions from all 4 bases as contributions from only wild type or mutated bases is to assess whether the approximation has any effect on determining binding site locations. We re-ran the false positive and false negative assessments discussed in the response to major reviewer comment 2, but instead calculated the effective energy parameters for producing information footprints as a sum of contributions from all four bases. We find that the literature binding sites that were properly identified, as summarized in Appendix 2—table 2, are identically identified. Specifically, any site which was identified using the previous method is still identified and any site that failed to be identified is still not observed. Similarly, when we only fit effective energy parameters for mutated or wild type bases there are no false positives identified in the promoters for *marR*, *relBE*, *dgoR*, *zupT*, or *lacZYA*. There are also no false positives when repeating the procedure while considering all 4 bases in the effective energy fits, implying that the simplification to only considering mutated or wild type bases does not have an effect on our ability to identify binding sites. We have amended the text in Appendix 3 subsection “Information footprints” to call attention to the limitations of the assumption we make and to justify our assumptions regarding effective energy used when inferring the parameters used in creating information footprints.

We now refer explicitly to the likelihood function used during MCMC inference in Appendix 3 subsection “Energy matrix inference” in Equation 27.

3.3) Could the authors present the accuracy of the inferred energy matrices as a function of the number of genomic variants used? A particular example is currently given in Appendix 3. Can the authors use a simulation for that purpose? This is required to understand the validity of the results and the ability to scale-up such an experiment (if for example a lower resolution is sufficient, can larger regulatory regions be analyzed in future experiments?)

We appreciate the reviewer’s comment and, in this resubmitted manuscript, have expanded the analysis that was carried out in Appendix 3 to fully address the issue of how the number of mutated sequences for a given promoter affects inference of energy matrices. We wish to both prove that our results will not be strongly dependent on the number of sequences in the sequence regime that we are in, and also to explore, as the reviewer helpfully suggested, how a reduction in the number of mutated sequences could facilitate larger scale experiments in the future. While pure simulation of data from an energy matrix is a possible way to answer this question, we instead generated examples of smaller data sets by computationally sub-sampling the Reg-Seq data from 7 mutated promoters (*maoP*, *hslU*, *rpsA*, *leuABCD*, *aphA*, *araC*, and *tig*). These promoter regions are representative of a large cross section of the variety of transcriptional regulation that we see in our study, including constitutive expression (*hslU*), simple repression (*leuABCD*, *tig*), simple activation (*aphA*), and more complicated regulatory architectures (*maoP*, *rspA*, *araC*). Each sub-sampling was performed in triplicate, and we used the Pearson correlation coefficient as a comparison metric between the inference based on the full data set and the computationally sub-sampled data sets. The results of this analysis are displayed in Appendix 3—figure 1.

Based on this analysis, we find that there is room to moderately lower the resolution of the experiment to approximately 1000 unique, mutated sequences per 160-base pair “window” before our results are subjected to large deviations in the inference of energy matrices. A decrease in the number of unique sequences can give modest boosts to the number of genes that can be studied in our Reg-Seq experiments but would not, alas, be able to provide order of magnitude increases in the number of genes that can be explored. We have replaced the original discussion in Appendix 3 with this expanded analysis, which we hope will fully address the reviewer’s concerns.

4) The step for assigning regulatory logic to promoters is unclear. Do authors always assume OR-gate-like interactions between the TFs, which is what Equation 14 suggests? If so, what is the justification for that assumption? Is it just the most basic statistical mechanical assumption, or is there strong empirical support?

We appreciate that the reviewer has raised this question. First, we reiterate that the approach that we present in this study, if we as a field are being appropriately cautious, should be viewed as a powerful, quantitative tool for regulatory hypothesis generation. But it should be clear, based on many papers from our lab, that we are by no means satisfied with stopping at the current iteration of this method (Phillips et al., 2019; Razo-Mejia et al., 2018; Garcia and Phillips, 2011; Chure et al., 2019). The outcome of our study is a hypothesized regulatory architecture as characterized by a suite of binding sites for RNAP, repressors and activators, as well as the extremely potent binding energy matrices. We do not assume, *a priori*, that a particular collection of such binding sites is AND, OR, or any other logic (Buchler et al., 2003; Bintu et al., 2005; Galstyan et al., 2019). We are exploring a new round of experimental tools that will allow us to do the next step in our analysis of such architectures. Our new Figure 4, in the main body of this paper, shows the outcome of our analysis. For those promoters that contain multiple transcription factor binding sites, we do not yet know what detailed regulatory logic (AND, OR, NAND, and so forth) they imply. Using the traditional approach in our lab of using statistical physics to predict input-output functions, in conjunction with our methods for constructing energy matrices, we believe that there is a way forward to find out what the regulatory logic of a given promoter is. We also suspect, however, that information footprint and expression shifts under multiple growth conditions might produce substantial hints as to the logic as well. Ultimately, we believe the present work lays the foundation for asking and answering precisely these questions. To more fully address the reviewer’s concern, and our noted inability to identify regulatory logic, we have included a discussion of inferred regulatory logic based on this comment in the Results subsection entitled “Newly discovered *E. coli* regulatory architectures”.

5) The description of the method and analysis is hard to follow and unclear about many assumptions and limitations. To give a few examples:5.1) Appendix 3 outlines how to calculate the mutual information footprint from the read count data but a more explicit discussion would be helpful: For example, how should the quantities in Equation 5 (p(m, m𝜇), p(m), p(𝜇)) be calculate from the preceding example data (subsection “Information footprints”)?

We thank the reviewer for their comment and agree that, as written in the original manuscript, our mathematical explanations are insufficient to guide the reader through our equations in a logical manner. We sincerely hope to address this shortcoming in this resubmitted manuscript. Accordingly, we have expanded our discussion regarding the calculation of information footprints in Appendix 3 to include explicit calculations of *p*(*m*), *p*(*µ*), and *p*(*m*, *µ*) from the example data presented in Appendix 3—table 1.

In the subsection entitled “Information footprints” in Appendix 3, we now give a very precise, pedagogical example of the calculations for *p*(*m*), *p*(*µ*), and *p*(*m*, *µ*), and provide further explanations regarding calculations of the mutual information itself. This Appendix, in conjunction with the new GitHub Wiki and Jupyter notebooks, also now show how these calculations can be performed on real data obtained using the Reg-Seq methodology.

5.2) The methods rely on both DNA and RNA sequencing of the same sample and using ratios of RNA/DNA reads from the same reporter to estimate expression. The description of the sequencing protocol is insufficient. Importantly, the DNA sequencing is not even mentioned in the Materials and methods and it is not explained how precisely this expression quantification is done.

We thank the reviewer for their comment and apologize for the error. We have now updated our Materials and methods section to more fully explain how our DNA libraries are prepared, how we isolate RNA and DNA, and how our sequencing experiments are performed. We have also added several sentences that describe how we count barcodes, in cDNA and DNA form, and then use these data to quantify gene expression values for mutated promoter sequences. In addition, the GitHub Wiki provides a detailed description of these methods.

5.3) The Materials and methods mention correcting for correlated mutations, MCMC to assign p-values, and energy matrix reconstruction, but it is completely unclear how either of these are done.

We apologize that our Materials and methods failed to mention the corrections for correlated mutations, MCMC to assign p-values, and how we construct energy matrices. We take this exclusion very seriously, as it is our sincere hope that readers and other laboratories will be able to easily utilize Reg-Seq to dissect a greater number of regulatory architectures. Since our first submission, we have made an in-depth GitHub Wiki that outlines all experimental, computational, and mathematical steps involved in our experiments. However, we do not merely wish to direct readers to an external GitHub repository, as this exclusion warrants remedy in the main body of the text. Accordingly, we have added a subsection in Appendix 3 entitled “Energy matrix inference” to explain the process of creating energy matrices. In light of the suggestions of the reviewers, we have also implemented a method for automated motif finding that replaces the appeal to p-values. We now explain this process in a subsection entitled “Automated putative binding site algorithm” in the Materials and methods section. The generation of mutual information footprints that we use to address the problem of correlated mutations is also now addressed in Appendix 3, in the subsection entitled “Information footprints”.

6) The discussion of the methods in the main text is mostly qualitative and crucial details are scattered throughout the manuscript, in the appendices and in Materials and methods.6.1) For example, it requires digging into the appendices to understand that the authors study a region of 160 bp mostly upstream TSS, and so, is this method inadequate if the regulatory region is larger or if its location is unknown? A clear and concise explanation of the crucial details (e.g. size of the region, number of variants per promoter, etc) should be given in the main text.6.2) To further clarify the methods, it would be helpful to include a picture of the reporter construct, which based on the descriptions includes:-promoter including 45 bp downstream of annotated TSS.-then 64bp with primers for plasmid construction-then 11bp with stop codons in 3 frames.-then barcodes?-then a RBS-then GFP mRNA

**Author response image 1. respfig1:** Figure 10 from Rydenfelt et al., 2014b. Distribution of activating and repressing binding sites bound by global TFs and specific TFs, respectively. The y-axis shows the number of binding sites overlapping each nucleotide position, after aligning all promoters with respect to their transcription start site (TSS) for the different kinds of TFs.

We thank the reviewer for this constructive comment and apologize that the methodology in our initial submission was unclear and insufficient. In this resubmitted manuscript, we have made numerous efforts to address these concerns.

We have created an exhaustive Materials and methods section that outlines every experimental, mathematical, and computational procedure utilized in this project. The full text is available on the GitHub Wiki that accompanies this project. Additionally, we now include a spreadsheet that lists every gene/operon that was studied in our manuscript (provided as Supplementary file 1). For each gene/operon, we also provide details on *why* each TSS was selected and provide applicable references for each choice. We hope that this provides clarity on our methodologies. Finally, we have updated our methods, in both the main text and appendix, to more clearly disclose our choice of TSS for each gene/operon and the design of our genetic constructs.

When we generate a library of mutated “regulatory regions” for Reg-Seq, we include a 160 bp region that includes 45 bp downstream of each TSS, and 115 bp upstream of each TSS. It is certainly possible, as noted by the reviewer, that a 160 bp region does not include all of the TF binding sites and regulatory information for a given gene/operon. In cases where little is known about the genetic architecture for a gene, and no TF binding sites have been experimentally determined, we cannot say definitively that our selected region includes all regulatory information.

A previous study from our group (Rydenfelt et al., 2014b) (see Author response image 1) asked the question: What is the distribution of binding site locations for activator and repressor TFs in the *E. coli* genome, based on a computational analysis of the RegulonDB database? In that study, as seen in Author response image 1, we found that most (but not all) TF binding sites fall within the range of approximately 45 bp downstream and 115 bp upstream of a TSS. Repressors tend to bind further upstream of a TSS than do activators. We used this study as the basis for designing the size and positioning of the 160 bp regulatory ”windows” used in this study, but of course, we are fully cognizant of the broader distribution revealed in Author response image 1, and in addition, of the even more thorny issue of action at a distance in the form of DNA looping. In future iterations of our experiments to dissect the *E. coli* ”regulome”, we aim to broaden the length of our mutagenized region, a goal which poses few conceptual or technical obstacles, though the question of DNA looping (and enhancers in eukaryotes) will clearly require further innovation.

For each gene, we obtained an average of 2200 unique DNA mutants in the form of oligo pools from TWIST Biosciences. This number was found to be sufficient for the replicable inference of binding energy matrices based on a previous systematic, experimental effort to evaluate the minimum number of DNA mutants that are necessary for such inference; we have previously discussed this in the context of Appendix 3—figure 1. We also now include our discussion of these points in Appendix 3 subsection “Uncertainty due to number of independent sequences”.

When we choose a TSS to study based on peer-reviewed literature and our search through the EcoCyc database, we do so using a few criteria; we always select, where possible, a TSS based on experimental evidence. If there are multiple TSS listed for a gene or operon of interest, we typically select the TSS that is best positioned to include regulatory information for all of the nearby TSS. For full transparency, however, we now include a spreadsheet that provides our rationale for selecting the TSS for each and every gene in this study (provided as Supplementary file 1).

To better disclose the design of our genetic constructs, we have also created a “cartoon” figure of the cloned plasmid architecture, which we show in Appendix 1—figure 1. As noted by the reviewer, our reporter construct is a pSC101-based plasmid with a kanamycin resistance marker. Each mutated, 160 bp DNA sequence is flanked by primer sites that are 20 bp in length (forward and reverse sites). Immediately downstream of the cloned regulatory regions is a random, unique, 20 bp barcode, followed by a ribosome binding site, GFP gene, and terminator. We have added clarifications on our genetic constructs to the accompanying figure legend in Appendix 1. Additionally, exact details of the sequences flanking the mutated region are stored in Supplementary file 2.

7) Authors used TSS information to decide which promoters to pursue. How limiting is this for scaling up the procedure? It would be helpful to discuss briefly what fraction of promoters in E. coli have good TSS information and how valid is to assume that at the genome-wide level, each operon is dominated by a single TSS?

We appreciate the reviewer’s comment and aim to address these concerns in this resubmission. We argue that the TSS information available in public databases is a significant limitation in the scaling up of this work that will require further ingenuity and vigilance on our part.

To explore this assumption, we downloaded the ”all promoters” dataset from RegulonDB v10.5 (Santos-Zavaleta et al., 2019), which provides information on every promoter in *E. coli*, with details on each promoter’s name, its DNA strand orientation, *σ* factors that recognize the promoter, the promoter sequence, evidence to support the existence of each promoter, and the confidence level of the evidence. An examination using Python indicated that, of the 2600 operons listed in RegulonDB, 2500 have a TSS known to be related to the operon or to have a TSS, determined by transcription initiation mapping, within 300 bp of the start of the operon. If we restrict our consideration to those TSS classified as having “Strong” or “Confirmed” evidence by RegulonDB, then 1700 operons have an associated TSS. In most cases, the evidence for a promoter with this classification is experimental; namely, the TSS was identified using transcription initiation mapping or RNA- seq experiments. The number of operons with a known TSS, as listed in these databases, represent a more than 10x increase over the number of genes explored in our Reg-Seq experiments and thus will serve as a powerful jumping off point for future iterations of this work. Unfortunately, EcoCyc and RegulonDB are not comprehensive, and thus will still not permit a complete genome-wide treatment of the problem.

The reviewer has also asked whether it is safe to assume, at the genome-wide level, that each operon is dominated by a single TSS. This is certainly not an assumption that we make in this study, as many genes/operons in *E. coli* have experimental evidence for multiple TSS, meaning that there are sometimes two or three promoters that are active upstream of certain genes or operons. Indeed, one of the limitations of the Reg-Seq method is its limited DNA “mutational window”; we are only able to build mutated oligo libraries for DNA fragments up to approximately 200 base pairs in length. This is a limitation that comes down to modern oligo synthesis technologies and financial feasibility, as ordering about 2000 unique sequences for 100 genes, each of which is about 200 bases in length, is already a costly endeavour. Thus, we do our utmost to include all known or putative regulatory information in the DNA sequences that we do order. If there are multiple TSS for a given gene/operon, we choose the TSS with the most experimental evidence, or the TSS that is located most closely to the others. In this resubmitted manuscript, we now include a spreadsheet as Supplementary file 1 that lists our rationale for choosing every TSS in this study.

8) The authors rely on expression from plasmids and use mRNA/DNA ratio to handle the effect of variability in plasmid copy number between cells. However, if the plasmid copy number is of a similar order of magnitude as the transcription factor copy number, then the expression level measured (to calculate the energy matrices) is determined not only by the binding energy, but also by the TF availability leading to under-estimation of the binding energies. The authors should comment on this in the manuscript, and if they have the data available, show the measurements for plasmid and TF copy numbers to address this point. At this point we do not see a necessity for additional experiments.

We are grateful to the reviewer for this comment, and certainly agree that this point demands serious consideration and scrutiny. Indeed, one of our favorite works from our laboratory was precisely aimed at rigorously predicting and measuring this effect, regarding protein and plasmid abundance (see Weinert et al., 2014). In that paper, we demonstrated how to control this effect in a parameter-free manner. Further, we share the reviewer’s aggravation that, until now, our experiments have been performed on plasmids. We have been working hard with the Sri Kosuri lab to make sure that our next set of experiments will involve chromosomally integrated libraries, and so only a single copy of each mutated promoter will be present in future iterations of our experiments.

With that said, the plasmid used in our experiments is derived from pUA66, which contains a pSC101 origin of replication (Zaslaver et al., 2006). A study from 1997 quantified the copy number of various “replication origins” and found that pSC101 has a copy number (in log phase) of 3 or 4 (Lutz and Bujard, 1997). To assess this copy number, the experimenters encoded luciferase on plasmids with differing origins of replication, quantified the luciferase activities for each, and compared their obtained values to a strain in which a single copy of the luciferase gene was inserted into the chromosome. The authors used a different strain of *E. coli* (DH5*α*Z1) and we have not independently assessed the copy number of our plasmid.

The absolute copy number of various TFs identified in our study can also be found using data from prior studies, chiefly those that performed experiments to quantify whole-proteome *E. coli* samples. Specifically, our laboratory has long admired a 2016 study from Matthias Heinemann’s group, which provides the absolute quantification for roughly 55 percent of predicted proteins in the *E. coli* K12 proteome using LC-MS (see Supplementary Table S6 from Schmidt et al., 2016. For those TFs that were quantified in the Schmidt study, and also identified in our Reg-Seq experiments, we provide their absolute quantification in *E. coli* K12 for both glucose and LB growth media in Appendix 3—table 2.

This table lists the global, absolute quantification for the listed TFs for *E. coli* K12 grown in both glucose (5 g/L concentration in M9 minimal media) and LB. For most TFs, the purported copy number for nearly all of these TFs is much greater than the low-copy number plasmid used in this study, thus mitigating the concern that the limited availability of a TF could impact gene expression, though we reiterate that going forward, we are committed to moving beyond this approach to perform chromosomal integrations.

“There are a few transcription factors in this work that have copy number on the order of the plasmid copy number, however, including XylR, DicA, and YgbI. […] The rank-order is always preserved and so the presence of a multi-copy plasmid will not change the mutual information between model predictions and experimental data. As a result, the final inference of energy matrices will remain the same.”

Again, we agree wholeheartedly with the reviewer that this point merits serious deliberation and discussion. As this is a point that astute readers of our work will likely share, we have added a modified form of this explanation to Appendix 3 subsection “Effect on calculated energy matrices when transcription factor copy number ≈ plasmid copy number”.

9) The limitations of the method should be more concisely explained. Currently, limitations are scattered throughout the text.

We thank the reviewer for their feedback and have aimed to address this concern by updating the main text’s Discussion section accordingly. Specifically, we have identified the major limitations discussed in the Results section, and have added them to the Discussion section, so that all main limitations relating to the Reg-Seq methodology can be found in one place. We have also added numerous limitations to this Discussion section, including our concerns regarding the use of mass spectrometry to identify weaker TF:DNA interactions, and further discuss our efforts to ameliorate some of these limitations for future work and the transcription research community at-large.

Revision required for Figures and Tables:Figure 1: This figure is hard to read. It is difficult to distinguish the individual tick marks around the genome, because there are too many, they are too densely packed, and the colors are too mixed. Also, the caption describes the color of some ticks as "red," but in the printed figure they look more brown (They appeared closer to red on the screen.)

We thank the reviewer for their comment and agree that the lines on this figure are difficult to distinguish. When making this schematic, we wanted to emphasize the known and unknown regions of the *E. coli* genome and did not aim to encode precise information about their positions. The percentages in the center serve to summarize the main point: that the majority of operons have no known regulation. As a result, we have opted to leave the figure unchanged, as it bolsters our argument concerning our “regulatory ignorance” of the *E. coli* genome. However, to address the reviewer’s concern with respect to gene positions, we now include (in Supplementary file 1) the genomic location of each gene’s TSS.

Figure 2: You might want to clarify that the blue region likely corresponds to the σ factor binding site.

We thank the reviewer for noticing this error and have corrected the caption for Figure 2 accordingly. Specifically, we now state that the RNAP binding region is highlighted in blue, whilst repressor binding regions are in red.

We have also updated the legends for Figure 3, 5, 7 and 8 to point out that repressor binding regions are in red, activator binding regions in green, and RNAP binding regions in blue.

Figure 3: The number of (0,0) promoters should be shown as well. Moreover, it seems that the counts in Figure 3 add up to about 50 and the text mentions that there are 32 promoters with no sites found, i.e. type (0,0). Adding this in the sum still seems much less than 113 (i.e., the total number of promoters as indicated in the manuscript). Related to this, it seems that Table 2 has clearly less than 113 promoters in it. Where are the remaining ones?

This figure (now Figure 6) has been updated to include these (0,0) architectures. As summarized in Table 1 of the main text, we only include in Figure 6(B) those regulated promoters which have at least one newly discovered binding site. The remaining 18 promoters that are not accounted for in Figure 6(B) were those we deemed “inactive” as we were unable to recover an RNAP binding site in the growth conditions we explored here. Further discussion regarding those promoters for which we did not recover an active RNAP binding site is now included in Appendix 3 subsection “Examination of promoters for which no RNAP site was found”.

Figure 4: The authors state about the results in Figure 4: "In each of the cases shown in the figure, prior to the work presented here, these promoters had no regulatory information in relevant databases such as EcoCyc (Keseler et al., 2016) and RegulonDB (Santos- 318 Zavaleta et al., 2019)."This is simply not true. If you check EcoCyc for these genes you find:i) idnK regulated by CRP, IdnR and GtnR.ii) leuABCD regulated by LeuO, slyA, LrhA, and RcsB-BgU.iii) maoP regulated by hdfR.iv) rspA regulated by CRP and YdfH.Only for yjjJ, aphA, and ybjX nothing was previously reported. It should also be noted that except for the rspA promoter, the previously reported regulatory interactions were generally not identified by the method.Finally, it seems that there is no mass-spec data for yjjJ. Is the identification of marA as the regulator based on motif matching then?

We acknowledge and regret this mistake in claiming that these were newly discovered regulatory architectures. As an explanation, this figure was meant to display examples of data from a range of regulatory architectures, regardless of whether or not they were previously known, and the line referenced by the reviewer was intended to describe the following figure, but erroneously was used to describe the previous Figure 4. It was certainly not our intention to falsely over-state our discoveries. We have amended the previous Figure 4 caption to remove the erroneous text.

Unrelated to this comment, but as part of an overall push to make the methodology and results clearer, this figure has now replaced by a new Figure 4, listing all TF binding sites for which we have data. The new Figure 4 better encapsulates our point here: that this method is able to recover a myriad of regulatory architectures. Furthermore, MarA was identified as a regulator of *yjjJ* through motif matching to a putative binding site. A note to this effect is in the “Evidence” column of Table 2.

Table 2: This table is unclear. Does the "architecture" of a promoter (first column) correspond to what is inferred in this study, to what is known in the literature, or to a combination of both? For the newly found sites it should be listed whether the binding TF was identified and (if so) what it is. For the fourth column, “literature binding sites”, it is not clear whether this is the total number of known literature sites or only the number of known sites that were successfully identified here – both should be listed. It is not clear what the fifth column “identified binding sites” refers to. The evidence column is also unclear. Is this evidence for the newly found sites or the literature sites?Just to illustrate the confusion: consider the well-known lac operon. In Table 2 the lac operon (called lac as opposed to lacZYA, which is used elsewhere in the manuscript) is reported to have only 1 known literature site, whereas it is well established that it has a site for CRP and multiple sites for LacI. In the final column it is claimed that there is mass-spec evidence for LacI binding. However, on the website, the lac operon does not occur at all and in Appendix 2—figure 1 only the CRP site is reported. What makes this all even more confusing is that none of the experimental conditions included lactose or another inducer of the operon. As such, the lac operon should be highly repressed and so we expect very little effect of mutating the CRP site, and strong effects of abolishing the LacI site.In multiple cases in the table, less literature information is reported than it actually exists: e.g. tar is a reported target of FNR, uvrD is a reported target of LexA, cra is a reported target of PhoB, and so on. The authors should make clear what literature information is shown here.The authors should clear up this information so as to make it unambiguous and consistent across tables, figures, and website.

We thank the reviewer for their comment and hope to address it in this resubmitted manuscript. Our 160 bp promoter regions (with 45 bp downstream of the annotated TSS used) does not always encapsulate all possible regulatory mechanisms, especially long-distance binding of repressors, as is the case in the *lac* operon. The regulatory architectures that we list in Table 2 in the main text reflect only the binding sites that we are able to recover within our 160 bp “mutation” window that was analyzed for each promoter in this study. While this is a clear limitation of the method, in that we are only able to recover regulation in a rather small, 160 bp region, we are encouraged by previous work that found that the majority of annotated regulation is found within 100 bp of the TSS (see Rydenfelt et al., 2014b).

In that study (a figure from which is displayed here as Author response image 1 and is further discussed in Major Comment 6 in this response letter), Rydenfelt *et al.* were able to show, using TSS and binding site positions in RegulonDB, that ”75% of all reported TF interactions in RegulonDB (v8.5) take place within 100 bp of the transcription start site.” Regardless of this close proximity to the TSS, however, repressor and activator binding sites do have distinctly different profiles, as activator binding sites tend to reside slightly further upstream of the TSS compared to repressor binding sites.

Furthermore, we do not include binding sites in this analysis that lack a known binding site location or that are listed as “low-evidence” in EcoCyc, which generally means that they were found only by similarity to a consensus binding site or through gene expression analysis. We agree with the reviewers that our rationale for including or excluding binding sites, as well as our organization of Table 2, was inadequate in our previous submission. To amend this shortcoming, we have added a new section to the appendix; namely, subsection “Explanation of included binding sites” in Appendix 4, which provides our explanations, for each relevant gene, regarding binding sites that appear in the regulatory information in RegulonDB or EcoCyc that are not addressed in Reg-Seq as a ”literature binding site” in Table 2. As one specific example from *lacZYA*, the mutagenized region extends from the TSS (the primary TSS p1) to 75 base pairs upstream of the TSS. The location of the mutagenized region excludes the LacI sites, while including a single CRP binding site, a MarA binding site, and two HNS binding sites. The expression from *marA* is expected to be low, as we do not grow the cells in the presence of salicylate or antibiotic stress and so we do not expect to observe the MarA site. In fact, the precursor of the Reg-Seq experiment, Sort-Seq, mutagenized and studied the same 75-base pair region, and only observed binding by CRP (Kinney et al., 2010). As such, we only include CRP in Table 2, the regulatory cartoons, or the analysis of false positives and false negatives discussed in Appendix 2 subsection “False positive and false negative rates”.

In our Reg-Seq experiments, there were several genes for which we failed to identify any repressor or activator binding sites, and indeed a limitation of this method (which we also discuss in the Discussion section of the main text), is our inability to study those binding sites outside of our narrow, 160 bp window.

We also have amended Table 2 in the main text in an attempt to clarify what is meant by each column. We now indicate in the table caption that the first column is meant to reflect the total number of binding sites that could be seen in the Reg-Seq data, regardless of whether there was previous evidence for their existence. We also note that binding sites which are not in the mutagenesis window or are otherwise not expected to appear in the Reg-Seq data, as explained in Appendix 4 subsection “Explanation of included binding sites”, are not counted in this tally. Similarly, we also emphasize that the literature sites column contains those sites that are both expected to be and are, in actuality, observed in the Reg-Seq data.